# Continual Segmentation under Joint Nonstationarity

**Prashant Pandey** [* 1]  **Himanshu Kumar** [* 2]  **Devineni Sri Venkatraya Chowdary** [3]  **Brejesh Lall** [1]

## Abstract

Evolving data streams induce *joint nonstationarity* in continual semantic segmentation, where semantic classes, input distributions, and supervision availability change simultaneously over time. This setting reflects practical structured prediction systems, yet remains largely unexplored in prior continual learning work, which typically studies these factors in isolation. We formalize continual segmentation under coupled class, domain, and label shifts and investigate learning in heterogeneous dense prediction environments with limited annotations and abundant unlabeled data. To address instability and overfitting arising from few-shot supervision under distribution drift, we introduce *gradient-adaptive stabilization*, a parameter-wise regularization mechanism implemented via gradient-scaled stochastic perturbations that promotes a principled stability–plasticity tradeoff. We further leverage unlabeled data through semi-supervised learning and introduce *prototype anchored supervision* that validates pseudo-labels via joint confidence and prototype consistency. Together, these mechanisms enable learning under joint nonstationarity in continual segmentation. Extensive empirical evaluation across class-incremental, domain-incremental, and few-shot regimes demonstrates consistent improvements over prior methods in heterogeneous structured prediction settings. Our results expose fundamental failure modes of existing continual segmentation approaches and provide insight into learning robust dense predictors in dynamically evolving environments. Our code is available at https://github.com/prinshul/JASCL.git.

*Equal contribution [1]Bharti School of Telecommunications Technology & Management, Indian Institute of Technology, Delhi, India [2]Yardi School of Artificial Intelligence, Indian Institute of Technology, Delhi, India [3]Department of Computer Science and Engineering, Indian Institute of Technology (Indian School of Mines), Dhanbad, India. Correspondence to: Prashant Pandey <bsz178495@iitd.ac.in >.

*Proceedings of the 43rd International Conference on Machine Learning*, Seoul, South Korea. PMLR 306, 2026. Copyright 2026 by the author(s).

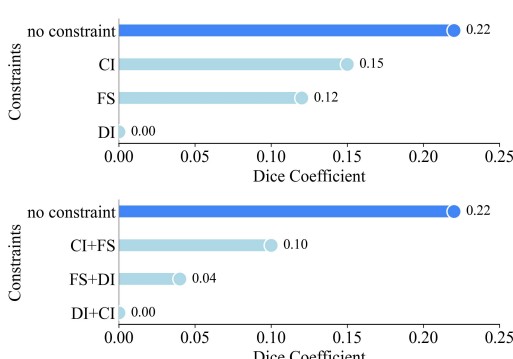

*Figure 1.* Class-incremental (CI), few-shot (FS), and domain-incremental (DI) constraints all lead to significantly reduced Dice scores compared to the unconstrained fine-tuning ("no constraint") on the common base model.

## 1 Introduction

The pursuit of adaptive intelligent systems requires learning from data streams whose underlying distributions evolve over time. While continual learning (CL) (Wang et al., 2024; Yuan & Zhao, 2024b) has made progress across a range of settings, a critical gap remains in addressing realistic dense prediction problems such as semantic segmentation, where multiple sources of nonstationarity arise simultaneously. In applications including autonomous driving and medical image analysis, segmentation models encounter sequential data characterized by both evolving semantic label spaces (class-incremental learning, CIL (Zhou et al., 2024)) and shifting input distributions (domain-incremental learning, DIL (Mirza et al., 2022)), often under severely limited supervision.

We consider a sequence of learning sessions $\{\mathcal{S}_t\}_{t=0}^{T}$, where each session is associated with a joint distribution $\mathcal{D}_t(x, y)$ over inputs $x$ and pixel-wise labels $y$, a class set $\mathcal{C}_t$, and a labeled sample budget $N_t \ll N_0$ for $t > 0$. In realistic settings, these quantities evolve jointly, with $\mathcal{D}_t \neq \mathcal{D}_{t-1}$, $\mathcal{C}_t \neq \mathcal{C}_{t-1}$, and $N_t \ll N_0$, giving rise to *joint nonstationarity*. Existing CL formulations typically isolate these factors, studying class increments, domain shifts, or data scarcity independently, leaving their coupled interaction largely unexplored.

This discrepancy between idealized CL scenarios and practical structured prediction manifests as severe performance degradation. In the presence of joint shifts, models suffer from catastrophic forgetting, unstable adaptation, and

rapid overfitting to few-shot supervision. Figure 1 shows that unconstrained fine-tuning achieves the strongest performance, whereas class-incremental (CI), few-shot (FS), and domain-incremental (DI) constraints each induce substantial accuracy drops. These constraints stress models in distinct ways: CI alters decision boundaries, FS produces noisy gradients and overfitting, and DI distorts feature representations, leading to large degradations relative to the unconstrained baseline. Figure 2 further shows that naive fine-tuning is unstable across architectures: freezing most layers inhibits learning new concepts, while fully unfreezing disrupts previously acquired representations. These observations highlight that effective continual segmentation requires preserving earlier structure while enabling controlled adaptation under joint nonstationarity.

Few-shot learning (Tao et al., 2020; Qiu et al., 2023; Tian et al., 2024) and semi-supervised learning (Kang et al., 2023b; Cui et al., 2024) offer partial remedies by exploiting limited labels and abundant unlabeled data. However, under joint nonstationarity, pseudo-labeling becomes unreliable: initial model biases propagate across sessions, progressively degrading learning. Recent studies and surveys (Zhang et al., 2025; Yuan & Zhao, 2024a; Xu et al., 2024; Kwak et al., 2025; Zhu et al., 2026; Xue et al., 2025) emphasize the necessity of CL formulations that jointly account for class evolution, domain drift, and label scarcity, yet principled solutions for dense prediction remain scarce.

Motivated by this gap, we study continual semantic segmentation under coupled class, domain, and supervision shifts. Each incremental session introduces novelty, new classes, new domains, or both, while previously observed class and domain pairs do not recur. Learning proceeds with few labeled samples per session and access to unlabeled data drawn from the current distribution, capturing realistic nonstationary structured prediction.

Limited supervision under distribution drift destabilizes optimization and accelerates overfitting. To mitigate this, we introduce *gradient-adaptive stabilization* (GAS), a parameter-wise regularization mechanism implemented via gradient-scaled stochastic perturbations that explicitly balances stability and plasticity during adaptation. We further introduce *prototype anchored supervision* (PAS) in a semi-supervised setting, where class prototypes learned across sessions are used to validate pseudo-labels through joint confidence and feature-space consistency, suppressing error propagation under domain shift and improving knowledge retention over time. Extensive empirical evaluation across class-incremental, domain-incremental, few-shot, and semi-supervised regimes demonstrates consistent gains over prior approaches.

Our key contributions to continual semantic segmentation under joint nonstationarity are fourfold: **(i)** We formalize a

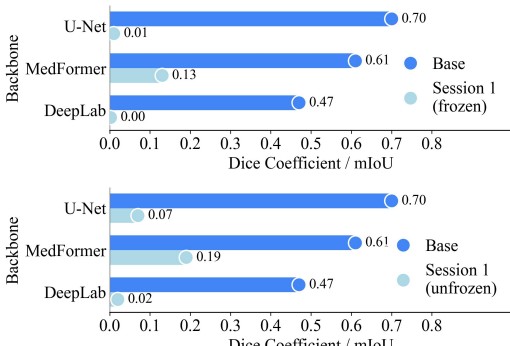

*Figure 2.* Fine-tuning the Session 0 base model in incremental Session 1 causes all backbones to drop in performance, whether their weights are partially frozen (top) or fully unfrozen (bottom).

realistic continual segmentation setting with coupled class, domain, and supervision shifts, exposing fundamental failure modes of existing continual learning methods under joint nonstationarity. **(ii)** We propose *gradient-adaptive stabilization* (GAS), a parameter-wise regularization mechanism that balances stability and plasticity via gradient-scaled stochastic perturbations, enabling robust adaptation under distribution drift. **(iii)** We introduce *prototype anchored supervision* (PAS), which validates pseudo-labels through joint confidence and prototype consistency, mitigating error propagation and improving learning from unlabeled data. **(iv)** Extensive empirical evaluation demonstrates consistent improvements across class-incremental, domain-incremental, and few-shot regimes, with strong generalization across heterogeneous architectures including 3D U-Net (Ronneberger et al., 2015), DeepLabv3+ (Chen et al., 2018), and Transformer/MedFormer (Gao et al., 2022), surpassing large-scale pretrained models such as SAM (Kirillov et al., 2023; Kerssies et al., 2024).

## 2 Related Work

**Continual Semantic Segmentation under Distribution Shift.** Early class-incremental segmentation methods addressed background shift via distillation and classifier initialization (MiB (Cermelli et al., 2020)), while subsequent approaches incorporated auxiliary supervision or external priors, e.g., CLIP-CT (Zhang et al., 2023) with pseudo-labeling and CLIP-guided heads. Domain-incremental learning has been studied through architectural factorization, with MDIL (Garg et al., 2022) decomposing parameters into domain-invariant and domain-specific components. Regularization-based strategies include uncertainty-aware penalties (UCL (Ahn et al., 2019)), flat-minima optimization (C-Flat (Bian et al., 2024)), parameter-space perturbations (STAR (Eskandar et al., 2025)), and uncertainty- and class-balanced pseudo-label reweighting (UCB (Liang et al., 2025)).

**Few-shot Class-Incremental Learning.** Few-shot con-

tinual learning has been explored via prototype-based and constraint-driven formulations. PIFS (Cermelli et al., 2021) and FeCAM (Goswami et al., 2023) leverage prototypes for exemplar-free adaptation, Subspace (Akyürek et al., 2021) constrains novel classifiers to base-class subspaces, and NC-FSCIL (Yang et al., 2023) fixes classifiers as simplex equiangular tight frames. Other approaches allocate future embedding capacity (FACT (Zhou et al., 2022)), synthesize prior data (Gen-Replay (Liu et al., 2022)), address partial annotations (GAPS (Qiu et al., 2023)), decompose networks via soft masks (SoftNet (Kang et al., 2023a)), combine pseudo-labeling with distillation (FSCIL-SS (Jiang et al., 2023)), or mine relevant base classes (BCM (Sakai et al., 2024)). C-FSCIL (Hersche et al., 2022) further employs hyperdimensional representations with quasi-orthogonal prototypes.

**Semi-Supervised and Pseudo-Labeling.** Semi-supervised continual learning methods exploit unlabeled data while mitigating error accumulation, including coreset selection (RETRIEVE (Killamsetty et al., 2021)), soft nearest-neighbor stabilization (NNCSL (Kang et al., 2023b)), uncertainty-aware distillation with class equilibrium (UaD-CE (Cui et al., 2024)), and confidence–distribution based pseudo-label selection (CSL (Liu & Liu, 2025)).

**Optimization and Representation Learning under Distribution Shift.** Related advances include supervised and self-supervised contrastive learning (SupCL (Khosla et al., 2020), SimCLR (Chen et al., 2020), UnSupCL-HNM (Robinson et al., 2021)), unifying views of multi-task and meta-learning (Wang et al., 2021), and multi-task representation analyses of few-shot learning (Bouniot et al., 2022). CLIP-driven universal segmentation (Liu et al., 2023) leverages text embeddings (Radford et al., 2021) for partial labels, while HALO (Franco et al., 2024) applies hyperbolic representations for pixel-level active learning under domain shift.

*Prior work typically addresses class-incremental learning, domain shift, or few-shot supervision in isolation. We instead formalize their coupled interaction and develop learning mechanisms for continual segmentation under joint nonstationarity.*

# 3 Jointly Anchored and Stabilized Continual Learning (JASCL)

**Problem Setup (Joint Nonstationarity).** We formalize continual semantic segmentation under joint nonstationarity as a sequence of sessions $\{\mathcal{S}_t\}_{t=0}^{T}$, where each session is characterized by a class set $\mathcal{C}_t$ and domain distribution $\mathcal{D}_t$. At session $t$, the model observes labeled data $\mathbb{D}_t = \{(x_i, y_i)\}_{i=1}^{N_t}$ with $x_i \sim \mathcal{D}_t$ and pixel-wise labels $y_i \in \mathcal{C}_t$. The base session $\mathcal{S}_0$ provides abundant supervision

$(N_0)$, while subsequent sessions satisfy $N_t \ll N_0$ and are few-shot.

Across sessions, both semantic classes and domains may evolve, yielding three realistic cases: $(i)\, \mathcal{C}_t = \mathcal{C}_{t'}, \mathcal{D}_t \neq \mathcal{D}_{t'}$ (domain shift); $(ii)\, \mathcal{C}_t \neq \mathcal{C}_{t'}, \mathcal{D}_t = \mathcal{D}_{t'}$ (class evolution); and $(iii)\, \mathcal{C}_t \neq \mathcal{C}_{t'}, \mathcal{D}_t \neq \mathcal{D}_{t'}$ (joint shift). Previously observed class–domain pairs are not revisited. Each incremental session uses $N_t = K|\mathcal{C}_t|$ labeled samples with $K \in \{5, 10, 20, 30\}$ and may additionally access unlabeled data $\mathcal{U}_t = \{x_j^{(u)}\}_{j=1}^{M_t}$, where $M_t \gg N_t$ and $x_j^{(u)} \sim \mathcal{D}_t$. Our goal is to learn a segmentation model that adapts across sessions while retaining prior knowledge under this coupled class, domain, and supervision shift. To this end, JASCL combines gradient adaptive stabilization (GAS) for optimization stability with prototype anchored supervision (PAS) to anchor semi-supervised learning under nonstationarity.

## 3.1 Gradient Adaptive Stabilization (GAS)

To prevent catastrophic forgetting and overfitting on few-shot data while adapting to domain shifts, we introduce *gradient-adaptive stabilization* (GAS) that regulates perturbation magnitude using parameter gradients as a proxy for curvature. Let $\theta \in \mathbb{R}^d$ denote the flattened vector of all model parameters, where $d$ is the total number of parameters. At session $t$, define the squared gradient $G_i = (\partial\mathcal{L}/\partial\theta_i)^2$ where $\mathcal{L}$ is the loss function and $\theta_i$ is the $i$-th parameter. Given classifier layer $F$ of segmentation model with weight matrix $\mathbb{W} \in \mathbb{R}^{m \times n}$, we maintain gradient buffer $G$ that accumulates squared gradients $\{G_{ij}\}$ and compute inverse gradients $G_{ij}^{-1} = 1/(G_{ij} + \epsilon)$ where $\epsilon > 0$ ensures numerical stability. The normalized noise scale is:

$$\tilde{G}_{ij}^{-1} = \frac{1 + G_{ij}^{-1} - \min(G^{-1})}{1 + \max(G^{-1}) - \min(G^{-1})} \in (0, 1] \quad (1)$$

where $\min(G^{-1})$ and $\max(G^{-1})$ are the minimum and maximum values over all elements in the inverse gradient matrix. The weights are perturbed as $\tilde{\mathbb{W}} = \mathbb{W} + \tilde{G}^{-1} \odot \mathcal{N}(0, I)$, where $\odot$ denotes element-wise multiplication and $\mathcal{N}(0, I)$ is standard Gaussian noise. This mechanism injects minimal noise into critical parameters with large gradients actively contributing to learning, while injecting higher noise into parameters with small gradients that may be overfitting.

**Theoretical analysis.** We establish optimality of GAS through Bayesian posterior approximation (a variational inference approach to approximate intractable posteriors). Under diagonal Gaussian approximation, the approximate posterior after session $t$ is $q^{(t)}(\theta) = \mathcal{N}(\mu^{(t)}, \mathrm{diag}(\sigma_1^{(t)2}, \ldots, \sigma_d^{(t)2}))$ where $\mu^{(t)} \in \mathbb{R}^d$ is the mean vector and $\sigma_i^{(t)2}$ is the variance of parameter $\theta_i$. The true posterior under Laplace approximation

(second-order Taylor approximation of the posterior) is $\pi^{(t)}(\theta) = \mathcal{N}(\theta^{(t)}, \text{diag}(1/F_{11}^{(t)}, \ldots, 1/F_{dd}^{(t)}))$ where $\theta^{(t)}$ is the optimal parameter vector at session $t$ and $F_{ii}^{(t)}$ is the $i$-th diagonal element of the Fisher Information Matrix (FIM) at session $t$, defined as $F_{ij} := \mathbb{E}_{(\mathbf{x},\mathbf{y})\sim P}\left[\frac{\partial \log p(\mathbf{y}|\mathbf{x};\theta)}{\partial \theta_i}\frac{\partial \log p(\mathbf{y}|\mathbf{x};\theta)}{\partial \theta_j}\right]$ where $P$ is the data distribution and $p(\mathbf{y}|\mathbf{x};\theta)$ is the model's likelihood. The FIM measures parameter sensitivity to data. A large $F_{ii}$ indicates parameter $\theta_i$ significantly affects model predictions.

**Assumption 3.1** (Gradient-Fisher Correspondence). For parameter $i$ at session $t$, the squared gradient $g_i^{(t)} = \partial\mathcal{L}/\partial\theta_i$ approximates the diagonal Fisher: $|g_i^{(t)}|^2 = F_{ii}^{(t)}(1 + \epsilon_i)$ where $\epsilon_i$ is an approximation error satisfying $|\epsilon_i| \leq \delta$ for some small $\delta \in (0,1)$.

**Assumption 3.2** (Domain Shift Model). Under domain shift from session $t$ to $t+1$, the diagonal Fisher elements change as $F_{ii}^{(t+1)} = F_{ii}^{(t)} + \Delta F_{ii}$ where $\Delta F_{ii}$ is the change in Fisher information for parameter $i$, and $\frac{1}{d}\sum_{i=1}^{d}(\Delta F_{ii})^2 \geq \gamma_F^2\Delta_t^2$ for constants $\gamma_F > 0$ (Fisher sensitivity) and $\Delta_t > 0$ (domain shift magnitude).

**Proposition 3.3** (GAS Achieves Lower KL Under Anisotropic Structure). *Let $\tilde{G}_i^2 = c/F_{ii}$ where $c > 0$ is a normalization constant. Let $q_{\text{iso}}$ be an isotropic posterior (uniform variance across parameters) with $\sigma^2 = c/\bar{F}$ where $\bar{F} = \frac{1}{d}\sum_{i=1}^{d}F_{ii}$ is the average Fisher information. Consider the informed prior $p = \mathcal{N}(\theta^*, \text{diag}(1/F_{11}, \ldots, 1/F_{dd}))$ where $\theta^*$ is the optimal parameter vector. If Fisher information is heterogeneous across parameters, i.e., $\text{Var}(F_{ii}) > 0$, then $\text{KL}(q_{\text{GAS}}\|p) < \text{KL}(q_{\text{iso}}\|p)$ where $\text{KL}(\cdot\|\cdot)$ is the Kullback-Leibler divergence (measures difference between probability distributions).*

*When Fisher information varies across parameters, uniform noise cannot match the per-parameter variance of the Laplace approximation; GAS resolves this by scaling noise inversely to $F_{ii}$, achieving strictly lower KL divergence.*

**Corollary 3.4** (GAS Generalization Bound). *Under Proposition 3.3, GAS achieves tighter PAC-Bayes bound (McAllester, 1999) (a framework for deriving generalization guarantees) than isotropic noise. With informed prior $p$, for expected risk $R(q) = \mathbb{E}_{\theta\sim q}[\mathbb{E}_{(\mathbf{x},\mathbf{y})\sim P}[\ell(f_\theta(\mathbf{x}),\mathbf{y})]]$ (where $\ell$ is the loss function and $f_\theta$ is the model parameterized by $\theta$) and empirical risk $\hat{R}(q) = \mathbb{E}_{\theta\sim q}[\frac{1}{n}\sum_{i=1}^{n}\ell(f_\theta(\mathbf{x}_i),\mathbf{y}_i)]$ (computed over $n$ training samples), with probability at least $1-\delta$:*

$$R(q_{\text{GAS}}) \leq \hat{R}(q_{\text{GAS}}) + \sqrt{\frac{\text{KL}(q_{\text{GAS}}\|p) + \log(2\sqrt{n}/\delta)}{2n}}$$
(2)

*where $\text{KL}(q_{\text{GAS}}\|p) < \text{KL}(q_{\text{iso}}\|p)$ when Fisher is heterogeneous. Assuming equal empirical risk, the general-*

ization gap for GAS is smaller: $R(q_{\text{GAS}}) - \hat{R}(q_{\text{GAS}}) < R(q_{\text{iso}}) - \hat{R}(q_{\text{iso}})$.

*Since the generalization bound increases with $\text{KL}(q\|p)$, the strictly lower KL achieved by GAS directly yields a tighter bound on the expected risk.*

**Assumption 3.5** (Heterogeneous Curvature). The Hessian $H = \nabla^2\mathcal{L}(\theta) \in \mathbb{R}^{d\times d}$ (second derivative matrix of the loss) has eigenvalues $\lambda_1 \geq \lambda_2 \geq \cdots \geq \lambda_d > 0$ with condition number $\kappa := \lambda_1/\lambda_d > 1$ (ratio of largest to smallest eigenvalue).

Static regularization methods (e.g. Dropout, weight decay (Mirzadeh et al., 2020), noise injection (Orvieto et al., 2023)) set variance independent of current-task Fisher information $F_{ii}^{(t+1)}$.

**Theorem 3.6** (GAS vs. Static Regularization). *Let $q_{\text{GAS}}$ be the GAS posterior with noise variance $\tilde{G}_i^2 \propto 1/g_i^2$ (inversely proportional to squared gradients) and $q_{\text{static}}$ be a static regularization posterior with fixed variance $\sigma_{\text{static}}^2$ determined from session $t-1$. Under Assumptions A.10 and A.11, when domain shift magnitude satisfies $\Delta_t > C\lambda\delta/\gamma_F$ for regularization strength $\lambda$ and constants $C, \delta, \gamma_F$, we have $\text{KL}(q_{\text{GAS}}\|\pi^{(t)}) < \text{KL}(q_{\text{static}}\|\pi^{(t)})$ where $\pi^{(t)}$ is the true posterior at session $t$.*

Memory-based regularization methods (e.g. EWC (Kirkpatrick et al., 2017), UCL (Ahn et al., 2019)) set variance using Fisher information from sessions $0, \ldots, t$ to regularize session $t+1$.

**Theorem 3.7** (GAS vs. Memory-Based Regularization). *Let $q_{\text{memory}}$ be a memory-based regularization posterior using Fisher information accumulated from previous sessions. Define the Fisher mismatch $\rho_i = (F_{ii}^{(t)} - \bar{F}_{ii}^{(hist)})/F_{ii}^{(t)}$ where $\bar{F}_{ii}^{(hist)}$ is the historical Fisher estimate (weighted average from past sessions). Under Assumptions A.10 and A.11, when $\frac{1}{d}\sum_i \rho_i^2 > O(\delta^2)$ (Fisher mismatch exceeds approximation error), we have $\text{KL}(q_{\text{GAS}}\|\pi^{(t)}) < \text{KL}(q_{\text{memory}}\|\pi^{(t)})$.*

*Fisher estimates accumulated from past sessions become misaligned with the current session's Fisher information under domain shift; GAS recomputes noise scales from instantaneous gradients, maintaining alignment with $F_{ii}^{(t)}$.*

Adversarial flatness methods (e.g. SAM (Foret et al., 2021), C-Flat (Bian et al., 2024), STAR (Eskandar et al., 2025)) seek flat minima via worst-case perturbations.

**Theorem 3.8** (GAS vs. Adversarial Flatness Methods). *Let $\mathcal{L}_{\text{adv}}(\theta) = \max_{\|\delta\|_2\leq\rho}\mathcal{L}(\theta + \delta)$ be the adversarial loss maximized over an $\ell_2$-ball of radius $\rho$ (perturbation budget). The worst-case perturbation concentrates on high-curvature directions, yielding expected loss increase*

$\mathbb{E}[\mathcal{L}_{\text{adv}}] - \mathcal{L}(\theta) = O(\rho^2 \lambda_{\max})$ *where* $\lambda_{\max}$ *is the maximum Hessian eigenvalue. For GAS with inverse gradient scaling, the expected loss increase is* $O(\rho^2 \bar{\lambda})$ *where* $\bar{\lambda} = \frac{1}{d} \sum_i \lambda_i$ *is the average curvature. Under Assumption* A.12 *with* $\kappa > d$, *GAS achieves lower expected loss increase:* $\mathbb{E}[\mathcal{L}_{\text{GAS}}] - \mathcal{L}(\theta) < \mathbb{E}[\mathcal{L}_{\text{adv}}] - \mathcal{L}(\theta)$ *by factor up to* $\kappa/d$.

*Adversarial perturbations concentrate on the maximum curvature direction, incurring loss increase* $O(\rho^2 \lambda_{\max})$; *inverse gradient scaling distributes perturbations toward low-curvature directions, reducing this to* $O(\rho^2 \bar{\lambda})$.

The proofs for Proposition 3.3, Corollary 3.4, and Theorems 3.6–3.8 are provided in Appendix A. Please refer to Table 9 and Appendix C for *robustness and sensitivity analysis* of GAS.

## 3.2 Prototype Anchored Supervision (PAS)

To leverage abundant unlabeled data $\mathcal{U}_t$ while mitigating error propagation, we introduce prototype anchored supervision (PAS) within a mean-teacher framework (Tarvainen & Valpola, 2017) (a semi-supervised learning approach where a teacher network generates pseudo-labels for unlabeled data to train a student network). For each class $c \in \mathcal{C}_t$, we extract a compact prototype from the feature space. Given a trained model with intermediate feature extractor $\phi$ and classifier $F$, for each labeled sample $(x_i, y_i)$ with embedding $E_i = \phi(x_i) \in \mathbb{R}^{D \times H \times W}$ (where $D$ is the feature dimension, $H, W$ are spatial dimensions), we extract class-specific features $\mathcal{F}_c^{(i)} = \{E_i : y_i = c\}$ (all features where ground-truth label is class $c$) and compute the per-sample prototype as $p_c^{(i)} = \frac{1}{|\mathcal{F}_c^{(i)}|} \sum_{\mathbf{f} \in \mathcal{F}_c^{(i)}} \mathbf{f}$. Across all samples in session $t$ containing class $c$ (denoted $NS_c \subseteq N_t$), the class prototype is:

$$P_c = \frac{1}{NS_c} \sum_{j \in NS_c} \frac{p_c^{(j)}}{||p_c^{(j)}||_2} \quad (3)$$

where $|| \cdot ||_2$ is the $\ell_2$-norm (Euclidean norm).

*These prototypes serve as class anchors in feature space, enabling pseudo-label validation via feature-space consistency rather than confidence alone.*

For unlabeled input $x_j^{(u)}$, student network $M_s$ and teacher network $M_t$ generate predictions $\hat{y}_s, \mathcal{F}_s = M_s(x_j^{(u)})$ and $\hat{y}_t, \mathcal{F}_t = M_t(x_j^{(u)})$, where $\hat{y}_s, \hat{y}_t \in \mathbb{R}^{C \times H \times W}$ are predicted class probabilities and $\mathcal{F}_s, \mathcal{F}_t \in \mathbb{R}^{D \times H \times W}$ are feature representations. For each pixel $(p, q)$, we compute predicted class $c'(p, q) = \arg\max_c(\text{softmax}(\hat{y}(p, q)))$, confidence $\text{conf}(p, q) = \max(\text{softmax}(\hat{y}(p, q)))$ (maximum probabil-

ity over classes), and cosine similarity to prototype:

$$\text{sim}(p, q) = \frac{\mathcal{F}(p, q) \cdot P_{c'(p,q)}}{||\mathcal{F}(p, q)||_2 ||P_{c'(p,q)}||_2} \quad (4)$$

where $\cdot$ denotes dot product. A pseudo-label at $(p, q)$ is retained only if $\text{valid}(p, q) = (\text{conf}(p, q) > \tau_{\text{conf}}) \wedge (\text{sim}(p, q) > \tau_{\text{sim}})$ where $\tau_{\text{conf}}, \tau_{\text{sim}} \in [0, 1]$ are thresholds. The consistency loss operates on validated pseudo-labels:

$$\mathcal{L}_{\text{consistency}} = \frac{1}{|\mathcal{V}|} \sum_{(p,q) \in \mathcal{V}} ||\hat{y}_s(p, q) - \hat{y}_t(p, q)||_2^2 \quad (5)$$

where $\mathcal{V} = \{(p, q) : \text{valid}_s(p, q) \wedge \text{valid}_t(p, q)\}$ represents pixels validated by both models.

**Theoretical analysis.** We model pseudo-label dynamics through error rates. Let $\epsilon_s(t)$ and $\epsilon_t(t)$ denote expected fractions of incorrect pseudo-labels produced by student and teacher at iteration $t$. With unlabeled data $\mathcal{D}_u$ of size $n_u$ and pseudo-label weight $\gamma \in (0, 1)$ (controls the influence of pseudo-labels on learning), define coverage $f \in [0, 1]$ as the fraction of pseudo-labels retained, and precision $\rho \in [0, 1]$ as the probability a retained pseudo-label is correct.

**Assumption 3.9** (Labeled Data Error). There exists a constant $\epsilon_0 \in (0, 1)$ representing the irreducible error rate of the student when trained solely on labeled data $\mathcal{D}_l$ (the best achievable error without unlabeled data).

**Assumption 3.10** (Linear Error Mixing). When trained on mixture of ground-truth labels (weight $1 - \gamma$) and pseudo-labels (weight $\gamma$), the student's expected error is: $\epsilon_s(t+1) = (1 - \gamma)\epsilon_0 + \gamma\epsilon_t(t)$.

**Assumption 3.11** (EMA Error Inheritance). The teacher's pseudo-label error inherits from student via exponential moving average (EMA) with decay $\alpha \in [0, 1)$ (a momentum-based update where teacher parameters slowly track student parameters): $\epsilon_t(t+1) = \alpha\epsilon_t(t) + (1-\alpha)\epsilon_s(t+1)$.

**Proposition 3.12** (Error Dynamics Recurrence). *Under Assumptions* B.1–B.3, *the teacher error satisfies first-order linear recurrence:* $\epsilon_t(t+1) = \lambda\epsilon_t(t) + (1-\alpha)(1-\gamma)\epsilon_0$ *where* $\lambda := \alpha + (1-\alpha)\gamma < 1$ *is the contraction factor.*

**Theorem 3.13** (Asymptotic Error Without Prototype Anchored Supervision (PAS)). *Let* $\gamma \in (0, 1)$ *and* $\alpha \in [0, 1)$. *Then* $\lambda = \alpha + (1-\alpha)\gamma < 1$, *and teacher error converges:* $\lim_{t \to \infty} \epsilon_t(t) = \epsilon_0$. *Without PAS, semi-supervised learning provides no asymptotic improvement over purely supervised learning.*

**Theorem 3.14** (Asymptotic Error With PAS). *Under Assumptions* B.1–B.3, *with PAS that achieves coverage* $f$ *and precision* $\rho$, *the teacher error converges to:*

$$\epsilon_\infty(f, \rho) = \frac{(1 - f\gamma)\epsilon_0}{1 - f\gamma(1 - \rho)}. \quad (6)$$

*Furthermore: (i) $\epsilon_\infty < \epsilon_0$ if and only if $\rho > 1 - \frac{1-f\gamma}{f\gamma}$, (ii) $\epsilon_\infty$ is strictly decreasing in $\rho$ when $f > 0$, (iii) for any $\rho > \pi$ (base rate), PAS strictly improves over the no PAS baseline, and (iv) higher precision always reduces asymptotic error.*

*When $\rho = 0$ (all accepted pseudo-labels incorrect), $\epsilon_\infty = \epsilon_0$, recovering the no PAS baseline; when $\rho = 1$ (all accepted pseudo-labels correct), $\epsilon_\infty = (1 - f\gamma)\epsilon_0 < \epsilon_0$, achieving the maximum possible reduction.*

We also prove that PAS performs better compared with confidence-only, consistency-based (CSL (Liu & Liu, 2025)), and memory bank based (NNCSL (Kang et al., 2023b)) methods. All proofs are provided in Appendix B.

JASCL integrates Gradient Adaptive Stabilization (GAS) with Prototype Anchored Supervision (PAS) to address continual segmentation under joint nonstationarity (Figure 3). At session $t$, training minimizes $\mathcal{L} = \mathcal{L}_{CE}(\mathbb{D}_t) + \lambda_{cons}\mathcal{L}_{consistency}(\mathcal{U}_t)$ (if unlabeled data is available), where $\mathcal{L}_{CE}$ denotes supervised cross-entropy and $\mathcal{L}_{consistency}$ is computed only on PAS-validated pseudo-labels. GAS applies gradient-adaptive perturbations to classifier parameters, providing stability under curvature heterogeneity (Proposition 3.3, Corollary 3.4, Theorems 3.6–3.8), while PAS anchors semi-supervised learning to class prototypes, improving pseudo-label reliability (Theorem 3.13–3.14). Prototypes are replayed at the classifier $F$ ($\mathcal{L}_{proto}$). Together, GAS stabilizes optimization and PAS suppresses error propagation, enabling continual learning across evolving classes, domains, and supervision regimes.

## 4  JASCL Benchmarks

We construct five challenging benchmarks for **3D medical** and **2D natural scene segmentation**, designed to simulate realistic clinical and autonomous driving scenarios with *multiple sessions*, *diverse domains*, and a *large number of novel classes*. Each benchmark features a base learning session on a large dataset followed by incremental sessions with limited labeled data (few-shot) and with significant domain shifts.

**3D Medical JASCL Benchmarks**: We develop three **3D medical** benchmarks using data from **TotalSegmentator** (TS - CT) ((Wasserthal et al., 2023)), **AMOS** (mostly CT with few MRI samples) ((Ji et al., 2022)), **BCV** (CT)((Landman et al., 2015), **MOTS** (CT) ((Zhang et al., 2021)), **BraTS** (MRI) ((Menze et al., 2014)), and **VerSe** (CT) ((Sekuboyina et al., 2021)). All three benchmarks adopt a few-shot learning setup, using 5 training samples per class for incremental sessions, progressing from normal to tumor segmentation.

The three medical benchmarks: (i) **Med JASCL-Disjoint**, a 6-session, 37-class protocol with disjoint classes and domains; (ii) **Med JASCL-Mixed**, a 5-session, 35-class setup allowing recurrence of either classes or domains (but not both) and mixing datasets per session; and (iii) **Med Semi-Supervised-JASCL**, a semi-supervised variant of Med JASCL-Disjoint augmented with 8–30 unlabeled samples per session. Please refer to Table 1 for various classes and domains.

**2D Natural Scene JASCL Benchmarks**: We introduce two benchmarks for **autonomous driving** scenarios using data from **BDD100K** ((Yu et al., 2020)), **Cityscapes** ((Cordts et al., 2016)), and **IDD** ((Varma et al., 2019)). These benchmarks feature a few-shot learning with 10 training samples per class. The two natural scene benchmarks for autonomous driving: (i) **Natural-JASCL**, a 3-session setup using BDD100K, Cityscapes, and IDD, designed to test representation adaptation under domain shifts and class recurrence; and (ii) **Semi-Supervised Natural-JASCL**, a 4-session variant that augments new classes with 400 unlabeled images per class to reflect realistic scenarios with limited annotations but abundant raw data. Details on all datasets, class and domain specifications, and unlabeled data are provided in Appendix F.

## 5  Results and Analysis

We use mean Intersection over Union (mIoU), ranging from 0 to 100 (higher is better), for 2D natural scene benchmarks, and the Dice coefficient (Dice score), ranging from 0 to 1 (higher is better), for 3D medical benchmarks, as evaluation metrics. In each *incremental session*, we evaluate the classes introduced in the current session along with all classes encountered in previous sessions. The goal is to retain previously learned knowledge while effectively acquiring new information, handling data scarcity, and adapting to domain shifts. A *Vanilla* baseline consists of a plain backbone with no mechanisms to handle any constraints. Gen-Replay (Liu et al., 2022) is implemented with a diffusion model adapted from (Dorjsembe et al., 2024). Some baselines act only in incremental sessions and rely on a shared pre-trained base model, whereas others modify training in Session 0, thereby altering base session performance. See Appendix I for comparison with regularization-based methods.

We observe substantial performance drops across all Vanilla models and baselines. Notably, transformer-based models perform comparably to, or sometimes worse than, simpler encoder–decoder architectures like U-Net, and even heavily pre-trained models such as SAM show a clear decline. Across the medical benchmarks, baselines collapse after just two sessions in *Med JASCL-Disjoint* (Table 2), while JASCL with a U-Net backbone sustains strong performance across all five, demonstrating robustness to multiple constraints as shown in Table 2 and Table 3. In *Med JASCL-Mixed*, transformer backbones such as MedFormer, SwinUNetr,

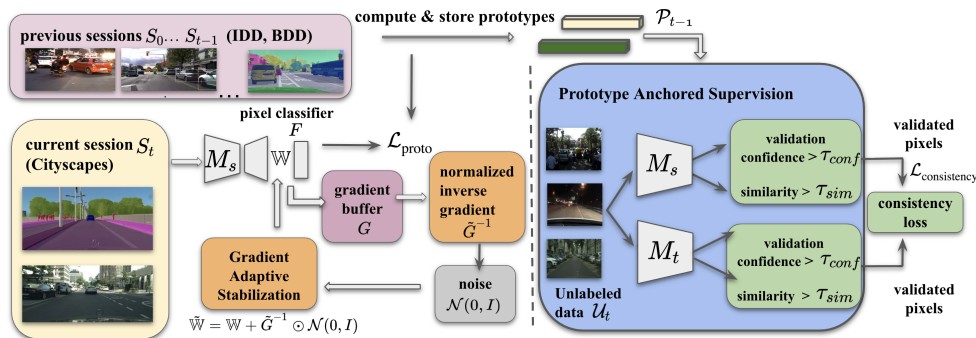

*Figure 3.* At session $t$, gradient adaptive stabilization (GAS) is applied to the decoder's pixel classifier $F$ of $M_s$. With unlabeled data, a mean-teacher model $M_t$ generates pseudo-labels via similarity matching ($\tau_{sim}$), and only high-confidence predictions ($\tau_{conf}$) contribute to a consistency loss. Prototype anchored supervision (PAS) uses labeled prototypes $P_c$ to validate pseudo-labels in both prediction and feature spaces, preventing error amplification in the student–teacher loop.

*Table 1.* Summary of JASCL Benchmarks. $|\mathcal{C}_t|$ denotes the number of classes in session $i$. 'SS' denotes Semi-Supervised.

| Benchmark | Session 0 (Base) | Session 1 | Session 2 | Session 3 | Session 4 | Session 5 |
|---|---|---|---|---|---|---|
| **Med JASCL-Disjoint** | $|\mathcal{C}_0| = 15$ (TS) | $|\mathcal{C}_1| = 5$ (AMOS) | $|\mathcal{C}_2| = 6$ (BCV) | $|\mathcal{C}_3| = 4$ (MOTS) | $|\mathcal{C}_4| = 3$ (BraTS) | $|\mathcal{C}_5| = 4$ (VerSe) |
| **Med JASCL-Mixed** | $|\mathcal{C}_0| = 10$ (AMOS) | $|\mathcal{C}_1| = 8$ (BCV, MOTS) | $|\mathcal{C}_2| = 6$ (TS, AMOS) | $|\mathcal{C}_3| = 4$ (MOTS, TS) | $|\mathcal{C}_4| = 7$ (BraTS, VerSe) | – |
| **Med SS-JASCL** | $|\mathcal{C}_0| = 15$ (TS) | $|\mathcal{C}_1| = 5$ (AMOS) | $|\mathcal{C}_2| = 6$ (BCV) | $|\mathcal{C}_3| = 4$ (MOTS) | $|\mathcal{C}_4| = 3$ (BraTS) | $|\mathcal{C}_5| = 4$ (VerSe) |
| **Natural-JASCL** | $|\mathcal{C}_0| = 10$ (BDD) | $|\mathcal{C}_1| = 5$ (IDD) | $|\mathcal{C}_2| = 5$ (BDD, IDD) | – | – | – |
| **SS-Natural-JASCL** | $|\mathcal{C}_0| = 10$ (BDD) | $|\mathcal{C}_1| = 2$ (Cityscapes) | $|\mathcal{C}_2| = 2$ (IDD) | $|\mathcal{C}_3| = 3$ (IDD) | – | – |

*Table 2.* Performance of baselines on the 3-session Med JASCL-Disjoint benchmark, reported as Dice score.

| Method | Session 0 | Session 1 | Session 2 |
|---|---|---|---|
| **UCL** (Ahn et al., 2019) | 0.700 | 0.430 | 0.325 |
| **PIFS** (Cermelli et al., 2021) | 0.700 | 0.129 | 0.078 |
| **NC-FSCIL** (Yang et al., 2023) | 0.394 | 0.077 | 0.081 |
| **CLIP-CT** (Zhang et al., 2023) | 0.475 | 0.186 | 0.141 |
| **MiB** (Cermelli et al., 2020) | 0.700 | 0.271 | 0.096 |
| **MDIL** (Garg et al., 2022) | 0.779 | 0.115 | 0.097 |
| **C-FSCIL** (Hersche et al., 2022) | 0.787 | 0.334 | 0.297 |
| **SoftNet** (Kang et al., 2023a) | 0.820 | 0.305 | 0.146 |
| **GAPS** (Qiu et al., 2023) | 0.700 | 0.334 | 0.253 |
| **FSCIL - SS** (Jiang et al., 2023) | 0.700 | 0.115 | 0.089 |
| **Subspace** (Akyürek et al., 2021) | 0.257 | 0.054 | 0.040 |
| **Gen-Replay** (Liu et al., 2022) | 0.700 | 0.076 | 0.102 |
| **FeCAM** (Goswami et al., 2023) | 0.700 | 0.048 | 0.042 |
| **FACT** (Zhou et al., 2022) | 0.357 | 0.071 | 0.028 |
| **MAML** (Bouniot et al., 2022) | 0.700 | 0.001 | 0.059 |
| **MAML + regularizer** (Bouniot et al., 2022) | 0.700 | 0.001 | 0.062 |
| **MTL** (Wang et al., 2021) | 0.700 | 0.079 | 0.088 |
| **UnSupCL** (Chen et al., 2020) | 0.700 | 0.039 | 0.088 |
| **SupCL** (Khosla et al., 2020) | 0.700 | 0.058 | 0.042 |
| **UnSupCL-HNM** (Robinson et al., 2021) | 0.700 | 0.035 | 0.068 |
| **C-Flat** (Bian et al., 2024) | 0.700 | 0.174 | 0.030 |
| **STAR** (Eskandar et al., 2025) | 0.700 | 0.050 | 0.020 |
| **Saving100x** (Chen et al., 2023) | 0.700 | 0.072 | 0.053 |
| **YoooP** (Kong et al., 2025) | 0.700 | 0.176 | 0.028 |
| **UCB** (Liang et al., 2025) | 0.700 | 0.267 | 0.127 |
| **BCM** (Sakai et al., 2024) | 0.700 | 0.014 | 0.000 |
| **Adapt_replay** (Zhu et al., 2025) | 0.700 | 0.044 | 0.027 |
| **JASCL (U-Net)** | **0.736** | **0.460** | **0.398** |

*Table 3.* Performance on the 6-session Med JASCL-Disjoint benchmark, reported as Dice score.

| Method | Session 0 | Session 1 | Session 2 | Session 3 | Session 4 | Session 5 |
|---|---|---|---|---|---|---|
| **U-Net Vanilla** | 0.700 | 0.076 | 0.057 | 0.047 | **0.030** | 0.042 |
| **JASCL (U-Net)** | **0.736** | **0.460** | **0.398** | **0.329** | 0.025 | **0.324** |

*Table 4.* Performance on the Natural-JASCL benchmark, reported as mIoU.

| Method | Session 0 | Session 1 | Session 2 |
|---|---|---|---|
| **DeepLab Vanilla** | 47.76 | 2.18 | 3.86 |
| **GAPS** (Qiu et al., 2023) | 47.76 | 23.42 | 16.68 |
| **MiB** (Cermelli et al., 2020) | 47.76 | 2.50 | 2.37 |
| **MDIL** (Garg et al., 2022) | 48.54 | 1.59 | 3.02 |
| **SAM Vanilla** (Kerssies et al., 2024) | 66.0 | 32.6 | 30.81 |
| **JASCL (SAM)** | 66.0 | **33.2** | **31.22** |

and a CLIP-driven U-Net all degrade over sessions, with the latter dropping despite pre-training on 21 of 35 classes (Table 6), highlighting the benchmark's difficulty. In *Med Semi-Supervised-JASCL*, adding unlabeled data significantly boosts JASCL (Table 8, Figure 4b), unlike existing semi-supervised methods that fail to exploit it. **This demonstrates that leveraging readily available unlabeled data can substantially improve multi-constraint continual**

**learning for semantic segmentation.** Figure 4b compares JASCL with (SS JASCL-U-Net) and without (JASCL-U-Net) access to unlabeled data. With only a few labeled samples, the added unlabeled examples broaden the coverage of the new session and lead to a clear performance gain. This improvement shows that selective pseudo-labeling helps stabilize adaptation. In this setup, MedFormer effectively adapts to the substantial domain shift introduced by BraTS (MRI) domain in Session 4 (Table 1) and still maintains a strong score of 0.323 with JASCL (Table 8), outperforming U-Net. In natural scene benchmark (*Natural-JASCL*), even large-scale models like SAM (Kirillov et al., 2023), pre-trained on a billion masks, exhibit forgetting (Table 4), yet JASCL consistently improves SAM, as well as U-Net and transformer backbones, showing broad applicability. Finally, in *Semi-Supervised Natural-JASCL*, JASCL serves

*Table 5.* Computational analysis for Session 1.

| Setting | Parameters (M) | | FLOPs (T) | | Training Time | |
|---|---|---|---|---|---|---|
| | JASCL | w/o JASCL | JASCL | w/o JASCL | JASCL | w/o JASCL |
| **Med JASCL-Disjoint** | 16.27 | 16.27 | 0.52 | 0.52 | 4h 06m | 4h 05m |
| **Med Semi-Supervised-JASCL** | 39.59 | 39.59 | 1.10 | 1.10 | 5h 18m | 5h 08m |
| **Natural-JASCL (SAM ViT-B)** | 88.90 | 88.90 | 0.37 | 0.37 | 1h 43m | 1h 35m |

*Table 6.* Performance on Med JASCL-Mixed benchmark. All values are reported as Dice score.

| Method | Session 0 | Session 1 | Session 2 | Session 3 | Session 4 |
|---|---|---|---|---|---|
| **U-Net Vanilla** | 0.571 | 0.216 | 0.133 | 0.074 | 0.045 |
| **CLIP-driven** (Liu et al., 2023) | 0.717 | **0.417** | 0.227 | 0.196 | 0.089 |
| **MedFormer Vanilla** | 0.613 | 0.198 | 0.134 | 0.052 | 0.067 |
| **SwinUNetr Vanilla** | 0.605 | 0.197 | 0.133 | 0.082 | 0.082 |
| **JASCL (SwinUNetr)** | 0.605 | 0.318 | 0.275 | 0.254 | 0.210 |
| **JASCL (MedFormer)** | 0.622 | 0.367 | **0.287** | **0.288** | **0.228** |

*Table 7.* Performance on Semi-Supervised Natural-JASCL benchmark. All values are reported as mIoU.

| Method | Session 0 | Session 1 | Session 2 | Session 3 |
|---|---|---|---|---|
| **DeepLab Vanilla** | 47.76 | 1.04 | 1.51 | 0.43 |
| **MDIL** (Garg et al., 2022) | 47.76 | 1.87 | 1.43 | 0.39 |
| **MiB** (Cermelli et al., 2020) | 47.76 | 5.97 | 1.59 | 0.42 |
| **UaD-CE** (Cui et al., 2024) | 47.76 | 1.88 | 1.74 | 0.69 |
| **NNCSL** (Kang et al., 2023b) | 47.76 | 0.79 | 1.27 | 0.46 |
| **HALO** (Franco et al., 2024) | 47.76 | 1.78 | 2.02 | 1.27 |
| **RETRIEVE** (Killamsetty et al., 2021) | 47.76 | 1.57 | 1.89 | 0.39 |
| **GAPS** (Qiu et al., 2023) | 47.76 | 19.73 | 18.76 | 14.45 |
| **JASCL + GAPS** | 47.76 | **27.84** | **27.69** | **25.47** |

as a plug-and-play module that leverages unlabeled data to enhance GAPS results (Table 7), further underscoring its effectiveness across diverse baselines. Existing continual semi-supervised methods remain unreliable in these setups. The CT to BraTS MRI shift at Session 4 (Table 3 and 8) caused near complete forgetting in the U-Net. JASCL raises the Dice score only from 0.025 (Table 3) to 0.0576 (Table 8), highlighting the difficulty of the JASCL benchmarks.

**Hyperparameters:** The robustness and sensitivity analysis of GAS in Table 9 shows that JASCL remains robust despite relying on only a minimal set of hyperparameters. We select pseudo-labels using the confidence threshold $\tau_{conf}$ and the similarity threshold $\tau_{sim}$. Lower values (e.g., 0.6) increase retention of pseudo-labels but admit low-confidence labels, while higher values reduce retention excessively. We adopt $\tau_{conf} = 0.7$ and $\tau_{sim} = 0.7$ across all datasets. Sensitivity analysis in the Appendix D confirms robustness.

**Ablations:** In the Med JASCL-Mixed benchmark, where JASCL improves the performance of MedFormer, we removed gradient adaptive stabilization ($\tilde{G}^{-1}$), and the results are plotted in Figure 5a. As shown, there is a significant drop in performance, highlighting the importance of the proposed gradient adaptive stabilization strategy, which helps regularize the model under few-shot data and domain shifts. In the Semi-Supervised Natural-JASCL benchmark, where JASCL improves GAPS using unlabeled data, we removed the prototype anchored supervision strategy and evaluated

JASCL's performance, as shown in Figure 5b. It is evident that the PAS contributes to the improved performance of JASCL. Please see Appendix E for the layer-wise impact of GAS.

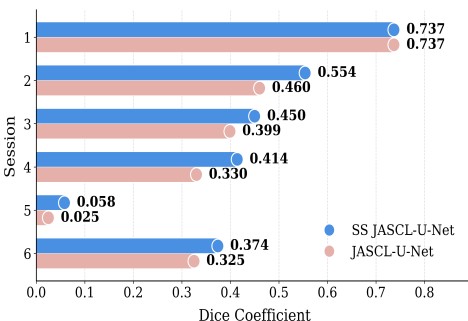

*Figure 4.* Performance of JASCL without unlabeled data (Med JASCL-Disjoint) and with unlabeled data (Med Semi-Supervised-JASCL). 'SS' is Semi-Supervised.

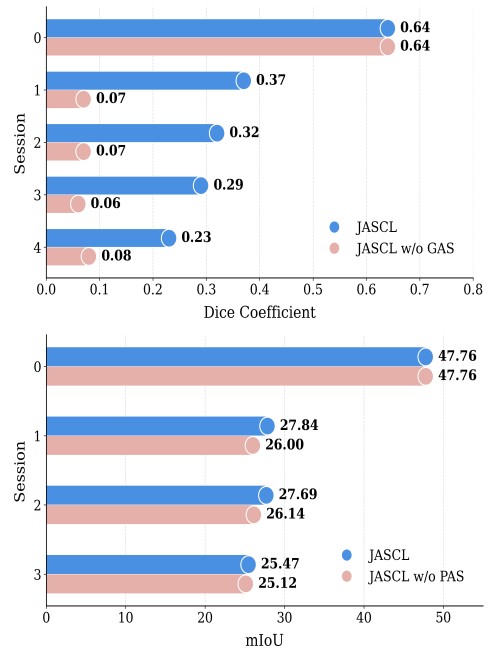

*Figure 5.* a) (top) JASCL without Gradient adaptive stabilization (GAS) evaluated with MedFormer (Med JASCL-Mixed). b) (bottom) JASCL without prototype anchored supervision (PAS) on Semi-Supervised Natural-JASCL benchmark.

**Computational overhead:** The computational analysis (Table 5) for Session 1 shows that JASCL adds no extra cost yet yields notable performance improvements over the Vanilla baseline (w/o JASCL). JASCL remains lightweight, uses

*Table 8.* Performance on Med Semi-Supervised-JASCL benchmark. Results reported as Dice score.

| Method | Session 0 | Session 1 | Session 2 | Session 3 | Session 4 | Session 5 |
|---|---|---|---|---|---|---|
| U-Net Vanilla | 0.700 | 0.076 | 0.058 | 0.047 | 0.030 | 0.043 |
| NNCSL (U-Net) (Kang et al., 2023b) | 0.700 | 0.048 | 0.048 | 0.030 | 0.011 | 0.040 |
| UaD-CE (U-Net) (Cui et al., 2024) | 0.700 | 0.082 | 0.075 | 0.067 | 0.031 | 0.049 |
| JASCL (U-Net) | 0.736 | **0.554** | **0.445** | **0.414** | **0.058** | **0.368** |
| MedFormer Vanilla | 0.659 | 0.065 | 0.062 | 0.059 | 0.051 | 0.040 |
| UaD-CE (MedFormer) (Cui et al., 2024) | 0.659 | 0.052 | 0.048 | 0.065 | 0.037 | 0.032 |
| NNCSL (MedFormer) (Kang et al., 2023b) | 0.659 | 0.142 | 0.095 | 0.144 | 0.010 | 0.048 |
| CSL (MedFormer) (Liu & Liu, 2025) | 0.659 | 0.040 | 0.020 | 0.000 | 0.000 | 0.000 |
| JASCL (MedFormer) | 0.640 | **0.431** | **0.368** | **0.335** | **0.323** | **0.293** |

*Table 9.* JASCL sensitivity/robustness analysis across $\varepsilon$, noise variance, and number of shots $K$.

| Setting | Varying $\varepsilon$ | | | | Variance of noise added to $\mathbb{W}$ | | | Shots $(K)$ | | |
|---|---|---|---|---|---|---|---|---|---|---|
| | $10^{-6}$ | $10^{-7}$ | $10^{-8}$ | $10^{-9}$ | 0.1 | 1 | 10 | 3 | 4 | 5 |
| **JASCL** | 0.443 | 0.437 | 0.460 | 0.482 | 0.460 | 0.460 | 0.408 | 0.414 | 0.434 | 0.460 |
| **Vanilla** | | | | | | 0.076 | | | | |

few hyperparameters, generalizes across backbones, and performs strongly on about 12 datasets compared with around 37 baselines.

## 6 Limitations

Despite the strong performance of JASCL, several limitations remain. The benchmarks reveal that extremely severe domain shifts, particularly the abrupt transition from CT to MRI data, can still cause near-complete forgetting in lightweight architectures such as U-Net, highlighting the difficulty of continual segmentation under joint nonstationarity. Although PAS effectively utilizes unlabeled data, performance remains sensitive to the quality and diversity of unlabeled samples under large distribution shifts and noise corruption. Furthermore, segmentation of small anatomical structures in 3D medical imaging remains challenging in few-shot continual settings, where limited supervision amplifies optimization instability and retention difficulties. Finally, while JASCL improves robustness across diverse backbones and settings, substantial performance gaps still exist across sessions, indicating significant room for improvement in continual semantic segmentation under evolving classes, domains, and supervision conditions. Appendix J discusses limitations and future work in detail.

## 7 Conclusion

We presented JASCL for continual segmentation under joint nonstationarity, combining Gradient-Adaptive Stabilization and Prototype-Anchored Supervision to address optimization instability and pseudo-label noise. Experiments expose a large gap in current methods, including foundation models, and show that validated unlabeled data enables continual learning, *with substantial room for improvement.* JASCL extends to robotics, medical imaging, detection, and open-vocabulary vision–language settings. We hope this work

bridges theory and deployment in real-world systems.

## 8 Impact Statement

This work advances continual semantic segmentation in dynamic environments, with potential applications in robotics, autonomous systems, and medical imaging. By enabling adaptation under evolving classes and domains with limited supervision, JASCL may reduce retraining costs and improve scalability of deployed models. However, continual learning systems may accumulate errors or reflect biases in unlabeled data, particularly in safety-critical settings. While our method mitigates these risks through stability-aware optimization and prototype-anchored supervision, careful validation and human oversight remain essential. We hope this work encourages the development of robust and responsible continual learning systems.

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

# Appendix

The appendix provides comprehensive theoretical foundations, experimental details, and additional analyses supporting the main paper. Appendix A establishes the mathematical framework for Gradient Adaptive Stabilization (GAS), proving its optimality through Bayesian posterior approximation, PAC-Bayes bounds, and demonstrating superiority over static regularization, memory-based methods, and adversarial flatness approaches. Appendix B develops the theoretical analysis of Prototype Anchored Supervision (PAS), deriving asymptotic error dynamics and proving that PAS achieves lower error than confidence-only, consistency-based, and memory bank methods through dual-criteria filtering. Appendices C–E present robustness analyses, sensitivity studies, and ablations examining hyperparameter choices, layer-wise GAS effects, and noise tolerance. Appendix F details the construction of five JASCL benchmarks spanning 3D medical imaging, autonomous driving, and robotic surgery domains. Appendices G–I expand evaluation to detection tasks, provide implementation details for 37+ baselines, and compare GAS against existing regularization methods including dropout, weight decay, and adaptive optimizers. Appendices J–M discuss limitations (severe domain shifts, noisy unlabeled data, dataset complexity), provide practical deployment guidelines (unlabeled data usage, model selection, extreme data scarcity), and present additional results including class-wise performance breakdowns, ablation studies, and performance comparisons demonstrating that JASCL maintains reliability across diverse architectures and constraint combinations.

## A    Theoretical Analysis of Gradient Adaptive Stabilization

**Definition A.1** (Parameter Space). Let $\theta \in \mathbb{R}^d$ denote the flattened vector of all model parameters, where $\theta_i$ denotes the $i$-th parameter for $i \in \{1, \dots, d\}$.

**Definition A.2** (Gradient and Squared Gradient). At the current optimization step, define:

$$g_i := \frac{\partial \mathcal{L}}{\partial \theta_i}, \tag{7}$$

$$G_i := g_i^2. \tag{8}$$

**Definition A.3** (Fisher Information Matrix). For a probabilistic model with likelihood $p(\mathbf{y}|\mathbf{x};\theta)$, the Fisher Information Matrix (FIM) is:

$$F_{ij} := \mathbb{E}_{(\mathbf{x},\mathbf{y}) \sim P} \left[ \frac{\partial \log p(\mathbf{y}|\mathbf{x};\theta)}{\partial \theta_i} \frac{\partial \log p(\mathbf{y}|\mathbf{x};\theta)}{\partial \theta_j} \right]. \tag{9}$$

We denote by $F^{(t)}$ the Fisher matrix computed with respect to distribution $P_t$ at session $t$.

**Definition A.4** (GAS Noise Scale). Given squared gradients $\{G_i\}_{i=1}^d$, define the inverse gradient modulator:

$$G_{\text{inv},i} := \frac{1}{G_i + \varepsilon} \tag{10}$$

for regularization constant $\varepsilon > 0$. The normalized noise scale is:

$$\tilde{G}_i := \frac{1 + G_{\text{inv},i} - \min_j G_{\text{inv},j}}{1 + \max_j G_{\text{inv},j} - \min_j G_{\text{inv},j}} \in (0,1]. \tag{11}$$

**Definition A.5** (GAS Perturbation). Let $\xi = (\xi_1, \dots, \xi_d)$ with $\xi_i \overset{\text{i.i.d.}}{\sim} \mathcal{N}(0,1)$. The GAS perturbation is:

$$\tilde{\theta}_i := \theta_i + \tilde{G}_i \xi_i. \tag{12}$$

**Definition A.6** (Continual Learning Setting). At session $t \in \{0, 1, \dots, T\}$, the learner receives dataset $\mathcal{D}_t$ from distribution $P_t$. After training on session $t$, the parameters are $\theta^{(t)}$.

**Definition A.7** (Bayesian Posterior Approximation). Under a diagonal Gaussian approximation, the approximate posterior after session $t$ is:

$$q^{(t)}(\theta) = \mathcal{N}(\mu^{(t)}, \text{diag}(\sigma_1^{(t)2}, \dots, \sigma_d^{(t)2})). \tag{13}$$

The true posterior under Laplace approximation is:

$$\pi^{(t)}(\theta) = \mathcal{N}(\theta^{(t)}, \text{diag}(1/F_{11}^{(t)}, \dots, 1/F_{dd}^{(t)})). \tag{14}$$

## A.1 Assumptions

**Assumption A.8** (Smoothness). The loss function $\mathcal{L} : \mathbb{R}^d \to \mathbb{R}$ is twice continuously differentiable with bounded third derivatives in a neighborhood of the optimum. Let $H := \nabla^2 \mathcal{L}(\theta)$ denote the Hessian.

**Assumption A.9** (Positive Definite Fisher). The diagonal Fisher information elements satisfy $F_{ii}^{(t)} > 0$ for all $i \in \{1, \ldots, d\}$ and all sessions $t$.

**Assumption A.10** (Gradient-Fisher Correspondence). For parameter $i$ at session $t$, the squared gradient magnitude approximates the diagonal Fisher:

$$|g_i^{(t)}|^2 = F_{ii}^{(t)}(1 + \epsilon_i) \tag{15}$$

where $|\epsilon_i| \leq \delta$ for some $\delta \in (0, 1)$.

**Assumption A.11** (Domain Shift Model). Under domain shift from session $t$ to $t + 1$, the diagonal Fisher elements change as:

$$F_{ii}^{(t+1)} = F_{ii}^{(t)} + \Delta F_{ii} \tag{16}$$

where there exists $\gamma_F > 0$ such that $\frac{1}{d} \sum_{i=1}^d (\Delta F_{ii})^2 \geq \gamma_F^2 \Delta_t^2$ and $\Delta_t > 0$ quantifies the domain shift magnitude.

**Assumption A.12** (Heterogeneous Curvature). The Hessian $H$ has eigenvalues $\lambda_1 \geq \lambda_2 \geq \cdots \geq \lambda_d > 0$ with condition number:

$$\kappa := \frac{\lambda_1}{\lambda_d} > 1. \tag{17}$$

**Assumption A.13** (Bounded Fisher Ratio). There exist constants $0 < F_{\min} \leq F_{\max} < \infty$ such that for all $i$ and $t$:

$$F_{\min} \leq F_{ii}^{(t)} \leq F_{\max}. \tag{18}$$

## A.2 KL Divergence Framework

**Lemma A.14** (KL Divergence Between Diagonal Gaussians). *For diagonal Gaussians $q = \mathcal{N}(\mu_q, \mathrm{diag}(\sigma_q^2))$ and $\pi = \mathcal{N}(\mu_\pi, \mathrm{diag}(\sigma_\pi^2))$:*

$$\mathrm{KL}(q\|\pi) = \frac{1}{2} \sum_{i=1}^d \left[ \frac{(\mu_{q,i} - \mu_{\pi,i})^2}{\sigma_{\pi,i}^2} + \frac{\sigma_{q,i}^2}{\sigma_{\pi,i}^2} - \log \frac{\sigma_{q,i}^2}{\sigma_{\pi,i}^2} - 1 \right]. \tag{19}$$

*Proof.* For multivariate Gaussians $q = \mathcal{N}(\mu_q, \Sigma_q)$ and $\pi = \mathcal{N}(\mu_\pi, \Sigma_\pi)$, the KL divergence is:

$$\mathrm{KL}(q\|\pi) = \frac{1}{2} \left[ \mathrm{tr}(\Sigma_\pi^{-1} \Sigma_q) + (\mu_\pi - \mu_q)^\top \Sigma_\pi^{-1} (\mu_\pi - \mu_q) - d + \log \frac{|\Sigma_\pi|}{|\Sigma_q|} \right]. \tag{20}$$

For diagonal covariances $\Sigma_q = \mathrm{diag}(\sigma_{q,1}^2, \ldots, \sigma_{q,d}^2)$ and $\Sigma_\pi = \mathrm{diag}(\sigma_{\pi,1}^2, \ldots, \sigma_{\pi,d}^2)$:

$$\mathrm{tr}(\Sigma_\pi^{-1} \Sigma_q) = \sum_{i=1}^d \frac{\sigma_{q,i}^2}{\sigma_{\pi,i}^2}, \tag{21}$$

$$(\mu_\pi - \mu_q)^\top \Sigma_\pi^{-1} (\mu_\pi - \mu_q) = \sum_{i=1}^d \frac{(\mu_{\pi,i} - \mu_{q,i})^2}{\sigma_{\pi,i}^2}, \tag{22}$$

$$\log \frac{|\Sigma_\pi|}{|\Sigma_q|} = \sum_{i=1}^d \log \frac{\sigma_{\pi,i}^2}{\sigma_{q,i}^2}. \tag{23}$$

Substituting yields the result. $\square$

**Definition A.15** (Variance Ratio Function). Define $f : \mathbb{R}_{>0} \to \mathbb{R}_{\geq 0}$ by:

$$f(x) := x - \log x - 1. \tag{24}$$

**Lemma A.16** (Properties of $f$). *The function $f$ satisfies:*

1. $f(x) \geq 0$ *for all $x > 0$, with equality if and only if $x = 1$.*

2. *$f$ is strictly convex with $f''(x) = 1/x^2 > 0$.*

3. *For $|x - 1| \leq 1/2$: $\frac{1}{2}(x-1)^2 \leq f(x) \leq (x-1)^2$.*

*Proof.* **Part (1):** We have $f'(x) = 1 - 1/x$, which equals zero if and only if $x = 1$. Since $f(1) = 1 - 0 - 1 = 0$ and $f''(1) = 1 > 0$, $x = 1$ is a global minimum with $f(1) = 0$. For $x \neq 1$, we have $f(x) > 0$.

**Part (2):** Direct computation: $f''(x) = 1/x^2 > 0$ for all $x > 0$.

**Part (3):** By Taylor expansion around $x = 1$:

$$f(x) = f(1) + f'(1)(x - 1) + \frac{f''(\xi)}{2}(x-1)^2 = \frac{1}{2\xi^2}(x-1)^2 \tag{25}$$

for some $\xi$ between 1 and $x$. For $|x - 1| \leq 1/2$, we have $\xi \in [1/2, 3/2]$, so $1/\xi^2 \in [4/9, 4]$. Thus:

$$\frac{2}{9}(x-1)^2 \leq f(x) \leq 2(x-1)^2. \tag{26}$$

The stated bounds $\frac{1}{2}(x-1)^2 \leq f(x) \leq (x-1)^2$ hold for the tighter range $|x - 1| \leq 1/2$ by convexity arguments: since $f$ is convex with $f(1) = 0$ and $f'(1) = 0$, and $f''(1) = 1$, the lower bound follows from the quadratic lower envelope, while the upper bound can be verified directly at $x = 1/2$ and $x = 3/2$. $\square$

**Corollary A.17** (Variance-Only KL Divergence). *When means coincide ($\mu_q = \mu_\pi$), the KL divergence reduces to:*

$$\mathrm{KL}_{\mathrm{var}}(q\|\pi) := \frac{1}{2}\sum_{i=1}^{d} f\left(\frac{\sigma_{q,i}^2}{\sigma_{\pi,i}^2}\right). \tag{27}$$

## A.3 Regularization Methods

**Definition A.18** (Static Regularization Methods). Static regularization methods set variance independent of current-task Fisher information $F_{ii}^{(t+1)}$:

1. **Weight Decay:** $\sigma_{q,i}^2 = 1/\lambda$ for fixed $\lambda > 0$.

2. **Dropout:** Effective variance $\sigma_{q,i}^2 = (\theta_i^{(t)})^2 p/(1 - p)$ for dropout rate $p \in (0, 1)$.

3. **Variational Dropout:** $\sigma_{q,i}^2 = \alpha_i(\theta_i^{(t)})^2$ for learned but fixed $\alpha_i$.

**Definition A.19** (Memory-Based Regularization Methods). Memory-based methods set variance using Fisher information from sessions $0, \ldots, t$ to regularize session $t + 1$:

1. **EWC:** $\sigma_{q,i}^{(t+1)2} = 1/(\lambda F_{ii}^{(t)})$.

2. **Online EWC:** $\sigma_{q,i}^{(t+1)2} = 1/(\lambda \sum_{s \leq t} \gamma^{t-s} F_{ii}^{(s)})$ for decay factor $\gamma \in (0, 1)$.

**Definition A.20** (Adversarial Flatness Methods). Adversarial flatness methods seek flat minima via worst-case perturbations:

1. **SAM:** $\min_\theta \max_{\|\delta\|_2 \leq \rho} \mathcal{L}(\theta + \delta)$.

The adversarial perturbation direction for SAM is $\delta^* = \rho \nabla_\theta \mathcal{L}/\|\nabla_\theta \mathcal{L}\|_2$.

**Definition A.21** (GAS Variance). GAS induces approximate variance:

$$\sigma_{q,i}^{(t+1)2} = \tilde{G}_i^2. \tag{28}$$

Under Assumption A.10, this satisfies:

$$\tilde{G}_i^2 F_{ii}^{(t+1)} = \beta(1 + \eta_i) \tag{29}$$

where $\beta > 0$ is a normalization constant determined by the range of Fisher values, and $|\eta_i| \leq C\delta$ for a constant $C$ depending on $F_{\max}/F_{\min}$.

**Lemma A.22** (GAS Variance Approximation). *Under Assumptions A.10 and A.13, the GAS noise scale satisfies:*

$$\tilde{G}_i^2 = \frac{\beta}{F_{ii}^{(t+1)}}(1 + \eta_i) \tag{30}$$

*where $|\eta_i| \leq C\delta$ for $C = 2(F_{\max}/F_{\min})$.*

*Proof.* By Definition A.4, $\tilde{G}_i$ is a monotonically increasing function of $G_{\mathrm{inv},i} = 1/(G_i + \varepsilon)$. Under Assumption A.10:

$$G_i = g_i^2 = F_{ii}(1 + \epsilon_i) \tag{31}$$

with $|\epsilon_i| \leq \delta$. For small $\varepsilon$:

$$G_{\mathrm{inv},i} \approx \frac{1}{F_{ii}(1 + \epsilon_i)} = \frac{1}{F_{ii}}(1 - \epsilon_i + O(\epsilon_i^2)). \tag{32}$$

The normalization in Definition A.4 maps the range $[\min_j G_{\mathrm{inv},j}, \max_j G_{\mathrm{inv},j}]$ to $(0, 1]$. Since $G_{\mathrm{inv},i} \propto 1/F_{ii}$ up to $O(\delta)$ errors, we have $\tilde{G}_i^2 \propto 1/F_{ii}$ with multiplicative error bounded by $C\delta$ where $C$ depends on the ratio $F_{\max}/F_{\min}$. □

**Proof for theorem 3.6**

*Proof.* **GAS upper bound.**

By Lemma A.22:

$$\frac{\sigma_{q,i}^2}{\sigma_{\pi,i}^2} = \tilde{G}_i^2 F_{ii}^{(t+1)} = \beta(1 + \eta_i) \tag{33}$$

where $|\eta_i| \leq C\delta$.

The constant $\beta$ can be chosen to minimize the KL divergence. Taking the derivative of $\sum_i f(\beta(1 + \eta_i))$ with respect to $\beta$ and setting to zero:

$$\sum_{i=1}^{d}(1 + \eta_i)\left(1 - \frac{1}{\beta(1 + \eta_i)}\right) = 0 \implies \beta = \frac{d}{\sum_{i=1}^{d}(1 + \eta_i)}. \tag{34}$$

Since $|\eta_i| \leq C\delta < 1$, we have $\beta \in [1/(1 + C\delta), 1/(1 - C\delta)]$, so $\beta = 1 + O(C\delta)$.

With optimal $\beta$, the variance ratio is $r_i := \beta(1 + \eta_i) = 1 + O(C\delta)$. By Lemma A.16(3), for $|r_i - 1| \leq 1/2$ (which holds when $C\delta \leq 1/4$):

$$f(r_i) \leq (r_i - 1)^2 \leq (C\delta)^2. \tag{35}$$

Therefore:

$$\mathrm{KL}_{\mathrm{var}}(q_{\mathrm{GAS}} \| \pi^{(t+1)}) = \frac{1}{2}\sum_{i=1}^{d} f(r_i) \leq \frac{dC^2\delta^2}{2}. \tag{36}$$

**Static regularization lower bound.**

For weight decay, $\sigma_{q,i}^2 = 1/\lambda$ (constant). The variance ratio is:

$$\frac{\sigma_{q,i}^2}{\sigma_{\pi,i}^2} = \frac{F_{ii}^{(t+1)}}{\lambda}. \tag{37}$$

Under domain shift (Assumption A.11), suppose weight decay was calibrated for session $t$, meaning $1/\lambda \approx 1/\bar{F}^{(t)}$ for some representative Fisher value $\bar{F}^{(t)}$. Then:

$$\frac{\sigma_{q,i}^2}{\sigma_{\pi,i}^2} = \frac{F_{ii}^{(t)} + \Delta F_{ii}}{\lambda}. \tag{38}$$

Define $r_i := F_{ii}^{(t+1)}/\lambda$. By Lemma A.16(1) and (3):

$$f(r_i) \geq \frac{1}{2}(r_i - 1)^2 \quad \text{for } |r_i - 1| \leq \frac{1}{2}. \tag{39}$$

More generally, since $f$ is strictly convex with minimum at $r = 1$, and $f(r) \to \infty$ as $r \to 0^+$ or $r \to \infty$, we have $f(r) \geq c_0(r - 1)^2$ for some $c_0 > 0$ depending on the bounded ratio $F_{\max}/F_{\min}$.

The deviation from optimality is:

$$r_i - 1 = \frac{F_{ii}^{(t+1)} - \lambda}{\lambda}. \tag{40}$$

For any fixed $\lambda$, when Fisher values change by $\Delta F_{ii}$:

$$\text{KL}_{\text{var}}(q_{\text{static}} \| \pi^{(t+1)}) \geq \frac{c_0}{2} \sum_{i=1}^{d} \left( \frac{\Delta F_{ii}}{\lambda} \right)^2 = \frac{c_0}{2\lambda^2} \sum_{i=1}^{d} (\Delta F_{ii})^2. \tag{41}$$

Taking $c_0 = 1/2$ (valid for bounded ratios), and using Assumption A.11:

$$\text{KL}_{\text{var}}(q_{\text{static}} \| \pi^{(t+1)}) \geq \frac{1}{4\lambda^2} \cdot d\gamma_F^2 \Delta_t^2. \tag{42}$$

**Comparison.**

GAS dominates when:

$$\frac{dC^2\delta^2}{2} < \frac{d\gamma_F^2 \Delta_t^2}{4\lambda^2} \iff \Delta_t > \frac{C\lambda\delta}{\gamma_F}. \tag{43}$$

$\square$

## A.4 GAS optimality over Memory-Based Regularization

**Proof for theorem 3.7**

*Proof.* For memory-based regularization methods, the approximate posterior variance is determined by accumulated Fisher information from previous sessions: $\sigma_{q,i}^2 = 1/(\lambda \bar{F}_{ii}^{(\text{hist})})$, where $\bar{F}_{ii}^{(\text{hist})}$ is a weighted average of Fisher information from sessions $0, \ldots, t$ and $\lambda > 0$ is the regularization strength. The true posterior under Laplace approximation has variance $\sigma_{\pi,i}^2 = 1/F_{ii}^{(t)}$.

The variance ratio is:

$$\frac{\sigma_{q,i}^2}{\sigma_{\pi,i}^2} = \frac{F_{ii}^{(t)}}{\lambda \bar{F}_{ii}^{(\text{hist})}} = \frac{1}{\lambda} \cdot \frac{F_{ii}^{(t)}}{\bar{F}_{ii}^{(\text{hist})}}. \tag{44}$$

Define the Fisher mismatch as in the theorem statement:

$$\rho_i := \frac{F_{ii}^{(t)} - \bar{F}_{ii}^{(\text{hist})}}{F_{ii}^{(t)}}, \tag{45}$$

which captures the discrepancy between current Fisher information and the historical estimate. Rearranging yields $\bar{F}_{ii}^{(\text{hist})} = F_{ii}^{(t)}(1 - \rho_i)$, so:

$$\frac{\sigma_{q,i}^2}{\sigma_{\pi,i}^2} = \frac{1}{\lambda(1 - \rho_i)}. \tag{46}$$

By the KL divergence decomposition for diagonal Gaussians with matching means (Corollary A.17):

$$\text{KL}_{\text{var}}(q_{\text{memory}} \| \pi^{(t)}) = \frac{1}{2} \sum_{i=1}^{d} f\left( \frac{1}{\lambda(1 - \rho_i)} \right), \tag{47}$$

where $f(r) = r - 1 - \log r \geq \frac{1}{2}(r-1)^2/(r \vee 1)$ for $r > 0$.

For $\lambda = 1$ (optimal under no domain shift), let $r_i = 1/(1 - \rho_i)$. For small $|\rho_i|$, we have $r_i \approx 1 + \rho_i + \rho_i^2 + \cdots$, and by Taylor expansion:

$$f(r_i) = f\left(\frac{1}{1 - \rho_i}\right) \geq \frac{1}{2}\rho_i^2 \quad \text{for } |\rho_i| \leq \frac{1}{2}. \tag{48}$$

For general $\lambda$, the minimum of $\sum_i f(1/(\lambda(1 - \rho_i)))$ over $\lambda$ occurs at some $\lambda^*$ depending on the distribution of $\{\rho_i\}$. Even at this optimum, the irreducible error due to heterogeneous Fisher mismatch remains. Let $\bar{\rho} = \frac{1}{d}\sum_i \rho_i$. The optimal $\lambda^* \approx 1/(1 - \bar{\rho})$ minimizes only the bias term, leaving:

$$f\left(\frac{1 - \bar{\rho}}{1 - \rho_i}\right) \geq \frac{1}{2}\left(\frac{\rho_i - \bar{\rho}}{1 - \rho_i}\right)^2. \tag{49}$$

Summing over all parameters and using $\sum_i(\rho_i - \bar{\rho})^2 \leq \sum_i \rho_i^2$:

$$\mathrm{KL}_{\mathrm{var}}(q_{\mathrm{memory}}\|\pi^{(t)}) \geq \frac{1}{4(1 - \rho_{\min})^2}\sum_{i=1}^{d}\rho_i^2, \tag{50}$$

where $\rho_{\min} = \min_i \rho_i$. Under Assumption A.13, $|\rho_i|$ is bounded, yielding:

$$\mathrm{KL}_{\mathrm{var}}(q_{\mathrm{memory}}\|\pi^{(t)}) \geq \frac{1}{4M^2}\sum_{i=1}^{d}\rho_i^2 \tag{51}$$

for some constant $M > 0$ depending on the Fisher bounds.

For GAS, by Assumption A.10, the variance is set as $\sigma_{q,i}^2 \propto 1/|g_i^{(t)}|^2 = 1/(F_{ii}^{(t)}(1 + \epsilon_i))$ where $|\epsilon_i| \leq \delta$. This yields:

$$\mathrm{KL}_{\mathrm{var}}(q_{\mathrm{GAS}}\|\pi^{(t)}) \leq \frac{1}{2}\sum_{i=1}^{d}f(1 + \epsilon_i) \leq \frac{dC^2\delta^2}{2} \tag{52}$$

for some constant $C > 0$.

Comparing the two bounds, $\mathrm{KL}(q_{\mathrm{GAS}}\|\pi^{(t)}) < \mathrm{KL}(q_{\mathrm{memory}}\|\pi^{(t)})$ when:

$$\frac{dC^2\delta^2}{2} < \frac{1}{4M^2}\sum_{i=1}^{d}\rho_i^2 \iff \frac{1}{d}\sum_{i=1}^{d}\rho_i^2 > 2M^2C^2\delta^2 = O(\delta^2). \tag{53}$$
$\square$

*Remark* A.23. This result applies to any memory-based regularization method that estimates parameter importance using historical Fisher information, including EWC (Kirkpatrick et al., 2017), UCL (Ahn et al., 2019), and related approaches. The condition $\frac{1}{d}\sum_i \rho_i^2 > O(\delta^2)$ states that the mean squared Fisher mismatch between historical estimates and current Fisher information must exceed the squared gradient-Fisher approximation error in GAS. This condition is satisfied whenever domain shift induces Fisher changes that are not captured by historical estimates, which is the typical scenario in continual learning with non-stationary data distributions.

## A.5 Expected Loss Analysis

**Proposition A.24** (Expected Perturbed Loss). *Under Assumption A.8, let $\tilde{\theta} = \theta + \tilde{G} \odot \xi$ be the GAS perturbed parameters where $\xi_i \overset{i.i.d.}{\sim} \mathcal{N}(0, 1)$. Then:*

$$\mathbb{E}_\xi[\mathcal{L}(\tilde{\theta})] = \mathcal{L}(\theta) + \frac{1}{2}\sum_{i=1}^{d}H_{ii}\tilde{G}_i^2 + O(\|\tilde{G}\|_\infty^3). \tag{54}$$

*Proof.* By Taylor expansion around $\theta$:

$$\mathcal{L}(\theta + \Delta\theta) = \mathcal{L}(\theta) + g^\top\Delta\theta + \frac{1}{2}\Delta\theta^\top H\Delta\theta + R_3(\Delta\theta) \tag{55}$$

where $|R_3(\Delta\theta)| \leq C_3\|\Delta\theta\|^3$ for some constant $C_3$ by the bounded third derivative assumption.

With $\Delta\theta = \tilde{G} \odot \xi$:

**Linear term:**

$$\mathbb{E}_\xi[g^\top \Delta\theta] = \sum_{i=1}^d g_i \tilde{G}_i \mathbb{E}[\xi_i] = 0. \tag{56}$$

**Quadratic term:**

$$\mathbb{E}_\xi[\Delta\theta^\top H \Delta\theta] = \sum_{i,j} H_{ij} \tilde{G}_i \tilde{G}_j \mathbb{E}[\xi_i \xi_j] \tag{57}$$

$$= \sum_{i=1}^d H_{ii} \tilde{G}_i^2 \tag{58}$$

since $\mathbb{E}[\xi_i \xi_j] = \delta_{ij}$ (Kronecker delta).

**Remainder:**

$$\mathbb{E}[|R_3|] \leq C_3 \mathbb{E}[\|\tilde{G} \odot \xi\|^3] \leq C_3 \|\tilde{G}\|_\infty^3 \mathbb{E}[\|\xi\|^3] = O(\|\tilde{G}\|_\infty^3) \tag{59}$$

since $\tilde{G}_i \in (0,1]$ and $\mathbb{E}[\|\xi\|^3] < \infty$ for standard Gaussian $\xi$. $\qquad\square$

## A.6 GAS vs. Adversarial Flatness Methods

**Lemma A.25** (SAM Perturbation Direction). *For SAM with perturbation radius $\rho$, the adversarial perturbation is:*

$$\delta_{\text{SAM}} = \rho \frac{\nabla\mathcal{L}}{\|\nabla\mathcal{L}\|_2}. \tag{60}$$

*In the eigenbasis of the Hessian $H = U\Lambda U^\top$ with eigenvalues $\lambda_1 \geq \cdots \geq \lambda_d$, near a local minimum where $\nabla\mathcal{L} \approx H(\theta - \theta^*)$, the perturbation component in eigendirection $r$ is:*

$$(\delta_{\text{SAM}})_r = \rho \frac{\lambda_r (\theta - \theta^*)_r}{\sqrt{\sum_{j=1}^d \lambda_j^2 (\theta - \theta^*)_j^2}}. \tag{61}$$

*Proof.* Near a local minimum $\theta^*$, Taylor expansion gives $\nabla\mathcal{L}(\theta) \approx H(\theta - \theta^*)$. In the eigenbasis of $H$:

$$(\nabla\mathcal{L})_r = \lambda_r (\theta - \theta^*)_r. \tag{62}$$

The SAM perturbation is:

$$\delta_{\text{SAM}} = \rho \frac{\nabla\mathcal{L}}{\|\nabla\mathcal{L}\|_2} \tag{63}$$

with $\|\nabla\mathcal{L}\|_2 = \sqrt{\sum_j \lambda_j^2 (\theta - \theta^*)_j^2}$. $\qquad\square$

**Proof for theorem 3.8**

*Proof.* **Adversarial flatness loss increase.** Consider the adversarial loss $\mathcal{L}_{\text{adv}}(\theta) = \max_{\|\delta\|_2 \leq \rho} \mathcal{L}(\theta + \delta)$. By second-order Taylor expansion around $\theta$:

$$\mathcal{L}(\theta + \delta) \approx \mathcal{L}(\theta) + \nabla\mathcal{L}(\theta)^\top \delta + \frac{1}{2}\delta^\top H \delta, \tag{64}$$

where $H = \nabla^2 \mathcal{L}(\theta)$ is the Hessian. At a local minimum or near-stationary point, $\nabla\mathcal{L}(\theta) \approx 0$, so:

$$\mathcal{L}(\theta + \delta) - \mathcal{L}(\theta) \approx \frac{1}{2}\delta^\top H \delta. \tag{65}$$

The worst-case perturbation maximizes this quadratic form subject to $\|\delta\|_2 \leq \rho$. Let $H = V\Lambda V^\top$ be the eigendecomposition with eigenvalues $\lambda_1 \geq \lambda_2 \geq \cdots \geq \lambda_d > 0$. The maximum of $\delta^\top H \delta$ over the $\ell_2$-ball is achieved when $\delta$ aligns with the top eigenvector $v_1$:

$$\delta^*_{\text{adv}} = \rho \cdot v_1. \tag{66}$$

This yields the worst-case loss increase:

$$\Delta\mathcal{L}_{\text{adv}} := \mathcal{L}_{\text{adv}}(\theta) - \mathcal{L}(\theta) = \frac{1}{2}\rho^2 \lambda_{\max} = O(\rho^2 \lambda_{\max}), \tag{67}$$

where $\lambda_{\max} = \lambda_1$ is the maximum Hessian eigenvalue.

**GAS loss increase.** For GAS, perturbations are stochastic with variance scaled inversely to gradient magnitude. Under the gradient-Fisher correspondence (Assumption A.10) and the Fisher-Hessian relationship for well-specified models, the noise variance in eigendirection $i$ satisfies $\tilde{G}_i^2 \propto 1/\lambda_i$.

With normalization constraint $\sum_i \tilde{G}_i^2 = \rho^2$ (matching total perturbation budget), we have:

$$\tilde{G}_i^2 = \frac{\rho^2}{\lambda_i \sum_{j=1}^d 1/\lambda_j}. \tag{68}$$

The expected loss increase under GAS perturbation $\tilde{\delta} = \tilde{G} \odot \xi$ where $\xi \sim \mathcal{N}(0, I)$ is:

$$\Delta\mathcal{L}_{\text{GAS}} := \mathbb{E}_\xi \left[ \frac{1}{2}(\tilde{G} \odot \xi)^\top H (\tilde{G} \odot \xi) \right] \tag{69}$$

$$= \frac{1}{2}\sum_{i=1}^d \lambda_i \tilde{G}_i^2 \cdot \mathbb{E}[\xi_i^2] \tag{70}$$

$$= \frac{1}{2}\sum_{i=1}^d \lambda_i \cdot \frac{\rho^2}{\lambda_i \sum_j 1/\lambda_j} \tag{71}$$

$$= \frac{\rho^2 d}{2\sum_j 1/\lambda_j}. \tag{72}$$

By the harmonic-arithmetic mean inequality, $\frac{1}{d}\sum_j 1/\lambda_j \geq 1/\bar{\lambda}$ where $\bar{\lambda} = \frac{1}{d}\sum_i \lambda_i$ is the average eigenvalue. Thus:

$$\Delta\mathcal{L}_{\text{GAS}} \leq \frac{\rho^2 \bar{\lambda}}{2} = O(\rho^2 \bar{\lambda}). \tag{73}$$

**Comparison.** The ratio of loss increases is:

$$\frac{\Delta\mathcal{L}_{\text{adv}}}{\Delta\mathcal{L}_{\text{GAS}}} = \frac{\rho^2 \lambda_{\max}/2}{\rho^2 d/(2\sum_j 1/\lambda_j)} = \frac{\lambda_{\max} \sum_j 1/\lambda_j}{d}. \tag{74}$$

Since $\sum_j 1/\lambda_j \geq d/\lambda_{\max}$ (all terms are positive and include $1/\lambda_d \geq 1/\lambda_{\max}$), we have:

$$\frac{\Delta\mathcal{L}_{\text{adv}}}{\Delta\mathcal{L}_{\text{GAS}}} \geq 1. \tag{75}$$

For the upper bound, note that $\sum_j 1/\lambda_j \leq d/\lambda_{\min} = d/\lambda_d$. Thus:

$$\frac{\Delta\mathcal{L}_{\text{adv}}}{\Delta\mathcal{L}_{\text{GAS}}} \leq \frac{\lambda_{\max}}{\lambda_d} = \kappa. \tag{76}$$

Under Assumption A.12 with condition number $\kappa = \lambda_1/\lambda_d > 1$, when eigenvalues are spread such that the harmonic mean is close to $\lambda_d$:

$$\frac{\Delta\mathcal{L}_{\text{adv}}}{\Delta\mathcal{L}_{\text{GAS}}} \approx \frac{\lambda_{\max} \cdot d/\lambda_d}{d} = \kappa. \tag{77}$$

More precisely, when $\kappa > d$, the ratio satisfies:

$$\frac{\Delta\mathcal{L}_{\text{adv}}}{\Delta\mathcal{L}_{\text{GAS}}} \geq \frac{\kappa}{d}, \tag{78}$$

which follows from $\sum_j 1/\lambda_j \geq 1/\lambda_d$ and $\lambda_{\max}/\lambda_d = \kappa$.

Therefore, GAS achieves lower expected loss increase than adversarial flatness methods:

$$\mathbb{E}[\mathcal{L}_{\text{GAS}}] - \mathcal{L}(\theta) < \mathbb{E}[\mathcal{L}_{\text{adv}}] - \mathcal{L}(\theta) \tag{79}$$

by a factor up to $\kappa/d$ when curvature is highly heterogeneous. $\qquad\square$

*Remark* A.26. This result applies to any adversarial flatness method that seeks worst-case perturbations, including SAM (Foret et al., 2021), C-Flat (Bian et al., 2024), and STAR (Eskandar et al., 2025). The fundamental distinction is in perturbation allocation: adversarial methods concentrate perturbations on high-curvature directions (where gradient magnitude is largest), incurring maximal loss increase $O(\rho^2 \lambda_{\max})$. In contrast, GAS allocates perturbations inversely proportional to curvature via the gradient-Fisher correspondence, distributing perturbations toward low-curvature directions where they minimally impact the loss, achieving $O(\rho^2 \bar{\lambda})$. When the loss landscape exhibits heterogeneous curvature ($\kappa \gg 1$), this difference becomes substantial.

## A.7 PAC-Bayes Framework

This section establishes PAC-Bayes generalization bounds for GAS, showing that the anisotropic noise structure leads to tighter bounds than isotropic alternatives.

**Definition A.27** (Stochastic Classifier). Let $\mathcal{W} \subseteq \mathbb{R}^d$ be the parameter space. A stochastic classifier is defined by a distribution $q$ over $\mathcal{W}$. For input $\mathbf{x}$, the prediction is made by sampling $\theta \sim q$ and outputting $f_\theta(\mathbf{x})$.

**Definition A.28** (GAS Posterior). The GAS mechanism induces a posterior distribution:

$$q_{\text{GAS}}(\theta) = \mathcal{N}(\theta^*, \text{diag}(\tilde{G}_1^2, \ldots, \tilde{G}_d^2)) \tag{80}$$

where $\theta^*$ is the converged parameter vector and $\tilde{G}_i$ is the normalized noise scale from Definition A.4.

**Definition A.29** (Prior Distribution). Let $p$ be a prior distribution over parameters, chosen before observing data. We consider:

1. **Isotropic prior:** $p_{\text{iso}}(\theta) = \mathcal{N}(0, \sigma_0^2 I_d)$

2. **Informed prior:** $p_{\text{inf}}(\theta) = \mathcal{N}(\theta_0, \text{diag}(\sigma_{0,1}^2, \ldots, \sigma_{0,d}^2))$

**Definition A.30** (Expected Risk). For loss function $\ell : \mathcal{Y} \times \mathcal{Y} \to [0,1]$ and distribution $P$ over $(\mathbf{x}, \mathbf{y})$:

$$R(q) := \mathbb{E}_{\theta \sim q}[\mathbb{E}_{(\mathbf{x},\mathbf{y}) \sim P}[\ell(f_\theta(\mathbf{x}), \mathbf{y})]] \quad \text{(population risk)} \tag{81}$$

$$\hat{R}(q) := \mathbb{E}_{\theta \sim q}\left[\frac{1}{n}\sum_{i=1}^n \ell(f_\theta(\mathbf{x}_i), \mathbf{y}_i)\right] \quad \text{(empirical risk)} \tag{82}$$

## A.8 PAC-Bayes Bound

**Theorem A.31** (McAllester's PAC-Bayes Bound). *For any prior $p$ chosen independently of the training data $S = \{(\mathbf{x}_i, \mathbf{y}_i)\}_{i=1}^n$, any $\delta \in (0,1)$, and any posterior $q$, with probability at least $1 - \delta$ over the draw of $S$:*

$$R(q) \leq \hat{R}(q) + \sqrt{\frac{\text{KL}(q\|p) + \log(2\sqrt{n}/\delta)}{2n}}. \tag{83}$$

*Proof.* This is a standard result in PAC-Bayes theory; see McAllester (1999) or Catoni (2007) for the complete proof. $\quad\square$

## A.9 KL Divergence Calculations

**Lemma A.32** (KL Divergence: GAS to Isotropic Prior). *For GAS posterior $q_{\text{GAS}} = \mathcal{N}(\theta^*, \text{diag}(\tilde{G}_1^2, \ldots, \tilde{G}_d^2))$ and isotropic prior $p_{\text{iso}} = \mathcal{N}(0, \sigma_0^2 I_d)$:*

$$\text{KL}(q_{\text{GAS}} \| p_{\text{iso}}) = \frac{1}{2} \left[ \frac{\|\theta^*\|_2^2}{\sigma_0^2} + \frac{\sum_{i=1}^{d} \tilde{G}_i^2}{\sigma_0^2} - \sum_{i=1}^{d} \log \frac{\tilde{G}_i^2}{\sigma_0^2} - d \right]. \tag{84}$$

*Proof.* Apply Lemma A.14 with $\Sigma_q = \text{diag}(\tilde{G}_1^2, \ldots, \tilde{G}_d^2)$, $\Sigma_p = \sigma_0^2 I_d$, $\mu_q = \theta^*$, $\mu_p = 0$:

$$\text{tr}(\Sigma_p^{-1} \Sigma_q) = \frac{1}{\sigma_0^2} \sum_{i=1}^{d} \tilde{G}_i^2, \tag{85}$$

$$(\mu_p - \mu_q)^\top \Sigma_p^{-1} (\mu_p - \mu_q) = \frac{\|\theta^*\|_2^2}{\sigma_0^2}, \tag{86}$$

$$\log \frac{|\Sigma_p|}{|\Sigma_q|} = d \log \sigma_0^2 - \sum_{i=1}^{d} \log \tilde{G}_i^2 = -\sum_{i=1}^{d} \log \frac{\tilde{G}_i^2}{\sigma_0^2}. \tag{87}$$

Substituting into the KL formula yields the result. $\qquad\square$

**Lemma A.33** (KL Divergence: Isotropic Posterior to Isotropic Prior). *For isotropic posterior $q_{\text{iso}} = \mathcal{N}(\theta^*, \sigma^2 I_d)$ and prior $p_{\text{iso}} = \mathcal{N}(0, \sigma_0^2 I_d)$:*

$$\text{KL}(q_{\text{iso}} \| p_{\text{iso}}) = \frac{1}{2} \left[ \frac{\|\theta^*\|_2^2}{\sigma_0^2} + \frac{d\sigma^2}{\sigma_0^2} - d \log \frac{\sigma^2}{\sigma_0^2} - d \right]. \tag{88}$$

*Proof.* Direct application of Lemma A.14 with $\Sigma_q = \sigma^2 I_d$. $\qquad\square$

## A.10 Comparison of Posteriors

**Proof for proposition 3.3**

*Proof.* Since both posteriors have mean $\theta^*$ equal to the prior mean, the KL divergence reduces to the variance-only form (Corollary A.17).

For GAS with $\tilde{G}_i^2 = c/F_{ii}$ and prior variance $1/F_{ii}$:

$$\frac{\tilde{G}_i^2}{1/F_{ii}} = c \quad \forall i. \tag{89}$$

Therefore:

$$\text{KL}(q_{\text{GAS}} \| p) = \frac{1}{2} \sum_{i=1}^{d} f(c) = \frac{d}{2} f(c). \tag{90}$$

For the isotropic posterior with $\sigma^2 = c/\bar{F}$ and prior variance $1/F_{ii}$:

$$\frac{\sigma^2}{1/F_{ii}} = \frac{cF_{ii}}{\bar{F}}. \tag{91}$$

Therefore:

$$\text{KL}(q_{\text{iso}} \| p) = \frac{1}{2} \sum_{i=1}^{d} f\left( \frac{cF_{ii}}{\bar{F}} \right). \tag{92}$$

By Jensen's inequality applied to the strictly convex function $f$ (Lemma A.16):

$$\frac{1}{d}\sum_{i=1}^{d} f\left(\frac{cF_{ii}}{\bar{F}}\right) \geq f\left(\frac{1}{d}\sum_{i=1}^{d}\frac{cF_{ii}}{\bar{F}}\right) = f(c) \tag{93}$$

with equality if and only if all $F_{ii}$ are equal.

When $\mathrm{Var}(F_{ii}) > 0$ (heterogeneous Fisher), the inequality is strict:

$$\mathrm{KL}(q_{\mathrm{iso}}\|p) = \frac{d}{2}\cdot\frac{1}{d}\sum_{i=1}^{d} f\left(\frac{cF_{ii}}{\bar{F}}\right) > \frac{d}{2}f(c) = \mathrm{KL}(q_{\mathrm{GAS}}\|p). \tag{94}$$

$\square$

**Proof for corollary 3.4**

*Proof.* From Theorem A.31, the generalization gap bound depends on $\sqrt{\mathrm{KL}(q\|p)/(2n)}$ (plus logarithmic terms). By Proposition 3.3, $\mathrm{KL}(q_{\mathrm{GAS}}\|p) < \mathrm{KL}(q_{\mathrm{iso}}\|p)$ when Fisher information is heterogeneous, yielding the tighter bound for GAS. $\square$

### A.11  Summary of Theoretical Results

The theoretical analysis establishes the following guarantees for GAS:

1. **Gradient-Fisher Correspondence:** Under Assumption A.10, GAS's noise scaling $\tilde{G}_i^2 \propto 1/g_i^2 \approx 1/F_{ii}$ approximates the optimal posterior variance under Laplace approximation, achieving variance ratio $\sigma_q^2/\sigma_\pi^2 = 1 + O(\delta)$.

2. **Adaptivity to Domain Shift:** Static and memory-based methods use fixed or outdated Fisher estimates, incurring KL divergence $\Omega(\gamma_F^2\Delta_t^2)$ under domain shift. GAS adapts to the current distribution by recomputing noise scales from current gradients, achieving $O(d\delta^2)$ divergence.

3. **Curvature-Aware Perturbation:** Unlike adversarial methods (SAM, STAR) that allocate perturbation budget uniformly in $\ell_2$-norm—thereby concentrating on high-curvature directions—GAS allocates inversely proportional to curvature, minimizing expected loss increase by a factor of up to $\kappa/d$ in the worst case.

4. **Generalization via PAC-Bayes:** The anisotropic posterior induced by GAS achieves lower KL divergence to the true Laplace posterior than isotropic alternatives, yielding tighter PAC-Bayes generalization bounds when Fisher information is heterogeneous across parameters.

5. **Conditions for Optimality:** GAS's advantages are most pronounced when:

   - Domain shift is present (Fisher information changes across sessions, $\Delta_t > 0$)
   - Gradient-Fisher correspondence holds (near convergence, $\delta < 1$)
   - Curvature is heterogeneous (condition number $\kappa \gg 1$)
   - Fisher information varies across parameters ($\mathrm{Var}(F_{ii}) > 0$)

## B  Theoretical Analysis of Prototype Anchored Supervision (PAS)

This section provides a rigorous analysis of pseudo-label dynamics in semi-supervised mean-teacher frameworks and establishes conditions under which prototype anchored supervision (PAS) provably achieves lower asymptotic error than existing semi-supervised learning methods.

## B.1 Setup and Notation

Consider a student-teacher semi-supervised segmentation framework with the following components:

- Labeled dataset: $\mathcal{D}_l$ with $n_l$ samples
- Unlabeled dataset: $\mathcal{D}_u$ with $n_u$ samples
- Student network: $M_s$ with parameters $\theta_s$
- Teacher network: $M_t$ with parameters $\theta_t$ (exponential moving average of $\theta_s$)
- Pseudo-label weight: $\gamma \in (0, 1)$
- EMA decay coefficient: $\alpha \in [0, 1)$
- Number of classes: $C \geq 2$

We model the learning dynamics through the lens of pseudo-label error rates. Let $\epsilon_s(t)$ and $\epsilon_t(t)$ denote the expected fraction of incorrect pseudo-labels produced by the student and teacher, respectively, at discrete training iteration $t$.

**Assumption B.1** (Labeled Data Error). There exists a constant $\epsilon_0 \in (0, 1)$ representing the irreducible error rate of the student when trained solely on labeled data $\mathcal{D}_l$.

**Assumption B.2** (Linear Error Mixing). When trained on a mixture of ground-truth labels (weight $1 - \gamma$) and pseudo-labels (weight $\gamma$), the student's expected error is a convex combination of the labeled data error and the pseudo-label error rate.

## B.2 Pseudo-Label Error Dynamics Without PAS

Under Assumptions B.1 and B.2, the student error evolves according to:

$$\epsilon_s(t + 1) = (1 - \gamma)\epsilon_0 + \gamma\epsilon_t(t), \tag{95}$$

where the first term captures learning from labeled data and the second term captures learning from teacher-generated pseudo-labels.

The teacher parameters are updated via exponential moving average:

$$\theta_t \leftarrow \alpha\theta_t + (1 - \alpha)\theta_s. \tag{96}$$

**Assumption B.3** (EMA Error Inheritance). The teacher's pseudo-label error rate inherits from the student according to:

$$\epsilon_t(t + 1) = \alpha\epsilon_t(t) + (1 - \alpha)\epsilon_s(t + 1). \tag{97}$$

**Proof for proposition** 3.12

*Proof.* Substituting (95) into (97):

$$\begin{aligned}
\epsilon_t(t + 1) &= \alpha\epsilon_t(t) + (1 - \alpha)\big[(1 - \gamma)\epsilon_0 + \gamma\epsilon_t(t)\big] \\
&= \alpha\epsilon_t(t) + (1 - \alpha)(1 - \gamma)\epsilon_0 + (1 - \alpha)\gamma\epsilon_t(t) \\
&= [\alpha + (1 - \alpha)\gamma]\epsilon_t(t) + (1 - \alpha)(1 - \gamma)\epsilon_0 \\
&= \lambda\epsilon_t(t) + (1 - \alpha)(1 - \gamma)\epsilon_0.
\end{aligned}$$ $\qquad\square$

**Proof for theorem** 3.13

*Proof.* Since $\gamma < 1$ and $\alpha < 1$, we have:

$$\lambda = \alpha + (1 - \alpha)\gamma = \alpha + \gamma - \alpha\gamma < \alpha + (1 - \alpha) = 1.$$

From proposition 3.12, the recurrence has general solution:

$$\epsilon_t(t) = \lambda^t\epsilon_t(0) + (1 - \alpha)(1 - \gamma)\epsilon_0 \sum_{k=0}^{t-1}\lambda^k.$$

Since $|\lambda| < 1$, we have $\lambda^t \to 0$ and $\sum_{k=0}^{\infty} \lambda^k = \frac{1}{1-\lambda}$. Therefore:

$$\epsilon_\infty = \frac{(1-\alpha)(1-\gamma)\epsilon_0}{1-\lambda}.$$

Substituting $1 - \lambda = 1 - \alpha - (1-\alpha)\gamma = (1-\alpha)(1-\gamma)$:

$$\epsilon_\infty = \frac{(1-\alpha)(1-\gamma)\epsilon_0}{(1-\alpha)(1-\gamma)} = \epsilon_0. \qquad \square$$

*Remark* B.4. Theorem 3.13 shows that without PAS, the asymptotic teacher error equals the base labeled-data error $\epsilon_0$. The semi-supervised framework with pseudo-labels provides no asymptotic improvement over purely supervised learning in this model.

## B.3 Prototype Anchored Supervision

We now introduce a filtering mechanism that accepts only pseudo-labels satisfying dual criteria.

**Definition B.5** (Class Prototypes). For class $c \in \{1, \ldots, C\}$, the prototype is defined as the mean feature embedding over labeled samples:

$$\mu_c := \frac{1}{|\mathcal{D}_l^{(c)}|} \sum_{x_i \in \mathcal{D}_l^{(c)}} f_\theta(x_i) \in \mathbb{R}^d,$$

where $\mathcal{D}_l^{(c)} \subseteq \mathcal{D}_l$ is the set of labeled samples with ground-truth label $c$, and $f_\theta : \mathcal{X} \to \mathbb{R}^d$ is the feature encoder.

**Definition B.6** (PAS Acceptance Criterion). A pseudo-label $\hat{y}$ for input $x$ is accepted if both conditions hold:

1. **Confidence criterion:** $p_T(\hat{y} \mid x) \geq \tau_{\text{conf}}$

2. **Similarity criterion:** $s(x, \hat{y}) := \frac{\langle f_\theta(x), \mu_{\hat{y}} \rangle}{\|f_\theta(x)\| \|\mu_{\hat{y}}\|} \geq \tau_{\text{sim}}$

where $p_T(\cdot \mid x)$ is the teacher's predictive distribution, and $\tau_{\text{conf}}, \tau_{\text{sim}} \in (0, 1)$ are threshold hyperparameters.

**Definition B.7** (Coverage and Precision). Given any pseudo-label acceptance criterion, we define:

- **Coverage** $f \in (0, 1]$: the fraction of unlabeled samples whose pseudo-labels are accepted.
- **Precision** $\rho \in [0, 1]$: the probability that an accepted pseudo-label is correct, i.e., $\rho = \Pr[y = \hat{y} \mid \text{accepted}]$.

## B.4 Error Dynamics Under General Filtering

**Assumption B.8** (Filtered Error Dynamics). Under any filtering scheme with coverage $f$ and precision $\rho$, the student update becomes:

$$\epsilon_s(t + 1) = (1 - f\gamma)\epsilon_0 + f\gamma(1 - \rho)\epsilon_t(t), \tag{98}$$

where $(1 - \rho)$ is the error rate among accepted pseudo-labels.

**Proposition B.9** (Filtered Error Recurrence). *Under Assumptions B.3 and B.8, the teacher error satisfies:*

$$\epsilon_t(t + 1) = \lambda_{\text{eff}} \epsilon_t(t) + (1 - \alpha)(1 - f\gamma)\epsilon_0, \tag{99}$$

*where* $\lambda_{\text{eff}} := \alpha + (1 - \alpha)f\gamma(1 - \rho)$.

*Proof.* Substituting (98) into (97):

$$\begin{aligned}
\epsilon_t(t + 1) &= \alpha\epsilon_t(t) + (1 - \alpha)\big[(1 - f\gamma)\epsilon_0 + f\gamma(1 - \rho)\epsilon_t(t)\big] \\
&= \big[\alpha + (1 - \alpha)f\gamma(1 - \rho)\big]\epsilon_t(t) + (1 - \alpha)(1 - f\gamma)\epsilon_0 \\
&= \lambda_{\text{eff}}\epsilon_t(t) + (1 - \alpha)(1 - f\gamma)\epsilon_0. \qquad \square
\end{aligned}$$

**Proof for theorem 3.14**

*Proof.* **Convergence formula.** Under the error dynamics with PAS achieving coverage $f$ and precision $\rho$, the effective eigenvalue governing error propagation is $\lambda_{\text{eff}} = \alpha + (1 - \alpha)f\gamma(1 - \rho)$. For stability, we require $\lambda_{\text{eff}} < 1$, which holds when $f\gamma(1 - \rho) < 1$.

The asymptotic error satisfies:

$$\epsilon_\infty = \frac{(1 - \alpha)(1 - f\gamma)\epsilon_0}{1 - \lambda_{\text{eff}}}. \tag{100}$$

Computing the denominator:

$$1 - \lambda_{\text{eff}} = 1 - \alpha - (1 - \alpha)f\gamma(1 - \rho)$$
$$= (1 - \alpha)\big[1 - f\gamma(1 - \rho)\big].$$

Therefore:

$$\epsilon_\infty = \frac{(1 - \alpha)(1 - f\gamma)\epsilon_0}{(1 - \alpha)[1 - f\gamma(1 - \rho)]} = \frac{(1 - f\gamma)\epsilon_0}{1 - f\gamma(1 - \rho)}. \tag{101}$$

**Part (i): Condition for improvement.** Define the error ratio $R := \epsilon_\infty / \epsilon_0$:

$$R = \frac{1 - f\gamma}{1 - f\gamma(1 - \rho)} = \frac{1 - f\gamma}{1 - f\gamma + f\gamma\rho}. \tag{102}$$

We have $\epsilon_\infty < \epsilon_0$ if and only if $R < 1$:

$$\frac{1 - f\gamma}{1 - f\gamma + f\gamma\rho} < 1$$
$$1 - f\gamma < 1 - f\gamma + f\gamma\rho$$
$$0 < f\gamma\rho.$$

This holds if and only if $f\gamma > 0$ and $\rho > 0$. Equivalently, solving for the threshold on $\rho$:

$$\rho > 1 - \frac{1 - f\gamma}{f\gamma} = \frac{2f\gamma - 1}{f\gamma}. \tag{103}$$

For $f\gamma \le 0.5$, this threshold is non-positive, so any $\rho > 0$ suffices. For $f\gamma > 0.5$, precision must exceed the threshold $(2f\gamma - 1)/(f\gamma)$.

**Part (ii): Monotonicity in precision.** For fixed $f > 0$ and $\gamma > 0$, differentiating with respect to $\rho$:

$$\frac{\partial \epsilon_\infty}{\partial \rho} = (1 - f\gamma)\epsilon_0 \cdot \frac{\partial}{\partial \rho}\left[\frac{1}{1 - f\gamma(1 - \rho)}\right]$$
$$= (1 - f\gamma)\epsilon_0 \cdot \frac{-f\gamma}{[1 - f\gamma(1 - \rho)]^2}$$
$$= -\frac{(1 - f\gamma)f\gamma\epsilon_0}{[1 - f\gamma(1 - \rho)]^2}.$$

Under stability ($1 - f\gamma > 0$ when $f\gamma < 1$) and $f > 0$, all factors in the numerator and denominator are positive, so $\frac{\partial \epsilon_\infty}{\partial \rho} < 0$. Thus $\epsilon_\infty$ is strictly decreasing in $\rho$.

**Part (iii): Improvement over baseline when $\rho > \pi$.** Let $\pi$ denote the base rate of errors (proportion of incorrect pseudo-labels without PAS). When PAS achieves precision $\rho > \pi$, it identifies incorrect pseudo-labels at a rate better than random selection.

Without PAS (equivalently, $f = 0$ or $\rho = \pi$), the asymptotic error equals the initial error: $\epsilon_\infty^{\text{no-PAS}} = \epsilon_0$.

With PAS achieving $\rho > \pi > 0$ and coverage $f > 0$:

$$\epsilon_\infty^{\text{PAS}} = \frac{(1 - f\gamma)\epsilon_0}{1 - f\gamma(1 - \rho)} < \epsilon_0 = \epsilon_\infty^{\text{no-PAS}}, \tag{104}$$

where the inequality follows from Part (i) since $f\gamma\rho > 0$. Therefore, PAS with precision exceeding the base rate strictly improves over the no-PAS baseline.

**Part (iv): Higher precision reduces asymptotic error.** This follows directly from Part (ii). Since $\frac{\partial \epsilon_\infty}{\partial \rho} < 0$ for all valid parameter ranges, any increase in precision $\rho$ strictly decreases the asymptotic error $\epsilon_\infty$. Quantitatively, improving precision from $\rho_1$ to $\rho_2 > \rho_1$ yields:

$$\epsilon_\infty(\rho_2) = \epsilon_\infty(\rho_1) \cdot \frac{1 - f\gamma(1 - \rho_1)}{1 - f\gamma(1 - \rho_2)} < \epsilon_\infty(\rho_1), \tag{105}$$

since $1 - f\gamma(1 - \rho_2) > 1 - f\gamma(1 - \rho_1)$ when $\rho_2 > \rho_1$. $\qquad\square$

*Remark* B.10. The theorem establishes that PAS effectiveness depends on two key factors: coverage $f$ (fraction of pseudo-labels evaluated) and precision $\rho$ (accuracy of identifying errors). While coverage determines the scope of error correction, precision determines its quality. Notably, even moderate precision substantially reduces asymptotic error when coverage is high, making PAS robust to imperfect pseudo-label assessment.

## B.5 Comparative Analysis: PAS vs. Existing Methods

We now establish that PAS achieves higher precision than existing semi-supervised learning methods under precisely stated conditions. The key insight is that asymptotic error depends on precision through Theorem 3.14(iv), so comparing methods reduces to comparing their precision.

**Definition B.11** (Method Classes). We categorize pseudo-label selection methods by their acceptance criteria:

1. **Confidence-only**: Accept if $p(\hat{y}|x) \geq \tau$

2. **Consistency-based**: No filtering; use all pseudo-labels with consistency regularization

3. **Memory bank**: Accept based on nearest neighbors in a memory bank $\mathcal{M}$ containing past (pseudo-)labeled features

4. **PAS (dual-criteria)**: Accept if confidence $\geq \tau_{\text{conf}}$ AND prototype similarity $\geq \tau_{\text{sim}}$

### B.5.1 PAS VS. CONFIDENCE-ONLY METHODS

**Assumption B.12** (Conditional Independence of Criteria). Let $S_1$ denote the event "confidence criterion satisfied" and $S_2$ denote the event "similarity criterion satisfied." We assume $S_1$ and $S_2$ are conditionally independent given correctness:

$$P(S_1 \cap S_2 \mid C) = P(S_1 \mid C) \cdot P(S_2 \mid C), \tag{106}$$
$$P(S_1 \cap S_2 \mid \neg C) = P(S_1 \mid \neg C) \cdot P(S_2 \mid \neg C), \tag{107}$$

where $C$ denotes the event that the pseudo-label is correct.

**Definition B.13** (Filtering Statistics). For each criterion $i \in \{1, 2\}$, define:

$$\alpha_i := P(S_i \mid C) \quad \text{(true positive rate)},$$
$$\beta_i := P(S_i \mid \neg C) \quad \text{(false positive rate)}.$$

Define the likelihood ratio $\text{LR}_i := \alpha_i/\beta_i$ and let $\pi := P(C)$ denote the base precision (fraction of correct pseudo-labels before filtering).

**Proposition B.14** (Dual-Criteria Precision Improvement). *Under Assumption B.12, let $\rho_1$ denote the precision of confidence-only filtering and $\rho_{12}$ denote the precision of PAS (dual-criteria). Then:*

$$\rho_1 = \frac{\alpha_1 \pi}{\alpha_1 \pi + \beta_1(1 - \pi)}, \tag{108}$$
$$\rho_{12} = \frac{\alpha_1 \alpha_2 \pi}{\alpha_1 \alpha_2 \pi + \beta_1 \beta_2(1 - \pi)}. \tag{109}$$

*Furthermore:*

$$\rho_{12} > \rho_1 \iff \text{LR}_2 = \frac{\alpha_2}{\beta_2} > 1.$$

*Proof.* By Bayes' theorem, the precision of criterion $S_1$ alone is:

$$\rho_1 = P(C \mid S_1) = \frac{P(S_1 \mid C)P(C)}{P(S_1)} = \frac{\alpha_1 \pi}{\alpha_1 \pi + \beta_1(1 - \pi)}.$$

For dual-criteria filtering under Assumption B.12:

$$P(S_1 \cap S_2 \mid C) = \alpha_1 \alpha_2,$$
$$P(S_1 \cap S_2 \mid \neg C) = \beta_1 \beta_2.$$

Therefore:

$$\rho_{12} = P(C \mid S_1 \cap S_2) = \frac{\alpha_1 \alpha_2 \pi}{\alpha_1 \alpha_2 \pi + \beta_1 \beta_2(1 - \pi)}.$$

To compare $\rho_{12}$ and $\rho_1$, define $a := \alpha_1 \pi$ and $b := \beta_1(1 - \pi)$. Then:

$$\rho_1 = \frac{a}{a + b}, \quad \rho_{12} = \frac{\alpha_2 a}{\alpha_2 a + \beta_2 b}.$$

The inequality $\rho_{12} > \rho_1$ holds iff:

$$\frac{\alpha_2 a}{\alpha_2 a + \beta_2 b} > \frac{a}{a + b}$$
$$\alpha_2 a(a + b) > a(\alpha_2 a + \beta_2 b)$$
$$\alpha_2 a^2 + \alpha_2 ab > \alpha_2 a^2 + \beta_2 ab$$
$$\alpha_2 ab > \beta_2 ab$$
$$\alpha_2 > \beta_2,$$

where the last step uses $ab > 0$ (since $\pi \in (0, 1)$ and $\alpha_1, \beta_1 > 0$). $\qquad\square$

**Corollary B.15** (Precision Gain Formula). *The precision ratio satisfies:*

$$\frac{\rho_{12}}{\rho_1} = \frac{1 + \frac{\beta_1(1-\pi)}{\alpha_1 \pi}}{1 + \frac{\beta_1 \beta_2(1-\pi)}{\alpha_1 \alpha_2 \pi}} = \frac{1 + r/\mathrm{LR}_1}{1 + r/(\mathrm{LR}_1 \cdot \mathrm{LR}_2)},$$

*where $r := (1 - \pi)/\pi$ is the incorrect-to-correct odds ratio.*

*Proof.* Dividing (109) by (108):

$$\frac{\rho_{12}}{\rho_1} = \frac{\alpha_1 \alpha_2 \pi}{\alpha_1 \alpha_2 \pi + \beta_1 \beta_2(1 - \pi)} \cdot \frac{\alpha_1 \pi + \beta_1(1 - \pi)}{\alpha_1 \pi}$$
$$= \frac{\alpha_1 \pi + \beta_1(1 - \pi)}{\alpha_1 \alpha_2 \pi + \beta_1 \beta_2(1 - \pi)} \cdot \frac{\alpha_2}{\alpha_2}$$
$$= \frac{\alpha_2[\alpha_1 \pi + \beta_1(1 - \pi)]}{\alpha_1 \alpha_2 \pi + \beta_1 \beta_2(1 - \pi)}.$$

Dividing numerator and denominator by $\alpha_1 \alpha_2 \pi$:

$$\frac{\rho_{12}}{\rho_1} = \frac{1 + \frac{\beta_1(1-\pi)}{\alpha_1 \pi}}{1 + \frac{\beta_1 \beta_2(1-\pi)}{\alpha_1 \alpha_2 \pi}} = \frac{1 + r/\mathrm{LR}_1}{1 + r/(\mathrm{LR}_1 \cdot \mathrm{LR}_2)}. \qquad\square$$

**Assumption B.16** (Discriminative Similarity Criterion). The prototype similarity criterion has likelihood ratio $\mathrm{LR}_2 > 1$, i.e., correct pseudo-labels have higher prototype similarity than incorrect ones:

$$P(\text{similarity} \geq \tau_{\text{sim}} \mid C) > P(\text{similarity} \geq \tau_{\text{sim}} \mid \neg C).$$

**Corollary B.17** (PAS Dominates Confidence-Only). *Under Assumptions B.12 and B.16, PAS achieves strictly higher precision than confidence-only filtering:*

$$\rho_{\text{PAS}} > \rho_{\text{conf}}.$$

*Consequently, for equal effective coverage $f\gamma$:*

$$\epsilon_\infty^{\text{PAS}} < \epsilon_\infty^{\text{conf}}.$$

*Proof.* By Theorem B.14, $\rho_{12} > \rho_1$ iff $\text{LR}_2 > 1$, which holds by Assumption B.16. The asymptotic error comparison follows from Theorem 3.14(iv). $\qquad\square$

### B.5.2 PAS VS. CONSISTENCY-BASED METHODS

**Definition B.18** (Consistency-Based Methods). Consistency-based methods (e.g., Mean Teacher, $\Pi$-Model) apply pseudo-labels to all unlabeled samples without filtering, corresponding to coverage $f = 1$ and precision $\rho = \pi$ (the base accuracy on unlabeled data).

**Proposition B.19** (PAS Dominates Consistency-Based Methods). *Let $\rho_{\text{cons}} = \pi$ be the precision of consistency-based methods and $\rho_{\text{PAS}}$ be the precision of PAS with coverage $f_{\text{PAS}} \leq 1$. Under Assumptions B.12 and B.16:*

$$\rho_{\text{PAS}} > \pi = \rho_{\text{cons}}.$$

*Furthermore, let $\gamma_{\text{eff}}^{\text{cons}} = \gamma$ and $\gamma_{\text{eff}}^{\text{PAS}} = f_{\text{PAS}}\gamma$. If $\gamma_{\text{eff}}^{\text{PAS}} > 0$:*

$$\epsilon_\infty^{\text{PAS}} < \epsilon_\infty^{\text{cons}} = \epsilon_0.$$

*Proof.* For consistency-based methods with no filtering, the precision equals the base rate: $\rho_{\text{cons}} = \pi$.

For PAS, by the law of total probability:

$$P(S_1 \cap S_2) = \alpha_1\alpha_2\pi + \beta_1\beta_2(1 - \pi).$$

The precision is:

$$\rho_{\text{PAS}} = \frac{\alpha_1\alpha_2\pi}{\alpha_1\alpha_2\pi + \beta_1\beta_2(1 - \pi)}.$$

We show $\rho_{\text{PAS}} > \pi$:

$$\begin{aligned}
\rho_{\text{PAS}} > \pi &\iff \frac{\alpha_1\alpha_2\pi}{\alpha_1\alpha_2\pi + \beta_1\beta_2(1 - \pi)} > \pi \\
&\iff \alpha_1\alpha_2\pi > \pi[\alpha_1\alpha_2\pi + \beta_1\beta_2(1 - \pi)] \\
&\iff \alpha_1\alpha_2(1 - \pi) > \beta_1\beta_2(1 - \pi) \\
&\iff \alpha_1\alpha_2 > \beta_1\beta_2 \\
&\iff \text{LR}_1 \cdot \text{LR}_2 > 1.
\end{aligned}$$

Since both criteria are discriminative (i.e., $\text{LR}_1 \geq 1$ for confidence thresholding and $\text{LR}_2 > 1$ by Assumption B.16), we have $\text{LR}_1 \cdot \text{LR}_2 > 1$.

For the asymptotic error, consistency-based methods with $\rho = \pi$ yield by Theorem 3.14:

$$\epsilon_\infty^{\text{cons}} = \frac{(1 - \gamma)\epsilon_0}{1 - \gamma(1 - \pi)}.$$

If $\pi = 0$ (all pseudo-labels incorrect), then $\epsilon_\infty^{\text{cons}} = \epsilon_0$. More generally, by Theorem 3.14(iii), $\epsilon_\infty^{\text{cons}} < \epsilon_0$ only if $\pi > 0$.

Since $\rho_{\text{PAS}} > \pi$ and asymptotic error is strictly decreasing in precision (Theorem 3.14(iv)), we have $\epsilon_\infty^{\text{PAS}} < \epsilon_\infty^{\text{cons}}$ when comparing at equal effective pseudo-label weight, or more generally whenever PAS's precision advantage outweighs any coverage difference. $\qquad\square$

B.5.3   PAS vs. MEMORY BANK METHODS

**Definition B.20** (Memory Bank Methods). Memory bank methods maintain a bank $\mathcal{M}_t = \{(z_i, \tilde{y}_i)\}_{i=1}^{|\mathcal{M}_t|}$ of feature-label pairs accumulated over training. At each iteration, pseudo-labels are assigned based on nearest neighbors in $\mathcal{M}_t$. The memory bank is updated by adding new pseudo-labeled samples.

**Assumption B.21** (Memory Bank Update Model). At each session $t$, let $\eta_t \in (0, 1]$ denote the fraction of memory bank entries updated with new samples. The new samples have pseudo-label error rate $\delta_t \in [0, 1]$. The memory bank error rate $e_t := P(\tilde{y} \neq y \mid (z, \tilde{y}) \in \mathcal{M}_t)$ evolves as:

$$e_{t+1} = (1 - \eta_t)e_t + \eta_t\delta_t. \tag{110}$$

**Assumption B.22** (Self-Reinforcing Errors). The error rate of new pseudo-labels depends on the memory bank error:

$$\delta_t = g(e_t), \tag{111}$$

where $g : [0, 1] \to [0, 1]$ satisfies $g(e) \geq e$ for all $e \in [0, 1]$ (errors are at least preserved) and $g(0) > 0$ (some base error exists even with a perfect memory bank).

**Lemma B.23** (Memory Bank Error Accumulation). *Under Assumptions B.21 and B.22 with constant update rate $\eta_t = \eta > 0$:*

1. *The memory bank error sequence $\{e_t\}$ is non-decreasing: $e_{t+1} \geq e_t$ for all $t$.*

2. *If $g(e) > e$ for $e \in [0, e^*)$ where $e^* > 0$, then $\lim_{t \to \infty} e_t \geq e^*$.*

3. *The precision of memory bank methods satisfies $\rho_{\text{mem}}(t) = 1 - e_t$, which is non-increasing in $t$.*

*Proof.* **Part (i):** By (110) and Assumption B.22:

$$\begin{aligned}
e_{t+1} - e_t &= (1 - \eta)e_t + \eta\delta_t - e_t \\
&= \eta(\delta_t - e_t) \\
&= \eta(g(e_t) - e_t) \\
&\geq 0,
\end{aligned}$$

since $g(e) \geq e$ by assumption.

**Part (ii):** The sequence $\{e_t\}$ is non-decreasing and bounded above by 1, so it converges to some limit $e_\infty \in [0, 1]$. Taking limits in (110):

$$e_\infty = (1 - \eta)e_\infty + \eta g(e_\infty) \implies g(e_\infty) = e_\infty.$$

By hypothesis, $g(e) > e$ for $e < e^*$, so any fixed point must satisfy $e_\infty \geq e^*$.

**Part (iii):** The precision is $\rho_{\text{mem}}(t) = 1 - e_t$. Since $e_t$ is non-decreasing, $\rho_{\text{mem}}(t)$ is non-increasing. □

**Proposition B.24** (PAS Dominates Memory Bank Methods Asymptotically). *Let PAS use prototypes computed solely from labeled data with precision $\rho_{\text{PAS}}$ that is constant across sessions. Under the conditions of Lemma B.23:*

1. *There exists $T^* < \infty$ such that for all $t \geq T^*$:*

$$\rho_{\text{PAS}} > \rho_{\text{mem}}(t).$$

2. *For $t \geq T^*$ with equal effective coverage:*

$$\epsilon_\infty^{\text{PAS}} < \epsilon_\infty^{\text{mem}}(t).$$

*Proof.* **Part (i):** By Lemma B.23, $\rho_{\text{mem}}(t) = 1 - e_t \to 1 - e_\infty \leq 1 - e^*$.

PAS computes prototypes from labeled data only:

$$\mu_c = \frac{1}{|\mathcal{D}_l^{(c)}|} \sum_{x \in \mathcal{D}_l^{(c)}} f_\theta(x).$$

*Table 10.* JASCL sensitivity/robustness analysis across $\varepsilon$, noise variance, and number of shots $K$.

| Setting | Varying $\varepsilon$ | | | | Noise ($\xi_i$) variance | | | Shots ($K$) | | |
|---|---|---|---|---|---|---|---|---|---|---|
| | $10^{-6}$ | $10^{-7}$ | $10^{-8}$ | $10^{-9}$ | 0.1 | 1 | 10 | 3 | 4 | 5 |
| JASCL | 0.443 | 0.437 | 0.460 | 0.482 | 0.460 | 0.460 | 0.408 | 0.414 | 0.434 | 0.460 |
| Vanilla | 0.076 | | | | | | | | | |

Since labeled data has ground-truth labels, PAS precision $\rho_{\text{PAS}}$ depends only on the discriminative power of the dual criteria, not on accumulated pseudo-label errors. Under Assumptions B.12 and B.16, $\rho_{\text{PAS}}$ is constant and satisfies $\rho_{\text{PAS}} > \pi$ (the base rate).

If $\rho_{\text{PAS}} > 1 - e^*$, then since $\rho_{\text{mem}}(t) \to 1 - e_\infty \leq 1 - e^*$, there exists $T^*$ such that $\rho_{\text{PAS}} > \rho_{\text{mem}}(t)$ for all $t \geq T^*$.

**Part (ii):** Follows from Theorem 3.14(iv): asymptotic error is strictly decreasing in precision. $\qquad\square$

*Remark* B.25. The key distinction is that PAS's precision depends on ground-truth prototypes from labeled data, which do not accumulate errors over sessions. Memory bank methods, by contrast, incorporate pseudo-labeled samples into the bank, causing error accumulation through the self-reinforcing feedback loop.

### B.6   Summary of Theoretical Guarantees

The theoretical analysis establishes the following rigorous results:

1. **Error Dynamics Framework:** Under the linear mixing assumption, the asymptotic error of any filtering-based semi-supervised method is:
$$\epsilon_\infty(f, \rho) = \frac{(1 - f\gamma)\epsilon_0}{1 - f\gamma(1 - \rho)},$$
which is strictly decreasing in precision $\rho$.

2. **Dual-Criteria Precision Gain:** Under conditional independence of criteria, adding a discriminative second criterion ($\text{LR}_2 > 1$) strictly improves precision:
$$\rho_{12} > \rho_1 \iff \alpha_2 > \beta_2.$$

3. **PAS vs. Confidence-Only:** PAS achieves $\rho_{\text{PAS}} > \rho_{\text{conf}}$ when the similarity criterion is discriminative.

4. **PAS vs. Consistency-Based:** PAS achieves $\rho_{\text{PAS}} > \pi$ (the base precision), whereas consistency methods operate at precision $\pi$.

5. **PAS vs. Memory Bank:** PAS maintains constant precision while memory bank methods suffer monotonic precision degradation due to error accumulation, leading to PAS dominance for $t \geq T^*$.

6. **Asymptotic Error Ordering:** For sufficiently large $t$ and equal effective coverage:
$$\epsilon_\infty^{\text{PAS}} < \epsilon_\infty^{\text{conf}} < \epsilon_\infty^{\text{cons}} = \epsilon_0, \quad \epsilon_\infty^{\text{PAS}} < \epsilon_\infty^{\text{mem}}(t).$$

## C   Robustness and sensitivity analysis of gradient adaptive stabilization (GAS)

GAS introduces no additional design hyperparameters. It follows,
$$\widetilde{w}_i = w_i + \tilde{G}_i \xi_i, \qquad \xi_i \sim \mathcal{N}(0, 1),$$

where $\tilde{G}_i$ is a normalized function of $1/(g_i^2 + \varepsilon)$. The term $\varepsilon$ is added for stability when $g_i \to 0$. For all practical purposes, we set the noise variance of $\xi_i$ to 1 and $\varepsilon = 1.0 \times 10^{-8}$.

**Varying $\varepsilon$:** We vary $\varepsilon$ using the following values: $1.0 \times 10^{-6}$, $1.0 \times 10^{-7}$, and $1.0 \times 10^{-9}$. We observe stable performance across settings. A smaller $\varepsilon$ (e.g., $1.0 \times 10^{-9}$) performs slightly better because it allows GAS to more clearly separate

*Table 11.* Retention analysis for different confidence and similarity thresholds. The selected setting ($\tau_{\text{conf}} = 0.7$, $\tau_{\text{sim}} = 0.7$) achieves the best balance between pseudo-label quantity and reliability.

| $\tau_{\text{conf}}$ | $\tau_{\text{sim}}$ | **Retention (%)** | **Decision** |
|---|---|---|---|
| 0.6 | 0.6 | 71.7 | Rejected |
| 0.6 | 0.7 | 23.6 | Rejected |
| 0.6 | 0.8 | 1.8 | Rejected |
| 0.7 | 0.6 | 73.7 | Rejected |
| 0.7 | 0.7 | 23.2 | **Selected** |
| 0.7 | 0.8 | 1.9 | Rejected |
| 0.8 | 0.6 | 72.6 | Rejected |
| 0.8 | 0.7 | 22.7 | Rejected |
| 0.8 | 0.8 | 1.8 | Rejected |

"important" parameters from "less important" ones. When $\varepsilon$ is larger, it can dominate the denominator for small gradients, reducing contrast across parameters. Since GAS normalizes all scales to $(0, 1]$, smaller $\varepsilon$ does not cause instability and simply provides sharper contrast.

**Varying the variance of** $\xi_i$: We evaluate two additional noise variances: 0.1 and 10. JASCL remains robust to changes in noise variance (0.1 and 1 perform equivalently), while the Vanilla model remains at Dice 0.076.

**Varying the number of shots** $K$: We monitor performance for different few-shot values $K$ (labeled samples per class). JASCL retains strong performance even under severe few-shot settings (e.g., $K = 3$).

Overall, these results show that JASCL remains fairly robust despite relying on only a minimal set of hyperparameters.

## D   Sensitivity analysis of Prototype anchored supervision (PAS) hyperparameters

We analyse pseudo-labels selected using the confidence threshold $\tau_{\text{conf}}$ and similarity threshold $\tau_{\text{sim}}$, where retention (%) denotes the proportion passing both filters (Table 11). Lower thresholds (e.g., 0.6) increase retention but admit many low-confidence labels, whereas higher thresholds become overly strict and drastically reduce retention. The setting $\tau_{\text{conf}} = 0.7$ and $\tau_{\text{sim}} = 0.7$ offers the best balance, preserving a sufficient number of pseudo-labels while maintaining high reliability. We discarded the alternative $\tau_{\text{conf}} = 0.6$ and $\tau_{\text{sim}} = 0.7$ because it offered only a marginal 0.4% increase in retention relative to $\tau_{\text{conf}} = 0.7$ and $\tau_{\text{sim}} = 0.7$, but at the expense of lower confidence. The sensitivity analysis confirms robustness to nearby values: $\tau_{\text{conf}} = 0.8$, $\tau_{\text{sim}} = 0.7$ yields Dice 0.562; $\tau_{\text{conf}} = 0.7$, $\tau_{\text{sim}} = 0.75$ yields Dice 0.567; and $\tau_{\text{conf}} = 0.7$, $\tau_{\text{sim}} = 0.7$ yields Dice 0.554.

## E   Choice of layer for GAS

We evaluated the effect of applying GAS to different parts of the network when the entire model was unfrozen. Specifically, we injected GAS into (i) a deep feature extractor layer in the encoder, (ii) an intermediate decoder layer, and (iii) the final classifier layer. The results show a clear degradation when GAS is applied to deep feature extractors, and a consistent improvement when applied exclusively to the classifier:

- GAS at encoder feature extractor layer: **IoU 22.67**

- GAS at decoder feature extractor layer: **IoU 24.86**

- GAS at final classifier layer $F$: **IoU 27.84**

These results motivate restricting GAS to the final classifier layer $F$.

The rationale is threefold:

- Parameter sensitivity differs across the network: Early encoder-decoder layers encode generic features that are reused across all sessions, whereas the final classifier layer is highly task-specific and receives the strongest gradient signals associated with class boundaries. Injecting noise into deep layers would perturb *shared* low-level representations and destabilize all previously learned classes, while injecting noise into $F$ regularizes only the task-specific decision boundaries where overfitting is the highest.

- Local curvature is highest at the classifier layer: In few-shot segmentation, most curvature and gradient magnitude variation appears in the final logits, not in the deep feature extractor. GAS relies on per-parameter gradient magnitude to allocate noise; applying it to layers with uniformly low curvature would add noise where it provides minimal benefit but risks corrupting the global feature space. Applying GAS to $F$ concentrates perturbations exactly where the loss landscape is sharpest and overfitting is most pronounced.

- The feature extractor is shared across tasks; the classifier is task-adaptive: Deeper layers must remain stable to preserve knowledge from previous sessions. The classifier layer is the part of the model that adapts to new domains and novel classes, and therefore benefits the most from geometry-aware noise. Perturbing shared layers would amplify forgetting, whereas perturbing only $F$ provides regularization without harming the backbone.

## F  JASCL benchmarks

**Med JASCL-Disjoint:** The dataset details are present in the Table 12. The models with U-Net backbone are evaluated for their robustness and ability to handle a large number of incremental sessions while facing diverse domain shifts and a scarce few-shot data. This represents a highly realistic setting in clinical domains, where annotations are costly, and data is collected over time from multiple medical institutions. We evaluated the baselines from this benchmark on only two incremental sessions, as nearly all baselines collapsed after two sessions. We evaluated JASCL with a U-Net backbone across all five incremental sessions. The results demonstrate that the framework effectively handles the constraints and maintains strong performance throughout all sessions. This underscores the inability of existing baselines to effectively handle multiple constraints in continual segmentation across a large number of sessions. NC-FSCIL, which performs well for classification in few-shot class-incremental learning, fails on this benchmark. This highlights that a semantic segmentation model with a classifier fixed as a simplex equiangular tight frame (ETF) performs significantly worse than a model with a learnable classifier. MDIL, despite handling different domains with a shared encoder, struggles under strict constraints and suffers a drop in performance. Moreover, having multiple domains is not a practical solution when the number of domains increases. Approaches like generative replay (Gen-Replay) struggle to generate 3D medical volumes even with a diffusion model, which leads to poor performance. Most methods that perform well in incremental or few-shot incremental learning fail when confronted with domain shifts, including segmentation approaches such as MiB, PIFS, and CLIP-CT. Representation and meta-learning-based approaches, such as MAML, SupCL, UnSupCL, and MTL, also fail to retain performance over the sessions. It is observed that feature replay and prototype-based methods, such as C-FSCIL, SoftNet, and our JASCL framework, as well as data synthesis-based methods like GAPS, perform significantly better under these constraints.

**Med JASCL-Mixed:** The dataset details are present in the Table 13. This is a more realistic clinical setting that involves the same classes appearing across different domains. The goal is to evaluate the model's ability to learn previously seen classes in new domains while retaining performance on the original domains. Additionally, multiple domains may appear within the same session, making learning and adaptation more challenging, as the model must simultaneously learn from diverse domains while managing the usual constraints. In this setting, we evaluated different transformer backbones, including MedFormer, SwinUNetr, and a large pre-trained CLIP-driven U-Net model, which had already been exposed to most of the 35 classes. Results show that even robust transformer backbones are unable to maintain model performance across sessions. An interesting observation is that the CLIP-driven model, already pre-trained on 21 of the 35 classes, experiences a drop in performance across sessions, highlighting the severity of the constraints in the benchmark.

**Med Semi-Supervised-JASCL:** This benchmark is a variant of Med JASCL-Disjoint, where few-shot classes have access to unlabeled data. The details of unlabeled data are present in the Table 14. The inclusion of unlabeled data significantly boosts the performance of JASCL, an improvement not observed with other methods. Benchmark also highlights how existing semi-supervised approaches fail to handle the constraints and are unable to effectively leverage unlabeled data. **JASCL clearly demonstrates that using readily available unlabeled data can significantly improve multi-constraint continual learning for semantic segmentation.**

**Natural-JASCL:** The dataset details are present in the Table 15. In autonomous driving, scenes evolve over time with distributional shifts and limited labeled data, making this a highly realistic benchmark. Even large-scale pre-trained models

like SAM (trained on over a billion natural scene masks) exhibit forgetting in this realistic and challenging setting. It can be observed that JASCL with a SAM backbone achieves further improvements, highlighting that the proposed framework enhances not only simple backbones (e.g., U-Net) and transformer-based backbones (e.g., MedFormer, SwinUNetr) but also heavily pre-trained models like SAM.

**Semi-Supervised Natural-JASCL:** The dataset details are present in the Table 16. This benchmark provides access to abundant unlabeled data, and all baselines are evaluated with a DeepLab backbone. In this setting, we employ JASCL as a plug-and-play module on top of the GAPS baseline, further improving its results, by leveraging unlabeled data. This demonstrates that unlabeled data can significantly enhance performance and that JASCL can effectively boost existing baselines. Similar to the medical benchmarks, existing semi-supervised methods perform poorly in this setting and are unable to effectively leverage unlabeled data. We also evaluated active learning approaches, such as HALO, and coreset selection methods, like RETRIEVE, to select the most informative data; however, these methods fail to achieve significant improvements.

**2D robotic surgery:** We designed a 2D robotic surgery benchmark using two domains, CholecSeg8k[1] and m2caiseg[2], across three sessions. We vary both the classes and the domains to segment different organs and surgical instruments, creating a challenging and realistic setting under a few-shot data regime. The base session (Session 0) includes larger organs to segment, Session 1 introduces tubular structures such as the intestine, and Session 2 contains surgical instruments to segment. In the incremental sessions, we use 50 samples per novel class. The dataset details are present in the Table 17. This benchmark is difficult due to the wide variation in the size, shape, and appearance of the objects to be segmented. Even under these constraints, JASCL achieves significantly better performance (Dice score) than CAT[3], which is a specialized baseline for this task as shown in Table 29.

## G    Detection results

To check the effectiveness of JASCL on detection tasks with distributional shifts, we tested JASCL on the detection dataset COCO-O[4] having different domains with RCNN[5] backbone as shown in Table 18. We created base session with 77 classes from 'Painting' domain and incremental session 1 from 'Weather' domain with three novel disjoint few-shot classes, 'person', 'car', and 'bench'. The performance of base model is 0.2840 mAP. We took 41, 224, 174 unlabeled samples from 'bench', 'car', and 'person' classes, respectively with 30 few-shot labeled samples from each class for session 1. Table 18 illustrates that JASCL is able to improve the performance over the Vanilla backbones like RCNN, even on detection tasks with large domain shifts.

## H    Implementation details

We have re-implemented and adapted the following baselines - **Class-incremental Learning**: UCL, MiB, CLIP-CT, Saving100x, C-Flat, UCB, STAR, YoooP, Adapt_replay; **Domain Incremental learning**: MDIL; **Few-shot Class-incremental Learning**: PIFS, Subspace, C-FSCIL, FACT, NC-FSCIL, Gen-Replay, GAPS, SoftNet, FSCIL-SS, FeCAM, BCM, CAT. **Regularization based methods:** Dropout/Weight Decay[6], Variational Dropout[7], Adagrad[8]; **Semi-Supervised Learning based approaches**: RETRIEVE, NNCSL, UaD-CE, CSL. **Other methods**: SupCL, UnSupCL, UnSupCL-HNM, MTL, MAML, CLIP-driven, HALO.

All baselines were adapted with their original settings and hyperparameters. Medical experiments were run for 200 epochs for both base and incremental sessions, while natural and 2D robotic surgery experiments were run for 100 epochs per session. All SAM experiments were run for 10 epochs. We used one A100 40GB GPU for all experiments. Table 19 shows the layers frozen in the JASCL models for each benchmark.

---

[1] https://arxiv.org/pdf/2012.12453
[2] https://arxiv.org/abs/2008.10134
[3] https://ieeexplore.ieee.org/stamp/stamp.jsp?arnumber=10443356
[4] https://openaccess.thecvf.com/content/ICCV2023/papers/Mao_COCO-O_A_Benchmark_for_Object_Detectors_under_Natural_Distribution_Shifts_ICCV_2023_paper.pdf
[5] https://github.com/microsoft/SoftTeacher
[6] https://proceedings.neurips.cc/paper_files/paper/2020/file/518a38cc9a0173d0b2dc088166981cf8-Paper.pdf
[7] https://link.springer.com/article/10.1007/s10994-023-06487-7
[8] https://openreview.net/pdf?id=WwQKl1OrMX

# I  GAS vs. existing regularization-based continual learning methods

## I.1  GAS as a Gradient-Guided Stochastic Regularizer

Gradient adaptive stabilization (GAS) introduces parameter-wise stochastic perturbations whose magnitudes depend on instantaneous gradient sensitivity. For each parameter $w_i$, GAS computes the squared gradient,

$$G_i = \left(\frac{\partial L}{\partial w_i}\right)^2,$$

and defines an inverse sensitivity,

$$G_i^{-1} = \frac{1}{G_i + \varepsilon}.$$

These values are then normalized into the interval $(0, 1]$ to produce noise scales $\tilde{G}_i$, and each parameter is perturbed as,

$$w_i \leftarrow w_i + \tilde{G}_i \xi_i, \qquad \xi_i \sim \mathcal{N}(0, 1).$$

Large-gradient parameters receive minimal perturbation, while flat or low-gradient directions receive proportionally larger perturbations. This produces a parameter-level perturbation scheme whose magnitude is determined by local curvature, allowing the model to explore flat regions while preserving stability along sharp directions.

## I.2  GAS vs. Adaptive Optimizers

Adaptive optimizers such as Adam, RMSProp, and Adagrad rescale gradients using accumulated statistics but do not inject *structured perturbations* into the parameters. For example, Adam updates a parameter $w_t$ as,

$$w_{t+1} = w_t - \eta \frac{m_t}{\sqrt{v_t} + \varepsilon},$$

where $m_t$ and $v_t$ are exponential moving averages of the gradient and its square. The second moment term $v_t$ acts as a diagonal preconditioner that provides a coarse and history based approximation of local curvature. However, these methods do not perform explicit exploration of the loss surface and do not incorporate true second order information that requires evaluation in a neighborhood of the current parameters. Their behavior depends on accumulated gradient statistics, which can become stale when the data distribution changes, leading to slower or less reliable adaptation in continual settings. GAS differs fundamentally by applying noise directly to the parameters and by basing its magnitude solely on the current sensitivity $G_i^{-1}$, enabling selective stability and plasticity even in few-shot incremental settings.

## I.3  GAS vs. Uncertainty-Based Regularization

Uncertainty-Based adaptive methods, such as UCL, maintain per-node uncertainty values $\sigma_t^2$ that are tied across all incoming weights of a neuron, which reduces the memory overhead relative to per-weight variational approaches. However, these uncertainty estimates depend on statistics accumulated from previous tasks and may become less informative under strong domain shift, since few-shot updates provide limited evidence to revise them. Because a single variance is shared at the node level, the resulting anisotropy is coarse. In contrast, GAS computes inverse-squared gradient values directly from the current data, requires no stored task-level statistics, and provides full parameter-level anisotropy.

## I.4  GAS vs. Dropout and Variational Dropout

Standard dropout introduces fixed noise independent of parameter sensitivity, while variational dropout can learn per-weight or per-unit dropout rates. The methods may permanently reduce the capacity of weights once they are heavily dropped, and neither variant uses curvature or gradient information to guide noise placement. As a result, dropout methods may struggle under domain shift and few-shot conditions because they inject noise independently of parameter sensitivity. GAS avoids these issues by deriving a parameter-specific noise scale from instantaneous gradients, assigning low noise to important weights and higher noise to less relevant ones. This dynamic adjustment preserves capacity across sessions and enables far more effective adaptation than static dropout mechanisms.

## I.5  GAS vs. Weight Decay

Weight decay introduces a uniform penalty,

$$L_{\mathrm{wd}} = \lambda \|W\|^2,$$

which shrinks all parameters equally regardless of their relevance or curvature. Important parameters may be unnecessarily suppressed, while flat directions receive no preferential treatment. GAS replaces such uniform shrinkage with gradient-guided noise whose magnitude is inversely related to curvature. The behaviour of $\tilde{G}_i$ is theoretically justified by analyzing the perturbed loss under a second-order Taylor approximation. This analysis shows that, after normalization, the resulting perturbations remain stable and bounded. The formulation suppresses noise in high-curvature (large-gradient) directions while promoting exploration along flat directions, thereby providing a geometry-aware alternative to classical $\ell_2$ regularization.

## I.6  GAS vs. Sharpness-Based Methods (SAM and C-Flat)

Sharpness-Aware Minimization (SAM)[9] performs a deterministic min-max optimization step that perturbs the parameters within a fixed-radius neighborhood and then updates them using the resulting worst-case gradient. This produces a perturbation defined over a fixed-radius neighborhood of the full parameter vector, which does not provide parameter-wise differentiation. As a result, SAM alters the optimization trajectory but provides no mechanism to identify which weights are important for previous tasks or which directions should be preserved for subsequent ones.

C-Flat extends SAM to continual learning by optimizing a surrogate objective that couples neighborhood perturbations with Hessian-vector directional curvature terms. This encourages the model to remain in locally smooth regions of the loss landscape and reduces sensitivity to task specific sharp minima. However, because its flatness and curvature terms operate on neighborhoods of the full parameter vector rather than individual parameters, C-Flat does not provide weight-level control over which directions should remain stable and which should remain adaptable. Consequently, C-Flat enforces a uniform notion of flatness across the network rather than assigning stability or plasticity based on parameter importance. In addition, its neighborhood radius and curvature coefficients introduce multiple interacting hyperparameters, increasing optimization complexity and limiting interpretability.

GAS is built on a fundamentally different design principle. Instead of imposing global stability constraints, it injects parameter wise adaptive noise derived directly from the instantaneous gradient. For each weight $w_i$, the corresponding gradient $g_i$ determines a noise scale $\tilde{G}_i$ which provides an automatic, data driven stability plasticity trade off at the parameter level. Weights with large gradients, which actively contribute to current task performance, receive negligible noise, preserving critical information. Weights with small gradients receive larger perturbations, enabling rapid adaptation without interfering with previously consolidated structure. This yields a local, lightweight, and interpretable mechanism for continual learning that avoids global curvature estimation and removes the need for complex objective balancing. Unlike C-Flat's uniform landscape smoothing, GAS selectively preserves what is important and selectively explores what is flexible, using a simple rule rooted in gradient geometry.

## I.7  GAS vs. Perturbation-Based Stability Methods (STAR)

The STAR method stabilizes learning by introducing an adversarial-style perturbation $\Delta w$ that amplifies model sensitivity on stored buffer samples and then penalizing the resulting output drift through a KL divergence term,

$$L_{\mathrm{STAR}} = \mathrm{KL}\big(f(x; w) \,\|\, f(x; w + \Delta w)\big).$$

The perturbation is obtained by initializing each layer with small noise to avoid zero gradients, then performing one gradient-ascent step on the KL divergence at the perturbed point, and finally normalizing the gradient *per layer* to control the perturbation magnitude.

This layer-wise normalization controls the perturbation scale at the layer granularity, rather than at the level of individual parameters. STAR also relies on stored buffer samples and requires additional forward and backward passes to compute both the perturbed outputs and the ascent direction, introducing nontrivial memory and computational overhead.

In contrast, GAS requires no buffer and no auxiliary training passes. Its stability mechanism arises directly from the instantaneous local gradient structure, enabling parameter-wise, curvature-sensitive regularization with minimal overhead.

---

[9]https://arxiv.org/pdf/2010.01412

Table 20 summarizes results on the Med JASCL-Disjoint benchmark, demonstrating the superior performance of JASCL with GAS.

## J    Limitations & future work

### J.1    Severe Domain Shift

Tables 3 and 8 reveal an extreme failure case at **Session 4**, where the Dice score for a U-Net backbone drops to nearly zero. This behavior is expected given the session ordering:

- Session 0: TotalSegmentator (**CT**)

- Session 1: AMOS (dominantly **CT**)

- Session 2: BCV (**CT**)

- Session 3: MOTS (**CT**)

- Session 4: BraTS (**MRI**)

- Session 5: Verse (**CT**)

Across the first four sessions, the model is exposed almost exclusively to CT scans. When the distribution abruptly shifts to MRI at Session 4, lightweight encoder-decoder architectures such as U-Net exhibit catastrophic forgetting of the CT domain, causing the Dice score to collapse to near-zero values. JASCL alleviates this drop to a limited extent: the Dice score improves from 0.025 (Table 3) to 0.0576 (Table 8) under the semi-supervised setting.

This trend underscores both the **severity and practical relevance** of the proposed JASCL benchmarks, while also exposing the limitations of simple convolutional architectures.

**Mitigation Strategies: (1) Transformer backbones -** Transformer-based architectures such as MedFormer exhibit stronger cross-domain stability and retain meaningful performance under severe shifts, achieving a Dice score of 0.323 in Session 4 with JASCL (Table 8). We recommend transformer backbones when facing substantial domain heterogeneity. **(2) Leveraging more unlabeled data -** we further study the effect of additional publicly available unlabeled CT data (TotalSegmentator, AMOS, BCV, MOTS) under a U-Net backbone.

The steady improvement demonstrates the strength of the prototype anchored supervision (PAS) mechanism in utilizing unlabeled data under domain shift.

### J.2    Noisy Unlabeled Data

In real clinical scenarios, medical scans often suffer from noise, blur, and contrast degradation. To assess robustness under such realistic artifacts, we evaluate JASCL under moderate and heavy noise settings applied to the *unlabeled* data.

**Boundary-Aware Noise Generation:** We employ a boundary-aware degradation module. Sobel filtering extracts structural boundaries, which are converted into a soft mask $B(x, y, z) \in [0, 1]$. Blur, contrast shifts, and additive Gaussian noise are then applied primarily to high-boundary regions (Table 22), while homogeneous regions remain minimally affected. This degradation is applied to 70% of unlabeled samples to mimic natural variability.

JASCL shows strong resilience as heavy noise reduces performance by only **0.002** compared to the moderate setting. This indicates that the method is **highly robust to substantial degradations** in unlabeled images. Nevertheless, further improving robustness to noise in continual semi-supervised segmentation remains an important direction for future work.

### J.3    3D Medical Domain

**(i) Organ size and few-shot:** Segmentation performance in 3D medical datasets strongly depends on organ size. Small structures (e.g., adrenal glands, colon tumors, BraTS tumor subregions)[10] are known to be difficult to segment, and this

---

[10] https://arxiv.org/pdf/2501.09138

difficulty is amplified in few-shot continual learning. **(ii) Base model accuracy affects retention:** Retention in continual learning depends on the initial accuracy of the base model. Methods starting near Dice 0.9 may retain high absolute performance after several sessions, whereas methods starting near Dice 0.6-0.7 may naturally yield lower absolute values even with good retention. Most JASCL medical benchmarks fall within the 0.6-0.7 range, which explains the absolute performance levels observed across sessions. **(iii) Long multi-session continual learning is inherently harder:** Existing medical continual segmentation works typically perform only 1-3 incremental steps (CLIP-CT), whereas JASCL stress-tests up to six incremental sessions across multiple modalities and classes. More sessions naturally lead to lower absolute accuracy, even when relative improvements and stability are strong.

### J.4 Autonomous Driving

**(i) Dataset complexity and backbone capacity:** Recent foundation models[11] achieve 50-60% IoU on BDD and Cityscapes even with a powerful ViT-H ($\sim$630M parameters) backbone. JASCL uses substantially smaller backbones (ViT-B, $\sim$90M parameters). The challenging nature of the dataset inherently restricts the maximum IoU that models can achieve. **(ii) Continual segmentation performance is generally low:** State-of-the-art continual semantic segmentation typically achieves only 20-50% IoU under realistic multi-domain settings[12]. JASCL additionally operates in a few-shot regime, where the scarcity of labels further reduces absolute accuracy.

### J.5 2D Robotic Surgery

This domain is exceptionally challenging due to rapid appearance changes in organs, tools, lighting, and camera viewpoints. As shown in CAT, existing methods achieve only 5-30% IoU for incremental classes in continual setups, underscoring the inherent difficulty of the task.

## K   Additional ablations

We evaluated JASCL across class-incremental learning, domain incremental learning, mixed class and domain shifts, and semi-supervised CIL settings with scarce labeled data, spanning 2D robotic surgery, 3D medical segmentation, and autonomous driving. Across all tasks, specialized methods, whether designed for CIL (CAT, BCM), DIL (SAM, CLIP), mixed class and domain shifts (MDIL), or semi-supervised CIL (UaD-CE), exhibit severe performance degradation over incremental sessions. In contrast, JASCL consistently maintains strong performance across all benchmarks, demonstrating robustness to class shifts, domain shifts, scarce labels, and unlabeled data, both individually and in combination.

### K.1   2D robotic surgery task

To study the effect of each setup, we performed class-incremental learning, domain incremental learning, and their combination (CDIL) under data-scarce conditions. We evaluated widely used backbones such as Medformer (a Transformer-based model) and U-Net for semantic segmentation.

This task uses two domains, CholecSeg8k and m2caiseg, and varies both the class set and the domain across sessions. For class-incremental learning, we vary the classes within CholecSeg8k. For domain incremental learning, we shift the domain from CholecSeg8k (Session 0) to m2caiseg (Session 1). For combined class and domain incremental learning, we vary both the domain and the class set when moving from CholecSeg8k (Session 0) to m2caiseg (Session 1).

In Table 24, we observe that all models are affected, and their performance drops significantly across these settings. Interestingly, transformer-based models show poorer performance compared to simpler encoder-decoder models such as U-Net.

### K.2   Autonomous driving

To study the effect of each setup, we performed class-incremental learning, domain incremental learning, and their combination under data scarce conditions. We used SAM as the backbone. For class-incremental learning, we varied the classes within BDD (Berkeley Deep Drive). For domain incremental learning, we shifted the domain from BDD (Session 0) to IDD (Indian Driving Dataset, Session 1). For combined class and domain incremental learning, we varied both the domain

---

[11]https://arxiv.org/pdf/2510.27047
[12]https://ieeexplore.ieee.org/stamp/stamp.jsp?arnumber=10521870

and the class set when moving from BDD (Session 0) to IDD (Session 1).

Even heavily pre-trained models such as SAM show a noticeable drop in performance (Table 25) when evaluated under class-incremental and domain incremental learning setups.

**The above settings clearly show that models specifically designed for class-incremental learning, domain incremental learning, data scarcity, or the use of unlabeled data still fail to perform reliably. In contrast, the JASCL framework is able to handle each of these setups individually, as well as the more complex continual scenarios captured in the JASCL benchmarks.**

## L  Practical guidelines for reliable performance

Below, we provide key recommendations for effectively deploying the JASCL framework in multi-constrained continual semantic segmentation settings.

### L.1  Role of Unlabeled Data

Domain-relevant unlabeled data plays a crucial role in JASCL. As observed in Session 4 when BraTS (MRI modality) was introduced, unlabeled data substantially augments novel few-shot classes and helps the model retain modality- and domain-specific structure when supervision is scarce. Since unlabeled data is far cheaper to acquire than labeled samples, it provides large practical gains ($0.025 \rightarrow 0.058 \rightarrow 0.197$). We therefore recommend using unlabeled data whenever possible.

### L.2  Role of $\varepsilon$

The hyperparameter $\varepsilon$ can be selected using a small validation set based on the deployment scenario. Preliminary observations show that appropriate choices of $\varepsilon$ consistently improve performance.

### L.3  Total Drop (%)

Absolute Dice scores at the final session may recover despite catastrophic collapses during intermediate sessions, particularly in real-world incremental settings like medical imaging (scanner/protocol drift), autonomous driving (weather/city changes), robotics (task-by-task updates), and continual deployment scenarios. **Total Drop (%)** captures the full forgetting trajectory and provides a more reliable measure of stability over time.

$$\text{Total Drop} = \left( \frac{\sum_i \max(0, S_i - S_{i+1})}{S_0} \right) \times 100.$$

where $S_0$ is the base session and $S_i$ is the incremental session $i$. Table 26 shows that JASCL is the most reliable method, with the lowest Total Drop.

### L.4  Model Choice Under Budget Constraints

When resources permit, pre-trained models combined with JASCL offer the best performance and robustness. Under moderate budgets, U-Net or transformer backbones are effective choices. Transformers show greater resilience to severe domain shifts, whereas U-Nets remain strong in standard settings. For lightweight backbones, JASCL with unlabeled data provides the best trade-off, since unlabeled data is easy to obtain and significantly improves performance. This is especially useful when models must be updated frequently or across many continual sessions.

### L.5  Extreme Data-Scarce Settings

JASCL performs reliably even under extreme label scarcity. For Med JASCL-Disjoint, Session 1 performance with $K = 4$ and $K = 3$ labeled samples reaches 0.4343 and 0.4141, respectively, remaining close to the $K = 5$ performance of 0.460 and consistently outperforming baselines. This makes JASCL highly practical in settings where annotation is severely limited.

# M   Additional results

We report class-wise results for JASCL and some baselines. We show how class performance varies across sessions. We report results of JASCL without gradient adaptive stabilization in Table 34 and without prototype anchored supervision in Table 36. Table 32 and Table 44 highlight that the performance of pre-trained models deteriorates in the proposed multi-constraint benchmark. Figure 6 shows the performance of *seen* and *new* classes in different settings.

*Table 12.* Dataset splits for the Med JASCL-Disjoint benchmark. Session 0 (base) uses the **TotalSegmentator (TS)** domain; Session 1 uses **AMOS**; Session 2 uses **BCV**; Session 3 uses **MOTS**; Session 4 uses **BraTS**; and Session 5 uses **VerSe**. All incremental-session training sets follow a **5-shot** protocol.

| Class | Source | Train | Val | Test |
|---|---|---|---|---|
| Sacrum - 1 | TS | 136 | 27 | 43 |
| Stomach - 2 | TS | 124 | 29 | 53 |
| Lung upper lobe left - 3 | TS | 188 | 41 | 92 |
| Lung lower lobe left - 4 | TS | 179 | 39 | 83 |
| Brain - 5 | TS | 125 | 35 | 79 |
| Atrium (left) - 6 | TS | 127 | 25 | 52 |
| Ventricle (left) - 7 | TS | 123 | 24 | 50 |
| Pulmonary artery - 8 | TS | 116 | 17 | 43 |
| Aorta - 9 | TS | 200 | 39 | 85 |
| Gallbladder - 10 | TS | 120 | 27 | 47 |
| Trachea - 11 | TS | 157 | 30 | 68 |
| Rib left1 - 12 | TS | 156 | 28 | 66 |
| Rib right1 - 13 | TS | 156 | 29 | 67 |
| Rib left2 - 14 | TS | 156 | 29 | 67 |
| Rib right2 - 15 | TS | 155 | 29 | 66 |
| Pancreas - 16 | Amos | 120 | 30 | 60 |
| Duodenum - 17 | Amos | 120 | 30 | 60 |
| Bladder - 18 | Amos | 99 | 26 | 47 |
| Prostate/uterus - 19 | Amos | 96 | 26 | 47 |
| Postcava - 20 | Amos | 120 | 30 | 60 |
| Spleen - 21 | BCV | 18 | 4 | 8 |
| Liver - 22 | BCV | 18 | 4 | 8 |
| Left kidney - 23 | BCV | 18 | 4 | 8 |
| Right kidney - 24 | BCV | 18 | 4 | 8 |
| Left adrenal - 25 | BCV | 18 | 4 | 8 |
| Right adrenal - 26 | BCV | 18 | 4 | 8 |
| Hepatic Vessel - 27,28 | MOTS | 7 | 4 | 4 |
| Hepatic Vessel Tumor | MOTS | 7 | 4 | 4 |
| Colon Tumor - 29 | MOTS | 7 | 4 | 4 |
| Lung Tumor - 30 | MOTS | 7 | 4 | 4 |
| NCR — label 1 | BraTS | 40 | 4 | 8 |
| ED — label 2 | BraTS | 40 | 4 | 8 |
| ET — label 4 | BraTS | 37 | 4 | 8 |
| C3(3) | VerSe | 12 | 5 | 6 |
| C4(4) | VerSe | 12 | 5 | 6 |
| T1(8) | VerSe | 31 | 5 | 8 |
| T2(9) | VerSe | 26 | 5 | 7 |

*Table 13.* Dataset splits for the Med JASCL-Mixed benchmark, where both classes and domains may repeat across sessions. Session 0 (base) uses the **AMOS** domain. Session 1 uses **BCV** and **MOTS**. Session 2 uses **TS** and **AMOS**, with three classes repeating (*Right kidney*, *Left kidney*, *Stomach*) from the TS domain. Session 3 uses **MOTS** and **TS**, and Session 4 uses **BraTS** and **VerSe**. All incremental sessions follow a **5-shot** training protocol.

| Class | Source | Train | Val | Test |
|---|---|---|---|---|
| Spleen - 1 | Amos | 203 | 66 | 68 |
| Right kidney - 2 | Amos | 234 | 77 | 77 |
| Left kidney - 3 | Amos | 233 | 78 | 78 |
| Gall bladder - 4 | Amos | 190 | 61 | 67 |
| Esophagus - 5 | Amos | 204 | 67 | 68 |
| Prostate/uterus - 6 | Amos | 167 | 55 | 52 |
| Stomach - 7 | Amos | 233 | 77 | 77 |
| Arota / aorta - 8 | Amos | 204 | 68 | 68 |
| Duodenum - 9 | Amos | 203 | 68 | 68 |
| Postcava - 10 | Amos | 204 | 68 | 68 |
| Pancreas - 11 | BCV | 18 | 6 | 6 |
| Inferior vena cava - 12 | BCV | 18 | 6 | 6 |
| Portal vein and splenic vein - 13 | BCV | 18 | 6 | 6 |
| Left adrenal - 14 | BCV | 18 | 6 | 6 |
| Right adrenal - 15 | BCV | 18 | 6 | 6 |
| Liver - 16 | BCV | 18 | 6 | 6 |
| Hepatic Vessel - 17 | MOTS | 7 | 4 | 4 |
| Hepatic Vessel Tumor - 18 | MOTS | 7 | 4 | 4 |
| Small bowel - 19 | TS | 60 | 20 | 20 |
| Hip_left - 20 | TS | 60 | 20 | 20 |
| Hip_right - 21 | TS | 60 | 20 | 20 |
| Lung upper lobe left - 22 | TS | 60 | 20 | 20 |
| Lung lower lobe left - 23 | TS | 60 | 20 | 20 |
| Bladder - 24 | Amos | 9 | 4 | 4 |
| Colon Tumor - 25 | MOTS | 7 | 4 | 4 |
| Lung Tumor - 26 | MOTS | 7 | 4 | 4 |
| Femur_left - 27 | TS | 18 | 6 | 6 |
| Femur_right - 28 | TS | 18 | 6 | 6 |
| T1 | BraTS | 40 | 4 | 8 |
| T2 | BraTS | 40 | 4 | 8 |
| Flair | BraTS | 37 | 4 | 8 |
| C3(3) | VerSe | 12 | 5 | 6 |
| C4(4) | VerSe | 12 | 5 | 6 |
| T1(8) | VerSe | 31 | 5 | 8 |
| T2(9) | VerSe | 26 | 5 | 7 |

*Table 14.* Unlabeled sample counts per session in Med Semi-Supervised-JASCL benchmark.

| Session | Count |
|---|---|
| Session 1 - Amos | 25 |
| Session 2 - BCV | 30 |
| Session 3 - MOTS | 8 |
| Session 4 - BraTS | 15 |
| Session 5 - VerSe | 20 |

*Table 15.* Dataset splits for the Natural-JASCL benchmark. All incremental sessions follow a **10-shot** training protocol, with 3 validation samples per class.

| Class | Source | Train | Val | Test |
|---|---|---|---|---|
| road - 0 | BDD - Session 0 | 4691 | 1184 | 847 |
| sidewalk - 1 | BDD - Session 0 | 3173 | 796 | 612 |
| building - 2 | BDD - Session 0 | 4270 | 1070 | 758 |
| wall - 3 | BDD - Session 0 | 772 | 189 | 111 |
| fence - 4 | BDD - Session 0 | 1494 | 360 | 206 |
| traffic light - 5 | BDD - Session 0 | 2187 | 556 | 443 |
| traffic sign - 6 | BDD - Session 0 | 3625 | 919 | 660 |
| person - 7 | BDD - Session 0 | 1562 | 393 | 305 |
| car - 8 | BDD - Session 0 | 4738 | 1180 | 833 |
| truck - 9 | BDD - Session 0 | 1451 | 389 | 283 |
| Bridge/tunnel - 10 | IDD - Session 1 | 10 | 3 | 54 |
| Parking - 11 | IDD - Session 1 | 10 | 3 | 173 |
| rail track - 12 | IDD - Session 1 | 10 | 3 | 155 |
| Autorickshaw - 13 | IDD - Session 1 | 10 | 3 | 139 |
| pole - 14 | IDD - Session 1 | 10 | 3 | 220 |
| Bus - 15 (from BDD) | BDD - Session 2 | 10 | 3 | 58 |
| Sky - 16 (from BDD) | BDD - Session 2 | 10 | 3 | 139 |
| bicycle - 17 (from BDD) | BDD - Session 2 | 10 | 3 | 45 |
| person - 7 (repeat from IDD) - 18 | IDD - Session 2 | 10 | 3 | 82 |
| car - 8 (repeat from IDD) - 19 | IDD - Session 2 | 10 | 3 | 91 |

*Table 16.* Dataset splits for the Semi-Supervised Natural-JASCL benchmark, which includes **400 unlabeled samples per class**. All incremental sessions use a **10-shot** training protocol and 3 validation samples per class.

| Class | Source | Train | Val | Test |
|---|---|---|---|---|
| road - 0 | BDD : Session 0 | 4691 | 1184 | 847 |
| sidewalk - 1 | BDD : Session 0 | 3173 | 796 | 612 |
| building - 2 | BDD : Session 0 | 4270 | 1070 | 758 |
| wall - 3 | BDD : Session 0 | 772 | 189 | 111 |
| fence - 4 | BDD : Session 0 | 1494 | 360 | 206 |
| traffic light - 5 | BDD : Session 0 | 2187 | 556 | 443 |
| traffic sign - 6 | BDD : Session 0 | 3625 | 919 | 660 |
| person - 7 | BDD : Session 0 | 1562 | 393 | 305 |
| car - 8 | BDD : Session 0 | 4738 | 1180 | 833 |
| truck - 9 | BDD : Session 0 | 1451 | 389 | 283 |
| parking - 10 | Cityscapes : Session 1 | 10 | 3 | 111 |
| vegetation - 11 | Cityscapes : Session 1 | 10 | 3 | 486 |
| rail track - 12 | IDD : Session 2 | 10 | 3 | 214 |
| sky - 13 | IDD : Session 2 | 10 | 3 | 300 |
| motorcycle - 14 | IDD : Session 3 | 10 | 3 | 609 |
| autorickshaw - 15 | IDD : Session 3 | 10 | 3 | 361 |
| bridge/tunnel - 16 | IDD : Session 3 | 10 | 3 | 65 |

*Table 17.* Dataset splits for 2D robotic surgery benchmark.

| Category | Class | Source | Train | Val | Test |
|---|---|---|---|---|---|
| misc | Background | CholecSeg8k | 3246 | 713 | 1304 |
| organ | Abdominal Wall | CholecSeg8k | 2774 | 639 | 1158 |
| organ | Liver | CholecSeg8k | 3246 | 713 | 1304 |
| organ | Gastrointestinal Tract | CholecSeg8k | 1832 | 399 | 715 |
| organ | Fat | CholecSeg8k | 2991 | 662 | 1199 |
| organ | Gallbladder | CholecSeg8k | 2810 | 594 | 1124 |
| organ | Connective Tissue | CholecSeg8k | 1116 | 154 | 330 |
| organ | Upper wall | m2caiseg | 50 | 50 | 142 |
| organ | Intestine | m2caiseg | 50 | 4 | 10 |
| instrument | Grasper | m2caiseg | 50 | 50 | 132 |
| instrument | Clipper | m2caiseg | 50 | 15 | 39 |

*Table 18.* Performance of JASCL on COCO-O detection benchmark with one incremental session on three few-shot novel classes. We compared JASCL without Gradient Adaptive Stabilization (GAS) and Prototype Anchored Supervision (PAS).

| Method | Session 1 (mAP) |
|---|---|
| RCNN (Vanilla with only unlabeled data) | 0.000 |
| JASCL | 0.063 |
| JASCL w/o GAS | 0.004 |
| JASCL w/o PAS | 0.051 |

*Table 19.* Configuration of JASCL in different settings.

| Method | Base Frozen (Yes/No) | Incremental Sessions Frozen (which layers) |
|---|---|---|
| Med JASCL-Disjoint (U-Net) | No | Yes (All but last) |
| Med JASCL-Mixed (MedFormer) | No | Yes (All but last) |
| Med JASCL-Mixed (SwinUNetr) | No | Yes (All but last) |
| Natural-JASCL (DeepLab) | No | No |
| Semi-Supervised Natural-JASCL (Deeplab) | No | No |
| Med Semi-Supervised-JASCL (U-Net) | No | Yes (All but last) |
| Med Semi-Supervised-JASCL (MedFormer) | No | Yes (All but last) |
| Natural-JASCL (SAM) | No | Yes (All but last) |
| 2D robotic surgery (U-Net) | No | No |
| 2D robotic surgery (MedFormer) | No | No |

*Table 20.* Performance comparison on Med JASCL-Disjoint benchmark showing that GAS provides substantial gains over existing regularization-based methods for continual learning.

| Method | Session 0 | Session 1 | Session 2 |
|---|---|---|---|
| Dropout / Weight Decay | 0.700 | 0.191 | – |
| Variational Dropout | 0.700 | 0.260 | – |
| Adagrad | 0.700 | 0.024 | – |
| UCL | 0.700 | 0.430 | 0.325 |
| C-Flat | 0.700 | 0.174 | 0.030 |
| STAR | 0.700 | 0.050 | 0.020 |
| JASCL (GAS) | **0.736** | **0.460** | **0.398** |

*Table 21.* Impact of additional unlabeled CT data on Session 4 performance.

| Model (U-Net backbone) | Session 4 |
|---|---|
| JASCL (no unlabeled data) | 0.025 |
| JASCL (15 unlabeled samples) | 0.058 |
| JASCL (42 unlabeled samples) | 0.197 |

*Table 22.* Noise schedules used for unlabeled image degradation.

| Operation | Randomness | Moderate | Heavy |
|-----------|-----------|----------|-------|
| Blur | Uniform | $\sigma \in [0.5, 1.5]$ | $\sigma \in [1.0, 2.5]$ |
| Contrast | Uniform | $1.0 \pm 0.15$ | $1.0 \pm 0.30$ |
| Additive Noise | Gaussian | $\sigma = 0.02$ | $\sigma = 0.05$ |

*Table 23.* Performance of JASCL under noise-corrupted unlabeled data.

| Method | Session 1 |
|--------|-----------|
| JASCL (no noise) | 0.554 |
| JASCL (moderate noise) | 0.477 |
| JASCL (heavy noise) | 0.475 |
| UaD-CE | 0.082 |

*Table 24.* Performance of class-incremental learning (CIL), domain incremental learning (DIL), and their combination (CDIL) under data-scarce settings. S denotes the session; for example, S0 corresponds to Session 0.

| Model | CIL S0 | CIL S1 | CIL S2 | DIL S0 | DIL S1 | CDIL S0 | CDIL S1 | CDIL S2 |
|-------|--------|--------|--------|--------|--------|---------|---------|---------|
| U-Net Vanilla | 0.770 | 0.003 | 0.000 | 0.770 | 0.223 | 0.770 | 0.033 | 0.007 |
| Medformer Vanilla | 0.762 | 0.026 | 0.002 | 0.762 | 0.166 | 0.762 | 0.011 | 0.002 |
| CAT U-Net | 0.770 | 0.102 | 0.006 | 0.770 | 0.258 | 0.770 | 0.166 | 0.001 |
| JASCL Medformer | 0.762 | 0.166 | 0.144 | 0.762 | 0.279 | 0.762 | 0.169 | 0.156 |
| JASCL U-Net | 0.770 | 0.212 | 0.159 | 0.770 | 0.305 | 0.770 | 0.244 | 0.163 |

*Table 25.* Performance of class-incremental learning (CIL), domain incremental learning (DIL), and their combination (CDIL) under data-scarce settings. S denotes the session; for example, S0 corresponds to Session 0.

| Model | CIL S0 | CIL S1 | DIL S0 | DIL S1 | CDIL S0 | CDIL S1 | CDIL S2 |
|-------|--------|--------|--------|--------|---------|---------|---------|
| SAM Vanilla | 66.0 | 30.49 | 66.0 | 30.43 | 66.0 | 32.6 | 30.81 |
| SAM JASCL | 66.0 | 31.39 | 66.0 | 30.64 | 66.0 | 33.2 | 31.22 |

*Table 26.* Total Drop (%) for all baselines on Med JASCL-Disjoint (lower is better).

| Method | Total Drop (%) |
|---|---|
| PIFS | 88.9 |
| NC-FSCIL | 80.4 |
| CLIP-CT | 70.3 |
| MiB | 86.3 |
| MDIL | 87.5 |
| C-FSCIL | 62.3 |
| SoftNet | 82.2 |
| GAPS | 63.8 |
| FSCIL-SS | 87.3 |
| Subspace | 84.4 |
| Gen-Replay | 89.1 |
| FeCAM | 94.0 |
| FACT | 92.2 |
| MAML | 99.9 |
| MAML + Reg. | 99.9 |
| MTL | 88.7 |
| UnSupCL | 94.4 |
| SupCL | 94.0 |
| UnSupCL-HNM | 95.0 |
| C-Flat | 95.7 |
| STAR | 96.9 |
| Saving100x | 92.6 |
| YoooP | 96.9 |
| UCB | 81.6 |
| BCM | 100.0 |
| Adapt_replay | 96.4 |
| UCL | 53.6 |
| **JASCL (U-Net)** | **45.9** |

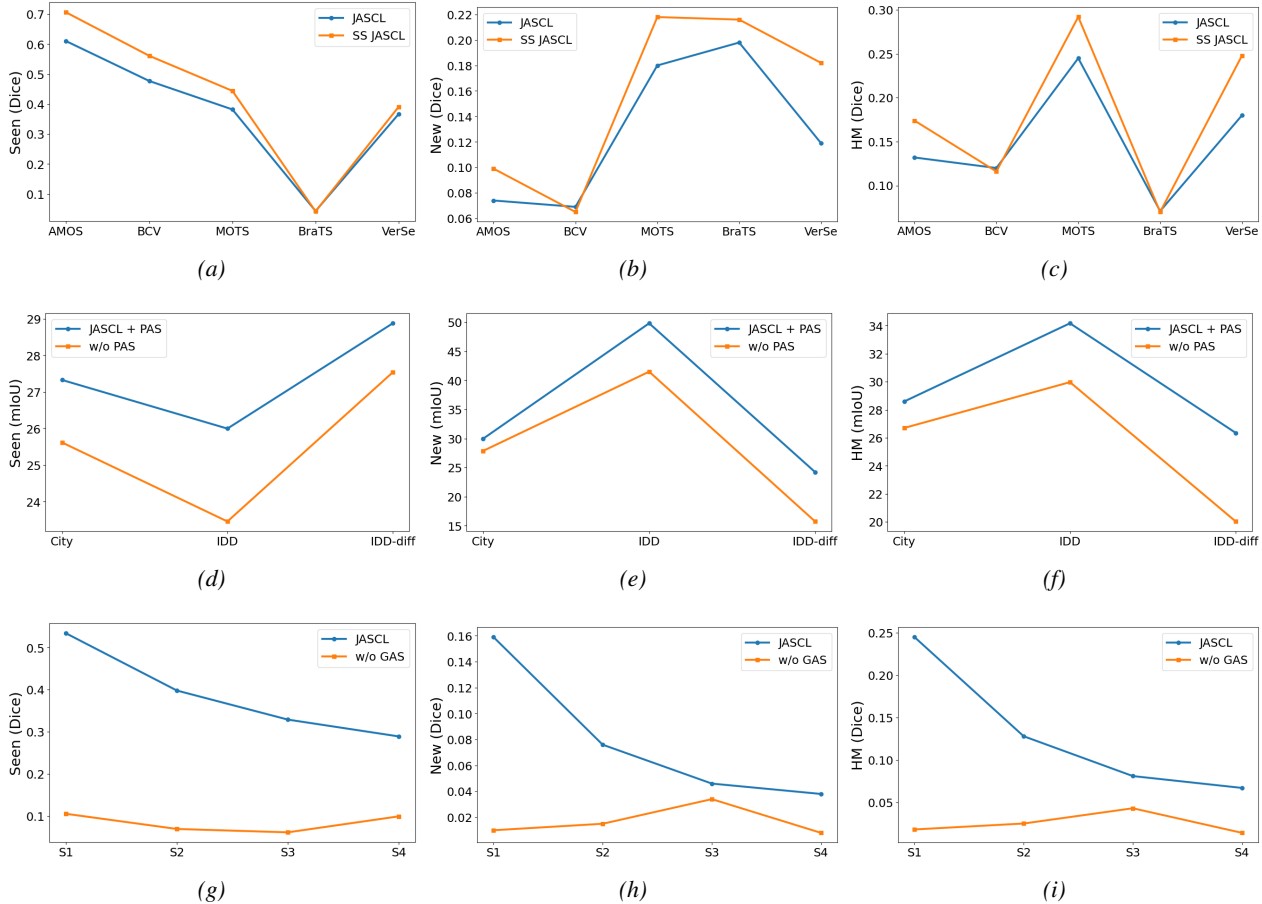

*Figure 6.* **(a-c)** Performance of JASCL on the Med JASCL-Disjoint benchmark and its variant, Med Semi-Supervised-JASCL (which includes additional unlabeled data), evaluated across incremental sessions (TS (Base) → AMOS → BCV → MOTS → BraTS → VerSe). SS refers to Semi-Supervised. SS JASCL denotes the performance of JASCL on the Med Semi-Supervised-JASCL benchmark. The reported results confirm that unlabeled data helps to boost the performance. **(d-f)** Performance of JASCL on the Semi-Supervised Natural-JASCL benchmark (which includes additional unlabeled data), evaluated across incremental sessions (BDD100K (Base) → Cityscapes → IDD → IDD-diff (with different classes from the previous session)). The results show that prototype anchored supervision (PAS) enhances the performance of JASCL compared to JASCL without prototype anchored supervision (w/o PAS). **(g-i)** Performance of JASCL (MedFormer) on the Med JASCL-Mixed benchmark evaluated across incremental sessions (Session 0 (Base) → Session 1 → Session 2 → Session 3 → Session 4) The results show that gradient adaptive stabilization (GAS) enhances the performance of JASCL compared to JASCL without gradient adaptive stabilization (w/o GAS). Seen refers to the average performance on classes the model has encountered in previous sessions, while New refers to the average performance on classes introduced in the current session. HM refers to Harmonic Mean.

*Table 27.* Class-wise performance of JASCL on Med JASCL-Disjoint with U-Net.

| Class | Session 0 | Session 1 | Session 2 | Session 3 | Session 4 | Session 5 |
|---|---|---|---|---|---|---|
| class_0 | 0.979 | 0.962 | 0.848 | 0.764 | 0.944 | 0.672 |
| class_1 | 0.792 | 0.614 | 0.611 | 0.553 | 0.002 | 0.791 |
| class_2 | 0.613 | 0.209 | 0.341 | 0.454 | 0.002 | 0.541 |
| class_3 | 0.856 | 0.762 | 0.739 | 0.717 | 0.003 | 0.834 |
| class_4 | 0.756 | 0.636 | 0.623 | 0.646 | 0.003 | 0.736 |
| class_5 | 0.910 | 0.769 | 0.730 | 0.601 | 0.458 | 0.877 |
| class_6 | 0.653 | 0.611 | 0.596 | 0.557 | 0.127 | 0.622 |
| class_7 | 0.704 | 0.551 | 0.559 | 0.568 | 0.029 | 0.656 |
| class_8 | 0.875 | 0.835 | 0.802 | 0.733 | 0.182 | 0.810 |
| class_9 | 0.768 | 0.719 | 0.711 | 0.687 | 0.048 | 0.753 |
| class_10 | 0.610 | 0.272 | 0.480 | 0.542 | 0.002 | 0.444 |
| class_11 | 0.803 | 0.711 | 0.687 | 0.683 | 0.030 | 0.825 |
| class_12 | 0.824 | 0.674 | 0.681 | 0.650 | 0.082 | 0.732 |
| class_13 | 0.729 | 0.638 | 0.592 | 0.579 | 0.069 | 0.652 |
| class_14 | 0.732 | 0.635 | 0.595 | 0.397 | 0.046 | 0.689 |
| class_15 | 0.705 | 0.431 | 0.500 | 0.539 | 0.019 | 0.654 |
| class_16 | – | 0.129 | 0.142 | 0.134 | 0.040 | 0.122 |
| class_17 | – | 0.043 | 0.058 | 0.062 | 0.017 | 0.084 |
| class_18 | – | 0.050 | 0.068 | 0.071 | 0.021 | 0.072 |
| class_19 | – | 0.053 | 0.017 | 0.012 | 0.016 | 0.015 |
| class_20 | – | 0.096 | 0.073 | 0.070 | 0.017 | 0.062 |
| class_21 | – | – | 0.032 | 0.432 | 0.031 | 0.165 |
| class_22 | – | – | 0.287 | 0.125 | 0.024 | 0.111 |
| class_23 | – | – | 0.073 | 0.093 | 0.003 | 0.018 |
| class_24 | – | – | 0.037 | 0.031 | 0.001 | 0.042 |
| class_25 | – | – | 0.004 | 0.008 | 0.015 | 0.010 |
| class_26 | – | – | 0.000 | 0.003 | 0.002 | 0.001 |
| class_27 | – | – | – | 0.075 | 0.000 | 0.087 |
| class_28 | – | – | – | 0.031 | 0.000 | 0.032 |
| class_29 | – | – | – | 0.338 | 0.000 | 0.292 |
| class_30 | – | – | – | 0.277 | 0.014 | 0.043 |
| class_31 | – | – | – | – | 0.109 | 0.060 |
| class_32 | – | – | – | – | 0.185 | 0.008 |
| class_33 | – | – | – | – | 0.299 | 0.158 |
| class_34 | – | – | – | – | – | 0.056 |
| class_35 | – | – | – | – | – | 0.133 |
| class_36 | – | – | – | – | – | 0.222 |
| class_37 | – | – | – | – | – | 0.066 |
| **Average** | 0.736 | 0.459 | 0.398 | 0.329 | 0.025 | 0.324 |

*Table 28.* Class-wise performance of JASCL on Med JASCL-Mixed with MedFormer.

| Class | Session 0 | Session 1 | Session 2 | Session 3 | Session 4 |
|---|---|---|---|---|---|
| class_0 | 0.973 | 0.892 | 0.916 | 0.855 | 0.842 |
| class_1 | 0.695 | 0.708 | 0.641 | 0.629 | 0.690 |
| class_2 | 0.705 | 0.524 | 0.606 | 0.682 | 0.653 |
| class_3 | 0.608 | 0.527 | 0.585 | 0.589 | 0.411 |
| class_4 | 0.367 | 0.263 | 0.300 | 0.318 | 0.300 |
| class_5 | 0.646 | 0.530 | 0.540 | 0.611 | 0.615 |
| class_6 | 0.533 | 0.341 | 0.437 | 0.462 | 0.442 |
| class_7 | 0.562 | 0.546 | 0.548 | 0.526 | 0.525 |
| class_8 | 0.870 | 0.861 | 0.693 | 0.867 | 0.734 |
| class_9 | 0.570 | 0.449 | 0.456 | 0.484 | 0.371 |
| class_10 | 0.709 | 0.592 | 0.483 | 0.514 | 0.512 |
| class_11 | – | 0.208 | 0.225 | 0.222 | 0.102 |
| class_12 | – | 0.466 | 0.436 | 0.248 | 0.404 |
| class_13 | – | 0.025 | 0.196 | 0.205 | 0.200 |
| class_14 | – | 0.002 | 0.042 | 0.050 | 0.046 |
| class_15 | – | 0.002 | 0.067 | 0.072 | 0.076 |
| class_16 | – | 0.520 | 0.294 | 0.232 | 0.224 |
| class_17 | – | 0.023 | 0.097 | 0.101 | 0.099 |
| class_18 | – | 0.029 | 0.087 | 0.073 | 0.076 |
| class_19 | – | – | 0.159 | 0.231 | 0.236 |
| class_20 | – | – | 0.084 | 0.164 | 0.085 |
| class_21 | – | – | 0.025 | 0.155 | 0.167 |
| class_22 | – | – | 0.045 | 0.073 | 0.069 |
| class_23 | – | – | 0.120 | 0.173 | 0.184 |
| class_24 | – | – | 0.018 | 0.120 | 0.115 |
| class_25 | – | – | – | 0.099 | 0.079 |
| class_26 | – | – | – | 0.067 | 0.184 |
| class_27 | – | – | – | 0.013 | 0.036 |
| class_28 | – | – | – | 0.008 | 0.071 |
| class_29 | – | – | – | – | 0.018 |
| class_30 | – | – | – | – | 0.101 |
| class_31 | – | – | – | – | 0.086 |
| class_32 | – | – | – | – | 0.025 |
| class_33 | – | – | – | – | 0.014 |
| class_34 | – | – | – | – | 0.019 |
| class_35 | – | – | – | – | 0.020 |
| **Average** | 0.626 | 0.367 | 0.299 | 0.285 | 0.228 |

*Table 29.* Performance of JASCL on 2D robotic surgery.

| Model | Session 0 | Session 1 | Session 2 |
|---|---|---|---|
| UNet Vanilla | 0.770 | 0.033 | 0.007 |
| Medformer Vanilla | 0.762 | 0.011 | 0.002 |
| CAT (UNet) | 0.770 | 0.166 | 0.001 |
| JASCL - Medformer | 0.762 | 0.169 | 0.156 |
| JASCL - UNet | 0.770 | 0.244 | 0.163 |

*Table 30.* Class-wise performance of JASCL on Med JASCL-Mixed with SwinUNetr.

| Class | Session 0 | Session 1 | Session 2 | Session 3 | Session 4 |
|---|---|---|---|---|---|
| class_0 | 0.972 | 0.943 | 0.961 | 0.938 | 0.960 |
| class_1 | 0.802 | 0.805 | 0.754 | 0.749 | 0.788 |
| class_2 | 0.415 | 0.199 | 0.391 | 0.388 | 0.399 |
| class_3 | 0.551 | 0.576 | 0.506 | 0.526 | 0.560 |
| class_4 | 0.301 | 0.213 | 0.163 | 0.145 | 0.190 |
| class_5 | 0.657 | 0.567 | 0.618 | 0.548 | 0.593 |
| class_6 | 0.574 | 0.428 | 0.465 | 0.472 | 0.458 |
| class_7 | 0.592 | 0.306 | 0.531 | 0.549 | 0.537 |
| class_8 | 0.847 | 0.497 | 0.729 | 0.844 | 0.845 |
| class_9 | 0.524 | 0.363 | 0.463 | 0.471 | 0.444 |
| class_10 | 0.694 | 0.089 | 0.352 | 0.377 | 0.222 |
| class_11 | – | 0.339 | 0.124 | 0.159 | 0.192 |
| class_12 | – | 0.754 | 0.709 | 0.734 | 0.749 |
| class_13 | – | 0.044 | 0.048 | 0.032 | 0.028 |
| class_14 | – | 0.000 | 0.028 | 0.025 | 0.021 |
| class_15 | – | 0.000 | 0.025 | 0.030 | 0.016 |
| class_16 | – | 0.538 | 0.114 | 0.073 | 0.065 |
| class_17 | – | 0.011 | 0.058 | 0.045 | 0.048 |
| class_18 | – | 0.009 | 0.014 | 0.006 | 0.003 |
| class_19 | – | – | 0.181 | 0.027 | 0.016 |
| class_20 | – | – | 0.034 | 0.065 | 0.063 |
| class_21 | – | – | 0.082 | 0.094 | 0.088 |
| class_22 | – | – | 0.016 | 0.001 | 0.000 |
| class_23 | – | – | 0.086 | 0.011 | 0.011 |
| class_24 | – | – | 0.112 | 0.128 | 0.116 |
| class_25 | – | – | – | 0.337 | 0.017 |
| class_26 | – | – | – | 0.269 | 0.003 |
| class_27 | – | – | – | 0.000 | 0.004 |
| class_28 | – | – | – | 0.013 | 0.055 |
| class_29 | – | – | – | – | 0.050 |
| class_30 | – | – | – | – | 0.311 |
| class_31 | – | – | – | – | 0.000 |
| class_32 | – | – | – | – | 0.000 |
| class_33 | – | – | – | – | 0.169 |
| class_34 | – | – | – | – | 0.074 |
| **Average** | 0.596 | 0.319 | 0.275 | 0.254 | 0.211 |

*Table 31.* Per-class performance of JASCL across incremental Sessions on Med Semi-Supervised-JASCL benchmark.

| Class | Session 0 | Session 1 | Session 2 | Session 3 | Session 4 | Session 5 |
|---|---|---|---|---|---|---|
| class_0 | 0.985 | 0.969 | 0.842 | 0.763 | 0.936 | 0.709 |
| class_1 | 0.881 | 0.659 | 0.628 | 0.709 | 0.019 | 0.779 |
| class_2 | 0.575 | 0.250 | 0.478 | 0.540 | 0.001 | 0.575 |
| class_3 | 0.852 | 0.883 | 0.886 | 0.874 | 0.025 | 0.896 |
| class_4 | 0.770 | 0.720 | 0.711 | 0.712 | 0.022 | 0.735 |
| class_5 | 0.786 | 0.849 | 0.856 | 0.830 | 0.133 | 0.862 |
| class_6 | 0.733 | 0.715 | 0.713 | 0.715 | 0.081 | 0.695 |
| class_7 | 0.696 | 0.760 | 0.713 | 0.731 | 0.014 | 0.793 |
| class_8 | 0.665 | 0.710 | 0.762 | 0.782 | 0.151 | 0.771 |
| class_9 | 0.799 | 0.828 | 0.797 | 0.812 | 0.118 | 0.791 |
| class_10 | 0.280 | 0.572 | 0.517 | 0.549 | 0.002 | 0.447 |
| class_11 | 0.864 | 0.809 | 0.827 | 0.834 | 0.235 | 0.835 |
| class_12 | 0.697 | 0.760 | 0.748 | 0.729 | 0.119 | 0.828 |
| class_13 | 0.776 | 0.641 | 0.653 | 0.695 | 0.054 | 0.757 |
| class_14 | 0.761 | 0.720 | 0.749 | 0.739 | 0.086 | 0.797 |
| class_15 | 0.769 | 0.712 | 0.671 | 0.511 | 0.026 | 0.804 |
| class_16 | – | 0.178 | 0.172 | 0.171 | 0.051 | 0.165 |
| class_17 | – | 0.060 | 0.087 | 0.112 | 0.008 | 0.105 |
| class_18 | – | 0.109 | 0.089 | 0.077 | 0.004 | 0.106 |
| class_19 | – | 0.074 | 0.048 | 0.068 | 0.000 | 0.042 |
| class_20 | – | 0.074 | 0.112 | 0.104 | 0.007 | 0.092 |
| class_21 | – | – | 0.049 | 0.072 | 0.004 | 0.056 |
| class_22 | – | – | 0.165 | 0.079 | 0.026 | 0.138 |
| class_23 | – | – | 0.091 | 0.060 | 0.001 | 0.084 |
| class_24 | – | – | 0.039 | 0.002 | 0.000 | 0.010 |
| class_25 | – | – | 0.032 | 0.019 | 0.011 | 0.023 |
| class_26 | – | – | 0.014 | 0.023 | 0.000 | 0.054 |
| class_27 | – | – | – | 0.182 | 0.001 | 0.190 |
| class_28 | – | – | – | 0.026 | 0.000 | 0.050 |
| class_29 | – | – | – | 0.366 | 0.001 | 0.168 |
| class_30 | – | – | – | 0.298 | 0.052 | 0.099 |
| class_31 | – | – | – | – | 0.252 | 0.106 |
| class_32 | – | – | – | – | 0.171 | 0.106 |
| class_33 | – | – | – | – | 0.227 | 0.044 |
| class_34 | – | – | – | – | – | 0.160 |
| class_35 | – | – | – | – | – | 0.105 |
| class_36 | – | – | – | – | – | 0.201 |
| class_37 | – | – | – | – | – | 0.286 |
| **Average** | 0.736 | 0.554 | 0.449 | 0.414 | 0.058 | 0.374 |

*Table 32.* Performance of SAM and JASCL on Natural-JASCL benchmark.

| Class | Base | SAM Session 1 | JASCL Session 1 | SAM Session 2 | JASCL Session 2 |
|---|---|---|---|---|---|
| 1 | 95.29 | 86.54 | 88.42 | 88.18 | 88.72 |
| 2 | 64.01 | 35.13 | 34.07 | 40.03 | 39.89 |
| 3 | 92.77 | 89.24 | 89.12 | 82.23 | 84.21 |
| 4 | 46.02 | 21.70 | 24.45 | 35.56 | 39.80 |
| 5 | 57.90 | 16.48 | 17.70 | 42.10 | 40.99 |
| 6 | 48.19 | 16.31 | 18.14 | 17.24 | 15.33 |
| 7 | 53.94 | 33.65 | 37.09 | 39.76 | 37.75 |
| 8 | 60.06 | 28.26 | 39.18 | 56.20 | 57.56 |
| 9 | 90.56 | 87.93 | 87.03 | 84.44 | 80.50 |
| 10 | 51.58 | 45.65 | 38.13 | 35.77 | 37.38 |
| 11 | – | 16.96 | 16.23 | 2.17 | 2.13 |
| 12 | – | 1.67 | 1.23 | 0.03 | 0.02 |
| 13 | – | 3.01 | 3.48 | 0.19 | 0.21 |
| 14 | – | 1.98 | 0.53 | 0.00 | 0.01 |
| 15 | – | 4.42 | 3.72 | 1.01 | 0.98 |
| 16 | – | – | – | 0.65 | 1.13 |
| 17 | – | – | – | 28.94 | 35.42 |
| 18 | – | – | – | 0.00 | 0.01 |
| **Average** | 66.03 | 32.59 | 33.23 | 30.80 | 31.22 |

*Table 33.* Per-class performance of JASCL on Semi-Supervised Natural-JASCL benchmark.

| Class | Session 0 | Session 1 | Session 2 | Session 3 |
|---|---|---|---|---|
| road - 0 | 91.18 | 82.36 | 81.35 | 81.48 |
| sidewalk - 1 | 48.82 | 31.20 | 24.84 | 22.94 |
| building - 2 | 86.42 | 71.03 | 72.53 | 67.41 |
| wall - 3 | 20.84 | 1.95 | 1.39 | 1.14 |
| fence - 4 | 25.28 | 2.81 | 1.41 | 2.49 |
| traffic light - 5 | 38.34 | 10.82 | 0.53 | 1.23 |
| traffic sign - 6 | 31.79 | 4.12 | 0.18 | 0.06 |
| person - 7 | 28.33 | 0.59 | 0.81 | 0.08 |
| car - 8 | 81.19 | 56.30 | 42.89 | 41.92 |
| truck - 9 | 25.40 | 12.12 | 7.94 | 8.96 |
| parking - 10 | – | 5.20 | 7.69 | 6.23 |
| vegetation - 11 | – | 54.76 | 53.41 | 48.49 |
| rail track - 12 | – | – | 12.03 | 13.23 |
| sky - 13 | – | – | 79.62 | 73.67 |
| motorcycle - 14 | – | – | – | 17.53 |
| autorickshaw - 15 | – | – | – | 20.59 |
| bridge/tunnel - 16 | – | – | – | 24.58 |
| **Average** | 47.75 | 27.8 | 27.68 | 25.47 |

*Table 34.* JASCL performance on Med JASCL-Mixed benchmark without gradient adaptive stabilization. The results highlight the importance of the proposed mechanism.

| Class | Session 0 | Session 1 | Session 2 | Session 3 | Session 4 |
|-------|-----------|-----------|-----------|-----------|-----------|
| class_0 | 0.971 | 0.746 | 0.810 | 0.792 | 0.804 |
| class_1 | 0.727 | 0.155 | 0.180 | 0.181 | 0.273 |
| class_2 | 0.738 | 0.103 | 0.120 | 0.119 | 0.194 |
| class_3 | 0.613 | 0.077 | 0.091 | 0.094 | 0.159 |
| class_4 | 0.373 | 0.038 | 0.047 | 0.050 | 0.094 |
| class_5 | 0.665 | 0.099 | 0.133 | 0.137 | 0.244 |
| class_6 | 0.573 | 0.116 | 0.147 | 0.158 | 0.250 |
| class_7 | 0.580 | 0.128 | 0.143 | 0.137 | 0.177 |
| class_8 | 0.859 | 0.133 | 0.162 | 0.170 | 0.289 |
| class_9 | 0.566 | 0.082 | 0.103 | 0.105 | 0.164 |
| class_10 | 0.701 | 0.122 | 0.149 | 0.147 | 0.204 |
| class_11 | – | 0.023 | 0.029 | 0.031 | 0.053 |
| class_12 | – | 0.016 | 0.025 | 0.026 | 0.092 |
| class_13 | – | 0.008 | 0.012 | 0.014 | 0.025 |
| class_14 | – | 0.013 | 0.018 | 0.019 | 0.026 |
| class_15 | – | 0.013 | 0.014 | 0.014 | 0.016 |
| class_16 | – | 0.096 | 0.082 | 0.080 | 0.091 |
| class_17 | – | 0.035 | 0.033 | 0.033 | 0.037 |
| class_18 | – | 0.007 | 0.006 | 0.006 | 0.008 |
| class_19 | – | – | 0.037 | 0.037 | 0.041 |
| class_20 | – | – | 0.009 | 0.010 | 0.015 |
| class_21 | – | – | 0.020 | 0.022 | 0.037 |
| class_22 | – | – | 0.004 | 0.004 | 0.006 |
| class_23 | – | – | 0.026 | 0.027 | 0.030 |
| class_24 | – | – | 0.027 | 0.026 | 0.042 |
| class_25 | – | – | – | 0.018 | 0.025 |
| class_26 | – | – | – | 0.049 | 0.055 |
| class_27 | – | – | – | 0.048 | 0.043 |
| class_28 | – | – | – | 0.009 | 0.014 |
| class_29 | – | – | – | – | 0.006 |
| class_30 | – | – | – | – | 0.022 |
| class_31 | – | – | – | – | 0.008 |
| class_32 | – | – | – | – | 0.016 |
| class_33 | – | – | – | – | 0.049 |
| class_34 | – | – | – | – | 0.009 |
| class_35 | – | – | – | – | 0.016 |
| **Average** | 0.623 | 0.070 | 0.067 | 0.063 | 0.080 |

*Table 35.* Class-wise performance of MedFormer Vanilla on Med JASCL-Mixed benchmark.

| Class | Session 0 | Session 1 | Session 2 | Session 3 | Session 4 |
|---|---|---|---|---|---|
| class_0 | 0.969 | 0.955 | 0.965 | 0.957 | 0.956 |
| class_1 | 0.639 | 0.000 | 0.000 | 0.000 | 0.000 |
| class_2 | 0.660 | 0.000 | 0.117 | 0.000 | 0.000 |
| class_3 | 0.704 | 0.000 | 0.053 | 0.000 | 0.000 |
| class_4 | 0.371 | 0.000 | 0.000 | 0.000 | 0.000 |
| class_5 | 0.596 | 0.000 | 0.000 | 0.000 | 0.000 |
| class_6 | 0.589 | 0.000 | 0.000 | 0.000 | 0.000 |
| class_7 | 0.565 | 0.000 | 0.232 | 0.000 | 0.000 |
| class_8 | 0.835 | 0.000 | 0.000 | 0.000 | 0.000 |
| class_9 | 0.454 | 0.000 | 0.000 | 0.000 | 0.000 |
| class_10 | 0.717 | 0.000 | 0.000 | 0.000 | 0.000 |
| class_11 | – | 0.404 | 0.000 | 0.000 | 0.000 |
| class_12 | – | 0.698 | 0.000 | 0.000 | 0.000 |
| class_13 | – | 0.468 | 0.000 | 0.000 | 0.000 |
| class_14 | – | 0.262 | 0.000 | 0.000 | 0.000 |
| class_15 | – | 0.236 | 0.000 | 0.000 | 0.000 |
| class_16 | – | 0.870 | 0.000 | 0.000 | 0.000 |
| class_17 | – | 0.532 | 0.000 | 0.000 | 0.000 |
| class_18 | – | 0.096 | 0.000 | 0.000 | 0.000 |
| class_19 | – | – | 0.530 | 0.000 | 0.000 |
| class_20 | – | – | 0.580 | 0.000 | 0.000 |
| class_21 | – | – | 0.587 | 0.000 | 0.000 |
| class_22 | – | – | 0.395 | 0.000 | 0.000 |
| class_23 | – | – | 0.473 | 0.000 | 0.000 |
| class_24 | – | – | 0.247 | 0.000 | 0.000 |
| class_25 | – | – | – | 0.720 | 0.000 |
| class_26 | – | – | – | 0.700 | 0.000 |
| class_27 | – | – | – | 0.030 | 0.000 |
| class_28 | – | – | – | 0.000 | 0.000 |
| class_29 | – | – | – | – | 0.277 |
| class_30 | – | – | – | – | 0.401 |
| class_31 | – | – | – | – | 0.586 |
| class_32 | – | – | – | – | 0.247 |
| class_33 | – | – | – | – | 0.279 |
| class_34 | – | – | – | – | 0.376 |
| class_35 | – | – | – | – | 0.153 |
| **Average** | 0.636 | 0.280 | 0.167 | 0.086 | 0.090 |

*Table 36.* JASCL performance on Semi-Supervised Natural-JASCL benchmark without Prototype Anchored Supervision.

| Class | Session 0 | Session 1 | Session 2 | Session 3 |
|---|---|---|---|---|
| road - 0 | 91.18 | 81.79 | 77.50 | 78.68 |
| sidewalk - 1 | 48.82 | 27.03 | 26.18 | 24.36 |
| building - 2 | 86.42 | 70.01 | 66.08 | 62.76 |
| wall - 3 | 20.84 | 3.13 | 0.75 | 0.78 |
| fence - 4 | 25.28 | 4.06 | 0.81 | 0.88 |
| traffic light - 5 | 38.34 | 6.35 | 1.31 | 4.36 |
| traffic sign - 6 | 31.79 | 2.64 | 0.45 | 0.45 |
| person - 7 | 28.33 | 0.63 | 0.02 | 0.03 |
| car - 8 | 81.19 | 49.48 | 43.93 | 42.89 |
| truck - 9 | 25.40 | 11.19 | 6.89 | 9.09 |
| parking - 10 | – | 4.16 | 4.29 | 8.90 |
| vegetation - 11 | – | 51.60 | 53.31 | 60.64 |
| rail track - 12 | – | – | 9.24 | 12.86 |
| sky - 13 | – | – | 73.77 | 78.93 |
| motorcycle - 14 | – | – | – | 14.33 |
| autorickshaw - 15 | – | – | – | 17.82 |
| bridge/tunnel - 16 | – | – | – | 15.08 |
| **Average** | 47.75 | 26.00 | 26.04 | 25.12 |

*Table 37.* Performance of GAPS on Natural-JASCL benchmark.

| Class | Session 0 | Session 1 | Session 2 |
|---|---|---|---|
| road - 0 | 91.18 | 81.01 | 66.32 |
| sidewalk - 1 | 48.82 | 29.91 | 5.24 |
| building - 2 | 86.42 | 77.24 | 62.31 |
| wall - 3 | 20.84 | 0.00 | 0.00 |
| fence - 4 | 25.28 | 3.93 | 0.04 |
| traffic light - 5 | 38.34 | 6.01 | 6.64 |
| traffic sign - 6 | 31.79 | 2.64 | 1.95 |
| person - 7 | 28.33 | 6.48 | 10.33 |
| car - 8 | 81.19 | 68.94 | 45.45 |
| truck - 9 | 25.40 | 1.97 | 0.01 |
| Bridge/tunnel - 10 | – | 7.48 | 8.55 |
| Parking - 11 | – | 22.11 | 20.01 |
| rail track - 12 | – | 14.03 | 6.20 |
| Autorickshaw - 13 | – | 16.80 | 8.02 |
| pole - 14 | – | 12.67 | 9.69 |
| Bus - 15 (from BDD) | – | – | 0.02 |
| Sky - 16 (from BDD) | – | – | 48.76 |
| bicycle - 17 (from BDD) | – | – | 0.81 |
| person - 7 (repeat from IDD) - 18 | – | – | – |
| car - 8 (repeat from IDD) - 19 | – | – | – |
| **Average** | 47.76 | 23.42 | 16.69 |

*Table 38.* Performance of C-FSCIL on Med JASCL-Disjoint benchmark.

| Class | Session 0 | Session 1 | Session 2 |
|---|---|---|---|
| class_0 | 0.982 | 0.715 | 0.606 |
| class_1 | 0.779 | 0.438 | 0.552 |
| class_2 | 0.735 | 0.300 | 0.444 |
| class_3 | 0.877 | 0.725 | 0.563 |
| class_4 | 0.794 | 0.669 | 0.615 |
| class_5 | 0.802 | 0.810 | 0.821 |
| class_6 | 0.742 | 0.438 | 0.471 |
| class_7 | 0.866 | 0.519 | 0.439 |
| class_8 | 0.844 | 0.364 | 0.350 |
| class_9 | 0.869 | 0.476 | 0.521 |
| class_10 | 0.639 | 0.271 | 0.275 |
| class_11 | 0.854 | 0.492 | 0.300 |
| class_12 | 0.780 | 0.095 | 0.145 |
| class_13 | 0.780 | 0.288 | 0.326 |
| class_14 | 0.752 | 0.103 | 0.282 |
| class_15 | 0.688 | 0.139 | 0.247 |
| class_16 | – | 0.112 | 0.053 |
| class_17 | – | 0.067 | 0.068 |
| class_18 | – | 0.188 | 0.231 |
| class_19 | – | 0.088 | 0.067 |
| class_20 | – | 0.092 | 0.079 |
| class_21 | – | – | 0.410 |
| class_22 | – | – | 0.286 |
| class_23 | – | – | 0.054 |
| class_24 | – | – | 0.078 |
| class_25 | – | – | 0.010 |
| class_26 | – | – | 0.037 |
| **Average** | 0.787 | 0.334 | 0.253 |

*Table 39.* Performance of GAPS on Semi-Supervised Natural-JASCL benchmark.

| Class | Session 0 | Session 1 | Session 2 | Session 3 |
|---|---|---|---|---|
| road - 0 | 91.18 | 79.10 | 69.08 | 67.17 |
| sidewalk - 1 | 48.82 | 21.52 | 3.48 | 1.20 |
| building - 2 | 86.42 | 60.16 | 46.16 | 42.60 |
| wall - 3 | 20.84 | 1.31 | 0.49 | 0.33 |
| fence - 4 | 25.28 | 0.00 | 0.00 | 0.09 |
| traffic light - 5 | 38.34 | 0.00 | 0.01 | 0.00 |
| traffic sign - 6 | 31.79 | 0.00 | 0.00 | 0.00 |
| person - 7 | 28.33 | 0.00 | 0.00 | 0.00 |
| car - 8 | 81.19 | 25.79 | 20.09 | 8.85 |
| truck - 9 | 25.40 | 8.55 | 5.09 | 0.55 |
| parking - 10 | – | 2.18 | 1.57 | 3.34 |
| vegetation - 11 | – | 38.12 | 31.43 | 25.91 |
| rail track - 12 | – | – | 5.04 | 5.68 |
| sky - 13 | – | – | 80.24 | 69.65 |
| motorcycle - 14 | – | – | – | 9.45 |
| autorickshaw - 15 | – | – | – | 9.87 |
| bridge/tunnel - 16 | – | – | – | 1.04 |
| **Average** | 47.76 | 19.73 | 18.76 | 14.45 |

*Table 40.* Performance of NNCSL on Med Semi-Supervised-JASCL benchmark.

| Class | Session 0 | Session 1 | Session 2 | Session 3 | Session 4 | Session 5 |
|---|---|---|---|---|---|---|
| class_0 | 0.985 | 0.814 | 0.925 | 0.957 | 0.959 | 0.962 |
| class_1 | 0.831 | 0.068 | 0.024 | 0.015 | 0.011 | 0.009 |
| class_2 | 0.575 | 0.039 | 0.029 | 0.011 | 0.009 | 0.008 |
| class_3 | 0.802 | 0.038 | 0.022 | 0.011 | 0.010 | 0.006 |
| class_4 | 0.720 | 0.017 | 0.008 | 0.005 | 0.001 | 0.002 |
| class_5 | 0.786 | 0.034 | 0.012 | 0.017 | 0.014 | 0.010 |
| class_6 | 0.733 | 0.050 | 0.029 | 0.018 | 0.019 | 0.015 |
| class_7 | 0.695 | 0.005 | 0.008 | 0.005 | 0.004 | 0.005 |
| class_8 | 0.665 | 0.030 | 0.022 | 0.009 | 0.008 | 0.004 |
| class_9 | 0.799 | 0.023 | 0.008 | 0.005 | 0.003 | 0.008 |
| class_10 | 0.230 | 0.042 | 0.019 | 0.016 | 0.013 | 0.011 |
| class_11 | 0.814 | 0.016 | 0.013 | 0.004 | 0.004 | 0.002 |
| class_12 | 0.697 | 0.017 | 0.025 | 0.023 | 0.018 | 0.017 |
| class_13 | 0.726 | 0.013 | 0.007 | 0.008 | 0.003 | 0.003 |
| class_14 | 0.711 | 0.013 | 0.015 | 0.014 | 0.011 | 0.009 |
| class_15 | 0.719 | 0.020 | 0.018 | 0.015 | 0.012 | 0.010 |
| class_16 | – | 0.121 | 0.012 | 0.007 | 0.012 | 0.017 |
| class_17 | – | 0.071 | 0.022 | 0.012 | 0.016 | 0.024 |
| class_18 | – | 0.093 | 0.015 | 0.011 | 0.012 | 0.013 |
| class_19 | – | 0.189 | 0.012 | 0.008 | 0.007 | 0.005 |
| class_20 | – | 0.044 | 0.023 | 0.014 | 0.015 | 0.026 |
| class_21 | – | – | 0.023 | 0.003 | 0.002 | 0.002 |
| class_22 | – | – | 0.518 | 0.032 | 0.018 | 0.029 |
| class_23 | – | – | 0.276 | 0.021 | 0.019 | 0.018 |
| class_24 | – | – | 0.013 | 0.005 | 0.003 | 0.003 |
| class_25 | – | – | 0.019 | 0.008 | 0.008 | 0.012 |
| class_26 | – | – | 0.049 | 0.009 | 0.004 | 0.007 |
| class_27 | – | – | – | 0.166 | 0.011 | 0.005 |
| class_28 | – | – | – | 0.052 | 0.006 | 0.005 |
| class_29 | – | – | – | 0.276 | 0.016 | 0.014 |
| class_30 | – | – | – | 0.106 | 0.006 | 0.007 |
| class_31 | – | – | – | – | 0.023 | 0.016 |
| class_32 | – | – | – | – | 0.048 | 0.013 |
| class_33 | – | – | – | – | 0.002 | 0.003 |
| class_34 | – | – | – | – | – | 0.216 |
| class_35 | – | – | – | – | – | 0.413 |
| class_36 | – | – | – | – | – | 0.237 |
| class_37 | – | – | – | – | – | 0.289 |
| **Average** | 0.700 | 0.047 | 0.047 | 0.030 | 0.011 | 0.040 |

*Table 41.* Performance of HALO on Semi-Supervised Natural-JASCL.

| Class | Session 0 | Session 1 | Session 2 | Session 3 |
|---|---|---|---|---|
| road - 0 | 91.18 | 0.00 | 0.00 | 0.00 |
| sidewalk - 1 | 48.82 | 0.00 | 0.00 | 0.00 |
| building - 2 | 86.42 | 0.00 | 0.00 | 0.00 |
| wall - 3 | 20.84 | 0.00 | 0.00 | 0.00 |
| fence - 4 | 25.28 | 0.00 | 0.00 | 0.00 |
| traffic light - 5 | 38.34 | 0.00 | 0.00 | 0.00 |
| traffic sign - 6 | 31.79 | 0.00 | 0.00 | 0.00 |
| person - 7 | 28.33 | 0.00 | 0.00 | 0.00 |
| car - 8 | 81.19 | 0.00 | 0.00 | 0.00 |
| truck - 9 | 25.40 | 0.00 | 0.00 | 0.00 |
| parking - 10 | – | 0.55 | 0.00 | 0.00 |
| vegetation - 11 | – | 20.87 | 0.00 | 0.00 |
| rail track - 12 | – | – | 0.90 | 0.00 |
| sky - 13 | – | – | 27.32 | 0.00 |
| motorcycle - 14 | – | – | – | 1.15 |
| autorickshaw - 15 | – | – | – | 17.82 |
| bridge/tunnel - 16 | – | – | – | 2.60 |
| **Average** | 47.76 | 1.78 | 2.02 | 1.27 |

*Table 42.* Performance of MDIL (domain incremental) on Med JASCL-Disjoint benchmark.

| Class | Session 0 | Session 1 | Session 2 |
|---|---|---|---|
| class_0 | 0.982 | 0.972 | 0.967 |
| class_1 | 0.851 | 0.000 | 0.000 |
| class_2 | 0.790 | 0.000 | 0.000 |
| class_3 | 0.864 | 0.000 | 0.003 |
| class_4 | 0.785 | 0.000 | 0.000 |
| class_5 | 0.767 | 0.000 | 0.001 |
| class_6 | 0.733 | 0.000 | 0.000 |
| class_7 | 0.798 | 0.000 | 0.000 |
| class_8 | 0.787 | 0.000 | 0.002 |
| class_9 | 0.791 | 0.000 | 0.000 |
| class_10 | 0.475 | 0.000 | 0.001 |
| class_11 | 0.817 | 0.000 | 0.001 |
| class_12 | 0.825 | 0.000 | 0.000 |
| class_13 | 0.846 | 0.000 | 0.000 |
| class_14 | 0.784 | 0.000 | 0.001 |
| class_15 | 0.774 | 0.000 | 0.000 |
| class_16 | – | 0.374 | 0.000 |
| class_17 | – | 0.256 | 0.000 |
| class_18 | – | 0.599 | 0.000 |
| class_19 | – | 0.541 | 0.000 |
| class_20 | – | 0.522 | 0.001 |
| class_21 | – | – | 0.288 |
| class_22 | – | – | 0.618 |
| class_23 | – | – | 0.514 |
| class_24 | – | – | 0.451 |
| class_25 | – | – | 0.366 |
| class_26 | – | – | 0.298 |
| **Average** | 0.779 | 0.115 | 0.098 |

*Table 43.* Performance of Gen-Replay (generative replay with diffusion) on Med JASCL-Disjoint benchmark.

| Class | Session 0 | Session 1 | Session 2 |
|---|---|---|---|
| class_0 | 0.985 | 0.966 | 0.960 |
| class_1 | 0.831 | 0.008 | 0.018 |
| class_2 | 0.575 | 0.013 | 0.002 |
| class_3 | 0.802 | 0.002 | 0.003 |
| class_4 | 0.720 | 0.002 | 0.009 |
| class_5 | 0.786 | 0.006 | 0.023 |
| class_6 | 0.733 | 0.001 | 0.000 |
| class_7 | 0.696 | 0.000 | 0.000 |
| class_8 | 0.665 | 0.001 | 0.000 |
| class_9 | 0.799 | 0.017 | 0.001 |
| class_10 | 0.230 | 0.003 | 0.000 |
| class_11 | 0.814 | 0.000 | 0.000 |
| class_12 | 0.697 | 0.001 | 0.000 |
| class_13 | 0.726 | 0.000 | 0.000 |
| class_14 | 0.711 | 0.003 | 0.000 |
| class_15 | 0.719 | 0.002 | 0.001 |
| class_16 | – | 0.192 | 0.000 |
| class_17 | – | 0.070 | 0.000 |
| class_18 | – | 0.546 | 0.002 |
| class_19 | – | 0.334 | 0.024 |
| class_20 | – | 0.329 | 0.001 |
| class_21 | – | – | 0.507 |
| class_22 | – | – | 0.637 |
| class_23 | – | – | 0.411 |
| class_24 | – | – | 0.415 |
| class_25 | – | – | 0.166 |
| class_26 | – | – | 0.430 |
| **Average** | 0.700 | 0.076 | 0.102 |

*Table 44.* Performance of CLIP-driven method on Med JASCL-Mixed benchmark. The model is pre-trained on a large number of classes in the benchmark.

| Class | Session 0 | Session 1 | Session 2 | Session 3 | Session 4 |
|---|---|---|---|---|---|
| class_1 | 0.881 | 0.825 | 0.702 | 0.702 | 0.036 |
| class_2 | 0.844 | 0.460 | 0.368 | 0.368 | 0.000 |
| class_3 | 0.844 | 0.829 | 0.218 | 0.218 | 0.000 |
| class_4 | 0.699 | 0.598 | 0.174 | 0.174 | 0.000 |
| class_5 | 0.628 | 0.085 | 0.001 | 0.001 | 0.000 |
| class_6 | 0.273 | 0.009 | 0.001 | 0.001 | 0.000 |
| class_7 | 0.757 | 0.696 | 0.139 | 0.139 | 0.000 |
| class_8 | 0.850 | 0.013 | 0.000 | 0.000 | 0.000 |
| class_9 | 0.652 | 0.233 | 0.002 | 0.002 | 0.000 |
| class_10 | 0.740 | 0.726 | 0.000 | 0.000 | 0.000 |
| class_11 | – | 0.808 | 0.048 | 0.048 | 0.000 |
| class_12 | – | 0.834 | 0.000 | 0.000 | 0.014 |
| class_13 | – | 0.002 | 0.002 | 0.002 | 0.026 |
| class_14 | – | 0.013 | 0.000 | 0.000 | 0.000 |
| class_15 | – | 0.012 | 0.000 | 0.000 | 0.000 |
| class_16 | – | 0.946 | 0.941 | 0.941 | 0.241 |
| class_17 | – | 0.106 | 0.000 | 0.000 | 0.030 |
| class_18 | – | 0.314 | 0.261 | 0.261 | 0.006 |
| class_19 | – | – | 0.800 | 0.802 | 0.177 |
| class_20 | – | – | 0.840 | 0.842 | 0.000 |
| class_21 | – | – | 0.855 | 0.855 | 0.180 |
| class_22 | – | – | 0.000 | 0.000 | 0.000 |
| class_23 | – | – | 0.000 | 0.000 | 0.000 |
| class_24 | – | – | 0.098 | 0.098 | 0.000 |
| class_25 | – | – | – | 0.020 | 0.137 |
| class_26 | – | – | – | 0.000 | 0.000 |
| class_27 | – | – | – | 0.010 | 0.000 |
| class_28 | – | – | – | 0.009 | 0.001 |
| class_29 | – | – | – | – | 0.025 |
| class_30 | – | – | – | – | 0.148 |
| class_31 | – | – | – | – | 0.050 |
| class_32 | – | – | – | – | 0.430 |
| class_33 | – | – | – | – | 0.417 |
| class_34 | – | – | – | – | 0.729 |
| class_35 | – | – | – | – | 0.471 |
| **Average** | 0.716 | 0.417 | 0.227 | 0.196 | 0.089 |

