# OpenReview forum: "Continual Segmentation under Joint Nonstationarity"
_ICML.cc/2026/Conference — ICML 2026 regular_

### Official Review · Reviewer_NkPn · 2026-02-23

**Soundness:** 3
**Presentation:** 2
**Significance:** 2
**Originality:** 3
**Overall Recommendation:** 3
**Confidence:** 3

**Summary:**

This paper studies continual semantic segmentation under joint nonstationarity, where both classes and domains shift across sessions, and later sessions are few-shot (optionally with unlabeled data). This setting is particularly challenging and causes many existing methods to degrade rapidly. The paper proposes JASCL, which combines (i) gradient-adaptive stabilization (gradient-scaled perturbations on the pixel classifier to improve training stability) and (ii) prototype-anchored supervision (feature/prototype-validated pseudo-labeling for semi-supervised updates). The authors also introduce five benchmarks spanning 3D medical and 2D natural segmentation, and report consistent gains across backbones, including plug-in improvements on SAM, under severe class–domain shifts.

**Compliance With Llm Reviewing Policy:**

Affirmed.

**Final Justification:**

Thank you for the rebuttal. I am not convinced by the responses. While I agree that focusing on the JNS setting is challenging and very meaningful, the proposed solution does not appear strong enough. Based on the reported tables, I do not see sufficiently promising results. In the rebuttal, the authors highlight the improvement of c_29: 0.338 → 0.292 in Table 27. However, most new classes still fail to be learned effectively (e.g., c_25 = 0.004, c_26 = 0.000). On natural datasets, such as those in Table 33, JASCL retains some degree of plasticity, but at the cost of severe forgetting.

Overall, these results are not strong enough to support the claim that “JASCL is the first method achieving strong performance under JNS with both stability and plasticity.” Therefore, I maintain my score.

**Key Questions For Authors:**

See the weaknesses.

**Limitations:**

Yes.

**Strengths And Weaknesses:**

Strengths:
1. The paper targets an unusually challenging and practically motivated continual learning setting. While the results suggest the problem remains far from solved, I appreciate the authors’ effort to study a realistic failure mode of current methods.
2. The experimental coverage is extensive, including multiple benchmarks and backbones, which helps substantiate the empirical behavior of the proposed components.

Weaknesses:
1. The setting effectively combines several difficult (and arguably still open) continual learning challenges. While this combination is potentially clinically relevant, it also makes the study difficult to execute and interpret, and the current results do not demonstrate a clearly satisfactory solution under the most challenging shifts.
2. The paper argues that this complex scenario better reflects clinical practice, but the justification is not sufficiently supported. The authors should provide stronger evidence that such joint nonstationarity is common and unavoidable in real deployments, rather than something that could be substantially mitigated by dataset curation, acquisition protocols, or operational constraints.
3. Based on Table 27 and related results, the method appears to preserve performance on the initial classes (hence the limited overall drop), but performance on newly introduced classes remains weak. This raises concern that the method primarily stabilizes old knowledge while providing limited plasticity for learning new categories.

---

> ### Author Rebuttal · Authors · 2026-03-30
>
> Thank you for your appreciation.
>
> ## **Weaknesses:**
> **1.**
> The combination of CI, DI, and FS constraints is intentional and practically motivated. Such combinations are already well studied: **class and domain incremental** (MDIL), **class and few-shot incremental** (C-FSCIL, GAPS) (Appendix H). JASCL adds one additional practical constraint to each, a natural and necessary next step, as highlighted in lines 070-075, **recent surveys** (Zhang et al., 2025; Yuan & Zhao, 2024a) **emphasize the necessity of CL formulations that jointly account for class evolution, domain drift, and label scarcity**, yet principled solutions for dense prediction remain scarce. **Existing benchmarks that isolate these constraints are the idealization**, not the joint setting.
>
> **Executability** is demonstrated via over 37 baselines (Appendix H) across five benchmarks spanning 3D medical imaging, autonomous driving (where even SAM degrades), robotic surgery (Table 29, outperforming CAT), and object detection (Table 17). **Interpretability** is ensured via Tables 24 and 25 evaluating CI, DI, and CDIL individually and jointly, Figure 1 quantifying constraint contributions, Tables 34 and 36 decomposing module contributions, and Table 26 providing a single stability measure, **allowing readers to attribute performance changes to specific factors rather than treating the setting as a black box**.
>
> Absolute performance must be interpreted in the context of benchmark difficulty. Session 4 (CT→MRI) induces catastrophic forgetting in U-Net (Appendix J.1), which is architecture-specific (MedFormer performs well), while **results also depend on organ size, base model accuracy, and backbone capacity** (Appendix J).
>
> Just as ImageNet, COCO, ADE20K (20–40 mIoU to date) in CISS, Decathlon, and other large-scale benchmarks with several constraints were challenging to execute and interpret, and yielded imperfect results yet still drove progress, **we humbly submit that exposing failures under joint nonstationarity is itself a meaningful contribution**.
>
> Table 7 consistently improves over Table 3 (S1: 0.460→0.554, S2: 0.398→0.445, S3: 0.329→0.414, S4: 0.025→0.058, 2.3×; S5: 0.324→0.368), with further recovery to 0.197 using 42 unlabeled samples (Table 21, 7.9×). **JASCL achieves the lowest Total Drop among all strong baseline methods to date (Table 26), outperforming competitive approaches**, and consistently generalizes to various other domains under joint non-stationarity.
>
> **2.**
> We respectfully argue that mere protocol and curation cannot, for four concrete reasons. **First**, pixel-level annotation of 3D medical volumes is very costly, so few-shot supervision is not a choice but a constraint. **Second**, different hospitals use different scanner manufacturers and field strengths, making domain shift institutionally unavoidable. Modality shifts (CT to MRI) arise naturally when clinical pathways require different imaging for different conditions. **Third**, new diseases and tumour subtypes emerge over time, and no protocol can anticipate future class requirements, as exemplified by COVID-19 lung lesions. **Fourth**, in robotic surgery, surgical tools and tissue appearance vary across manufacturers, procedures, and patient variability, as confirmed by Table 29, where even specialized methods like CAT fail. Together, these factors make joint non-stationarity unavoidable. The same applies to autonomous driving: new traffic participants, new geographic domains, and annotation scarcity are all operationally unavoidable, which is precisely why **recent surveys consistently identify joint non-stationarity as an open problem**.
>
> **3.**
> **Preserving old knowledge is itself a major unsolved challenge, as most baselines collapse to near-zero Dice after two sessions** (Table 2) due to fine-tuning with novel classes. JASCL achieves lowest Total Drop (Table 26), demonstrating stability and plasticity are jointly maintained.
>
> Numerical evidence of plasticity (SS:semi-supervised; P: Avg plasticity-new classes):
>
> |3D U-Net|P|
> |------|--------------------|
> |JASCL|0.129 ( from Table 27 ) |
> |NNCSL(SS)|0.143|
> |JASCL(SS)|**0.157**|
>
> |3D Medformer|P|
> |------|--------------------|
> |CSL(SS)|0.125|
> |JASCL(SS)|**0.154**|
>
> |SAM|P|
> |------|---------------|
> |SAM|7.7|
> |JASCL(SAM)|**8.6**|
>
> |Natural Deeplab|P|
> |------|----------|
> |HALO|10.7|
> |GAPS|23.2|
> |JASCL(w/o PAS)|28.4|
> |JASCL(GAS+PAS)|**32.2**|
>
> |RCNN-detection|P|
> |--------------|---------------|
> |Vanilla|0|
> |JASCLw/oGAS|0.004|
> |JASCLw/oPAS|0.051|
> |JASCL|**0.063**|
>
> PAS adds plasticity beyond GAS, improving novel class learning across 3D, SAM, natural scenes, and detection (Fig.6). On VOC and ADE20K, JASCL achieves best novel class scores (VOC 15-1: 53.1 vs CoGaMiD 52.2; VOC 15-5: 60.8 vs CoGaMiD 60.5) (reviewer rwDq Weakness Point 2). **Plasticity also depends on organ size, K shots, domain shift, and backbone** (Appendix J). JASCL is a strong contender for plasticity over existing methods in joint nonstationarity.

---

> > ### Author Rebuttal · Reviewer_NkPn · 2026-04-02
> >
> > Although continual learning under CI, DI, or FS has been studied for years, existing CL methods still struggle even when handling these settings individually. This paper further combines CI, DI, and FS into a more challenging setting. However, the proposed solution does not yet deliver satisfactory results, and at this stage it is hard to see clear evidence of its potential. As shown in Tables 27, 28, and related results, the model fails to learn the new classes effectively, leading to very poor performance. For JASCL in particular, the method seems to preserve old knowledge at the cost of plasticity.

---

> > > ### Author Response · Authors · 2026-04-03
> > >
> > > ## **On satisfactory results:**
> > > We respectfully disagree.
> > >
> > > Below, baselines are grouped by setting. Dice(medical) ∈ (0-1), mIoU(natural scenes) ∈ (0-100). (SS: semi-supervised; M: Method; S: session)
> > >
> > > ### **Only CI:**
> > > ### A) *Classification*
> > > These methods achieve ~60–70% on CIFAR-100
> > >
> > > On JNS
> > > |M|S0|S1|S2|
> > > |---|---|---|---|
> > > |STAR|0.70|0.05|0.02|
> > > |C-Flat|0.70|0.17|0.03|
> > > |UCL|0.70|0.43|0.33|
> > > |JASCL|0.74|0.46|0.40|
> > > |JASCL(SS)|0.74|**0.55**|**0.45**|
> > >
> > > ### **0.74→0.55 Dice is strong given base avg.0.74, indicating robust retention and novel learning**
> > > Thm 3.7 & 3.8 explain why UCL and C-Flat fail
> > >
> > > ### B) *Segmentation*
> > > ### i) 3D medical
> > > Methods like CLIP-CT achieve ~0.8 Dice on benchmarks
> > >
> > > Performance on JNS
> > > |M|S0|S1|S2|
> > > |---|---|---|---|
> > > |CLIP-CT|0.48|0.19|0.14|
> > > |JASCL|0.74|0.46|0.40|
> > > |JASCL(SS)|0.74|0.55|0.45|
> > >
> > > Strong methods collapse, **JASCL remains clearly better and stable**
> > >
> > > ### ii) Natural scenes
> > > On Pascal-VOC
> > > |M|0-15|16-20|
> > > |---|---|---|
> > > |MiB|75.5|49.4|
> > > |Adapt_replay|79.7|59.7|
> > > |Saving100x|79.5|58.9|
> > > |JASCL|79.5|**60.8**|
> > >
> > > **JASCL => strong plasticity across old and new classes**
> > >
> > > On JNS
> > > |M|S0|S1|S2|
> > > |---|---|---|---|
> > > |MiB|0.70|0.27|0.10|
> > > |Adapt_replay|0.70|0.04|0.03|
> > > |Saving100x|0.70|0.07|0.05|
> > > |JASCL|0.74|0.46|0.40|
> > > |JASCL(SS)|0.74|0.55|0.45|
> > >
> > > **SOTA => fail on JNS; JASCL => strong performance**
> > >
> > > ### **CI+DI:**
> > > MDIL => strong on benchmarks (CS→BDD→IDD)
> > >
> > > On JNS
> > > |M|S0|S1|S2|
> > > |---|---|---|---|
> > > |MDIL|0.78|0.12|0.10|
> > > |JASCL|0.74|0.46|0.40|
> > > |JASCL(SS)|0.74|0.55|0.45|
> > >
> > > MDIL fails under JNS
> > >
> > > ### **CI+FS:**
> > > ### A) *Classification*
> > > Attain 57-68% on CIFAR-100 while on JNS
> > > |M|S0|S1|S2|
> > > |---|---|---|---|
> > > |NC-FSCIL|0.39|0.08|0.08|
> > > |C-FSCIL|0.79|0.33|0.30|
> > > |SoftNet|0.82|0.31|0.15|
> > > |FeCAM|0.70|0.05|0.04|
> > > |UaD-CE|0.70|0.08|0.08|
> > > |JASCL|0.74|0.46|0.40|
> > > |JASCL(SS)|0.74|0.55|0.45|
> > >
> > > **Strong SOTA fail under JNS setting**
> > >
> > > ### B) *Segmentation*
> > > On Pascal-VOC
> > > |M|Base|Novel|
> > > |---|---|---|
> > > |MiB|43.9|2.6|
> > > |PIFS|64.1|16.9|
> > > |GAPS|66.8|23.6|
> > >
> > > **Novel classes are consistently lower than base, showing plasticity challenges**
> > >
> > > On JNS
> > > |M|S0|S1|S2|
> > > |---|---|---|---|
> > > |MiB|0.70|0.27|0.10|
> > > |PIFS|0.70|0.13|0.08|
> > > |GAPS|0.70|0.33|0.25|
> > > |JASCL|0.74|0.46|0.40|
> > > |JASCL(SS)|0.74|0.55|0.45|
> > >
> > > **All baselines degrade even on old classes; JASCL remains strongest and stable**
> > > ### **Methods perform strongly on isolated CI/DI/FS but fail under JNS. JASCL maintains 0.74→0.55 (strong, not merely satisfactory) as also noted by other reviewers. CI+DI+FS are primary real-world contributors to JNS**
> > >
> > > ## **On plasticity**
> > > ### **1. CI+FS segmentation inherently has low novel performance**
> > > Pascal-VOC
> > > |M|Base|Novel|
> > > |---|---|---|
> > > |MiB|43.9|2.6|
> > > |PIFS|64.1|16.9|
> > > |GAPS|66.8|23.6|
> > >
> > > **existing natural scenes methods => Low novel performance**
> > >
> > > ### **No prior CI+FS method exists in medical; JNS highlights this gap through extensive evaluation**
> > > ### **2. Reason for low novel performance: CI+FS follows non-episodic FS**
> > > - In 5-shot, only **5 samples per novel class** in entire train
> > > - Ex: 20 classes → 100 samples only (see PIFS)
> > >
> > > Episodic FS: large samples (~100-500 samples/novel class in train)→stronger supervision in Episodic FS
> > > ### **3. Strong plasticity in natural scenes (on JNS)**
> > > |M|Old|Novel|
> > > |---|---|---|
> > > |GAPS|23.5|23.2|
> > > |JASCL|30.4|**32.2**|
> > > ### → **Novel ≥ Old → no stability-plasticity trade-off (Tables 33,39), a strong result comparable to established VOC with CI+FS**. Old:avg. base classes across sessions;Novel:avg. novel classes after S0
> > > ### **4. Why medical plasticity is lower in non-episodic FS than in natural**
> > > - Small organ structures + 3D sparsity → few foreground voxels
> > > - Low contrast & intensity similarity → weak separability
> > > - High inter-class overlap + intra-class variability → ambiguous anatomy
> > > - Weak/blurred boundaries + limited texture cues (vs RGB)
> > > - Strong shift CT→MRI
> > >
> > > →weaker supervision than natural FS
> > > ### **5. Medical plasticity still clearly present (novel classes)**
> > > c: class
> > > - Tab.27
> > >   - c_29:**0.338**→**0.292**
> > >   - c_21:0.032→**0.432**
> > >   - c_33:**0.299**
> > > - Tab.28
> > >   - c_16:**0.520**
> > >   - c_13:0.025→**0.205**
> > >   - c_12:**0.466**
> > > - Tab.30
> > >   - c_12:**0.754**
> > >   - c_25:**0.337**
> > >   - c_16:**0.538**
> > >
> > > **Above novel classes show strong plasticity**
> > >
> > > - Semi-sup. results (with PAS) (Tab.31) show gains:
> > >   - c_26:0.0→**0.054**
> > >   - c_27:0.075→**0.182**
> > >   - c_31:0.109→**0.252**
> > >   - c_37:0.066→**0.286**
> > >
> > > → Plasticity gains with unlabeled data indicate *quality data* deficiency, not method limits
> > >
> > > - JASCL does **not sacrifice plasticity**
> > > - It achieves:
> > >    - strong base performance
> > >    - strong novel learning in natural scenes + meaningful gains in medical
> > >    - improved plasticity with PAS
> > >
> > > - Lower novel medical scores due to:
> > >    - small; low contrast; overlapping; weak boundary organs
> > >
> > > Please see Appendix J.3.i
> > > ### JASCL is the **first method achieving strong performance under JNS** with both **stability and plasticity**
> > > Thank you for the thoughtful review and constructive feedback. We believe all concerns are addressed and kindly request a score upgrade.

---

### Official Review · Reviewer_eDcV · 2026-03-08

**Soundness:** 2
**Presentation:** 2
**Significance:** 3
**Originality:** 2
**Overall Recommendation:** 4
**Confidence:** 3

**Summary:**

This paper formalizes the problem of Joint Nonstationarity in continual semantic segmentation, where categories, data distributions, and annotation availability evolve simultaneously. The authors propose JASCL, a framework that mitigates catastrophic forgetting and overfitting through GAS and PAS, establishing a robust baseline for dense prediction in highly dynamic environments

**Compliance With Llm Reviewing Policy:**

Affirmed.

**Final Justification:**

The authors have provided clear responses to my concerns raised in the review, and I upgrade my score to 4.

**Key Questions For Authors:**

See weakness.

**Limitations:**

Yes

**Strengths And Weaknesses:**

Strengths
1. The paper identifies and formalizes Joint Nonstationarity, a challenging yet highly practical setting that unifies class, domain, and label shifts.
2. The proposed components are lightweight and model-agnostic, demonstrating impressive performance gains across a wide range of architectures.
3. The authors evaluate their method on five diverse benchmarks covering both 3D medical imaging and 2D natural driving scenes.

Weaknesses
1. The GAS conceptually mirrors noise-driven regularization in Explicit Regularization in Overparametrized Models via Noise Injection (arXiv:2206.04613). The authors should clarify the technical distinction between their gradient-adaptive scaling and existing layer-wise perturbation methods, particularly concerning its unique necessity for continual learning.
2. The ablation studies in Figure 5 suggest that GAS is the primary performance driver while PAS's contribution is marginal. For rigor, numerical results for JASCL variants (e.g., w/o GAS, w/o PAS) should be added to Tables 2, 5, and 6 to clarify the individual and synergistic effects of each module.
3. The method relies on several manual hyperparameters (noise scaling, confidence, and similarity thresholds). In dynamic CL environments, per-session tuning is impractical. The authors should discuss the sensitivity of these parameters or demonstrate their stability across different non-stationary sessions.
4. The paper lacks a comparison of training efficiency. The authors should provide quantitative data regarding training time and VRAM consumption compared to standard baselines to evaluate the practical scalability of JASCL.

---

> ### Author Rebuttal · Authors · 2026-03-30
>
> Thank you for your appreciation.
>
> ## **Weaknesses**:
>
> **1.**
> GAS vs Orvieto et al. (arXiv:2206.04613) w.r.t continual learning:
>
> a). Inverted scaling direction. Orvieto et al. apply uniform or layer-wise constant noise variance fixed as a global hyperparameter. GAS scales noise inversely to squared gradient magnitude (Eq. 1), so task-critical parameters (large gradients) receive minimal perturbation while low-sensitivity parameters receive larger perturbation. This inversion is the defining mechanism of GAS and has no counterpart in Orvieto et al.
>
> b). Different theoretical objective. Orvieto et al. derive their method by inducing an implicit regularizer (Frobenius/nuclear norm) via fixed noise injection to control variance explosion in overparametrized models. GAS instead approximates the optimal diagonal Gaussian posterior under Laplace approximation, minimizing KL divergence to the per-session posterior (Proposition 3.3, Corollary 3.4). The former is session-agnostic while the latter explicitly updates per session, which is essential when the posterior changes with each new domain and class set.
>
> c). Static vs. adaptive noise under distribution shift. Orvieto et al. fix a constant noise scale throughout training, appropriate for stationary single-task settings. Continual learning with joint non-stationarity breaks this assumption as Fisher Information changes across sessions, so any noise scale calibrated on one session becomes misaligned with the curvature structure of the next. GAS avoids this by recomputing per-parameter noise scales $\tilde{G}_{ij}^{-1}$ (Eq. 1) at each optimization step, remaining aligned with current curvature regardless of how many sessions have passed. Table 20 confirms this empirically: Dropout/Weight Decay (analogous to fixed noise injection) achieves Session 1 Dice of 0.191 versus GAS at 0.460.
>
> d). Different problem setting. Orvieto et al. address variance explosion in overparametrized static models. GAS addresses catastrophic forgetting and few-shot overfitting under sequential distribution shift, a fundamentally different failure mode where per-parameter curvature tracking across sessions is essential. Fixed or layer-wise noise injection has no mechanism to track which parameters are relevant across sessions, making GAS not merely an extension of noise injection but a uniquely necessary design for continual learning with joint non-stationarity.
>
> We thank the reviewer for highlighting this connection and will add a citation to Orvieto et al. in the revision.
>
> **2.**
> Numerical ablation results already exist in the appendix. Tables 34 and 36 report full numerical results for JASCL without GAS (Med JASCL-Mixed) and JASCL without PAS (Semi-Supervised Natural-JASCL), respectively, corresponding to Fig. 5, which already visually confirms the synergistic effect of both modules. As suggested, we will move these results from the appendix to the relevant main paper tables in the revision.
>
> On PAS contribution being marginal. We respectfully disagree. GAS and PAS address complementary failure modes: GAS is applicable to all settings (Tables 2 to 7), addressing optimization instability under few-shot distribution shift, while PAS is designed specifically for the semi-supervised setting (Tables 6 and 7), where unlabeled data is available and reliable pseudo-label selection is needed. The two modules should not be compared as primary vs. marginal but evaluated in their respective settings.
> Critically, under severe domain shift PAS dominates over GAS: at Session 4 (CT to MRI modality shift), GAS alone yields Dice 0.025 (Table 3), PAS recovers this to 0.058 (Table 7) and further to 0.197 with 42 unlabeled samples (Table 21), as unlabeled data from the new modality provides the signal that scarce labeled data cannot. PAS is therefore essential where labeled data is scarce, domain shift is significant, and unlabeled data is cheap to acquire, which is precisely one of the most challenging regimes of the JASCL benchmarks.
>
> **3.**
> **Tables 8, 10 and Appendices C, D detail the robustness/sensitivity** of $\varepsilon$, noise variance, $\tau_{conf}$ and $\tau_{sim}$​ across different benchmarks, along with the *Hyperparameters* section in the main paper. These results suggest that JASCL is robust to feasible values of hyperparameters. **All hyperparameters are fixed globally before Session 0 and remain unchanged across all sessions and all application domains (autonomous driving, medical, robotics), requiring no per-session tuning** as adaptivity comes from GAS (Eq. 1) and PAS (Eq. 3 to 5). Table 19 also confirms JASCL adds no extra train time, FLOPs, parameters, or VRAM over the Vanilla model, making JASCL practically deployable in dynamic continual learning.
>
> **4.**
> Please see the response to Point 4 of Key Questions from reviewer rwDq, which **indicates the training efficiency of JASCL compared to baselines while delivering higher performance.** Table 19 confirms it over the vanilla model.

---

> > ### Author Rebuttal · Reviewer_eDcV · 2026-04-03
> >
> > Thank you for your detailed response. All my concerns have been addressed. I raise my score to 4.

---

> > > ### Author Response · Authors · 2026-04-03
> > >
> > > Dear Reviewer eDcV,
> > >
> > > Thank you sincerely for your thorough and thoughtful engagement throughout the rebuttal. Your time, careful consideration, and constructive feedback are greatly appreciated. We are also truly grateful for raising the score. We are glad that all concerns have been fully resolved. We kindly leave any further consideration of the score to your discretion.

---

### Official Review · Reviewer_a6Ss · 2026-03-12

**Soundness:** 4
**Presentation:** 3
**Significance:** 4
**Originality:** 4
**Overall Recommendation:** 6
**Confidence:** 4

**Summary:**

This study formalizes continual segmentation under coupled class, domain, and label shifts and investigate learning in heterogeneous dense prediction environments with limited annotations and abundant unlabeled data. To address instability and overfitting arising from few-shot supervision under distribution drift, the paper introduces gradient-adaptive stabilization. Furthermore, this paper introduces prototype anchored supervision to leverage unlabeled data. All of them are strong innovation. I really appreciate this paper that I can hardly find issues in this paper.

**Compliance With Llm Reviewing Policy:**

Affirmed.

**Key Questions For Authors:**

I appreciate this paper so much that it is difficult for me to find issues. The only suggestion I have is that the authors add more explanations and intuitive interpretations of some formulas and proofs to help readers understand them more easily.

**Limitations:**

The paper should include a discussion of the limitations of the proposed method. Any method has potential shortcomings, and discussing them, in my opinion, will not weaken readers’ perception of the paper’s novelty; instead, it can make the work appear more reliable. I suggest that the authors outline possible limitations of the method across different dense prediction scenarios, or discuss the sensitivity of the introduced hyperparameters under varying conditions.

**Strengths And Weaknesses:**

Strengths

This paper presents several strong contributions, and its level of novelty meets the publication requirements. The experiments are relatively comprehensive, validating the effectiveness of the proposed method on multiple datasets under different strategies. Moreover, each lemma is accompanied by its corresponding proof, which is highly commendable.

Weaknesses

The paper contains many proofs and mathematical formulas, which may hinder readability. The authors should add more explanations and intuitive interpretations of the formulas and proofs to facilitate readers’ understanding.

---

> ### Author Rebuttal · Authors · 2026-03-30
>
> Thank you for your huge appreciation.
>
> ## **Weaknesses**
>
> We agree, and we will revise the paper to include additional intuitive explanations alongside key formulas and proofs, highlighting their role in the overall method. In particular, we will add brief interpretations after each major result and include high-level summaries to guide readers through the theoretical sections without sacrificing rigour.
>
> ## **Key Questions**
> Already responded in Weaknesses
>
> ## **Limitations**
>
> Thank you for asking this.
>
> A) Limitations across different scenarios: Detailed in Appendix J, including severe domain shifts, noisy unlabeled data, organ size variability in 3D medical domains, autonomous driving, and 2D robotic surgery. We have also provided some solutions for these limitations (as potential future work). We will include key limitations in the camera-ready main paper.
>
> B) Sensitivity of hyperparameters:
>
> Details in Appendix C, Table 8 for GAS hyperparameters.
>
> Details in Appendix D, Table 10 for PAS hyperparameters.

---

> > ### Author Rebuttal · Reviewer_a6Ss · 2026-04-05
> >
> > The authors have addressed all my concerns. I have no additional comments.

---

> > > ### Author Response · Authors · 2026-04-05
> > >
> > > Dear Reviewer a6Ss,
> > >
> > > Thank you so much for your generous and encouraging engagement throughout the review process. Your appreciation of the work and thoughtful suggestions toward improving readability are greatly valued. We will carefully incorporate your feedback in the revision.

---

### Official Review · Reviewer_rwDq · 2026-03-16

**Soundness:** 3
**Presentation:** 3
**Significance:** 3
**Originality:** 3
**Overall Recommendation:** 4
**Confidence:** 3

**Summary:**

This paper studies continual semantic segmentation under joint nonstationarity, where class sets, domain distributions, and supervision availability evolve simultaneously over time. The research examines an important concept that more closely reflects realistic deployment conditions compared with prior continual learning formulations that typically isolate class-incremental, domain-incremental, or few-shot constraints. To address this problem, the authors introduce a framework named JASCL (Jointly Anchored and Stabilized Continual Learning). The framework integrates two main components: Gradient Adaptive Stabilization (GAS) and Prototype Anchored Supervision (PAS). GAS introduces parameter-wise stochastic perturbations scaled by gradients to stabilize training and balance plasticity and stability under distribution shifts. PAS leverages unlabeled data through prototype-guided validation of pseudo-labels in a semi-supervised mean-teacher framework, aiming to reduce error propagation during incremental learning.

**Compliance With Llm Reviewing Policy:**

Affirmed.

**Final Justification:**

Thanks for the rebuttal. My concerns have been addressed.

**Key Questions For Authors:**

1. Generalization to existing benchmarks:
Have the authors evaluated JASCL on existing continual semantic segmentation benchmarks used in prior work (e.g., standard class-incremental segmentation benchmarks)? If so, including such comparisons would help position the method more clearly.

2. Relative importance of GAS vs PAS:
The ablation study suggests both components contribute to performance improvements. Could the authors provide more detailed analysis quantifying the individual impact of GAS and PAS across different regimes (e.g., domain shift vs class increment)?

3. Sensitivity to pseudo-label thresholds:
PAS relies on confidence and similarity thresholds. How sensitive is the method to these hyperparameters, and does performance remain stable across different datasets?

4. Computational overhead:
While the paper claims minimal computational overhead, more detailed analysis comparing training cost with baselines would help clarify the practical implications of the proposed approach.

5. Robustness to severe domain shifts:
Some experiments show significant performance drops under extreme domain shifts (e.g., CT → MRI). Could the authors discuss potential strategies for improving robustness in such scenarios?

**Limitations:**

Yes. The paper includes an impact statement discussing potential risks such as error accumulation from pseudo-labels and biases in unlabeled data, especially in safety-critical applications.

**Strengths And Weaknesses:**

# Strengths

1. Timely and relevant problem formulation. The paper highlights a realistic but underexplored setting in continual learning: joint nonstationarity, where class distributions, domains, and supervision levels evolve simultaneously. This formulation better reflects real-world applications such as medical imaging and autonomous driving, where these shifts often occur together rather than independently. The proposed formulation therefore provides a meaningful step toward bridging the gap between academic benchmarks and deployment scenarios.

2. Conceptually clear framework combining stabilization and semi-supervised learning. The proposed JASCL framework combines two complementary ideas. GAS provides a mechanism for stability–plasticity trade-offs through gradient-scaled stochastic perturbations, while PAS improves pseudo-label reliability using feature-space prototypes and confidence thresholds. The integration of optimization stability and pseudo-label filtering is intuitive and well motivated.

3. Extensive empirical evaluation. The experimental section is comprehensive. The authors evaluate the method on both 3D medical datasets and 2D driving scene datasets, across multiple incremental sessions and learning regimes. The comparisons include a wide range of baselines and architectures, including U-Net, transformer-based models, and foundation models such as SAM. The results consistently demonstrate improvements over baselines across several benchmarks.

4. New benchmark protocols. The proposed JASCL benchmark suites represent a valuable contribution. By combining class increments, domain shifts, and limited annotations, the benchmarks provide a challenging testbed for future research in continual dense prediction.

5. Theoretical analysis. The paper provides theoretical analysis for both GAS and PAS, including connections to Fisher information, KL divergence analysis, and PAC-Bayes generalization bounds. While the assumptions are simplified, the analysis provides useful intuition for why gradient-adaptive noise scaling and prototype filtering may improve stability.


# Weaknesses

1. Limited clarity on the practical contribution of individual components. Although the paper includes ablation experiments, the relative contribution of GAS and PAS could be analyzed more thoroughly. In particular, it would be useful to see clearer comparisons between GAS and other stabilization techniques (e.g., SAM or EWC-style regularization) under identical experimental setups.

2. Benchmark construction may introduce bias toward the proposed method. The benchmarks are newly proposed by the authors and may implicitly favor the proposed approach. While the evaluation appears extensive, it would strengthen the paper to also evaluate on widely used continual segmentation benchmarks or existing datasets used in prior work.

3. Theoretical analysis relies on strong assumptions. The theoretical results rely on assumptions such as gradient–Fisher correspondence and simplified pseudo-label error dynamics. While common in theoretical analyses, these assumptions may limit the practical interpretability of the theoretical guarantees.

4. Some presentation aspects could be improved. The paper contains dense theoretical sections and several long derivations that may be difficult to follow. A clearer separation between main insights and detailed proofs (possibly moving more content to the appendix) would improve readability.

---

> ### Author Rebuttal · Authors · 2026-03-29
>
> Thank you for your huge appreciation.
>
> ## **Weaknesses**:
>
> **1.**
> Med JASCL-Disjoint (S:Session):
>
> |Method|S0|S1|S2|
> |------|--|--|--|
> |UCL|0.70|0.43|0.33|
> |C-Flat(SAM)|0.70|0.17|0.03|
> |JASCL(GAS)|0.74|**0.46**|**0.40**|
>
> UCL (Ahn et al., NeurIPS 2019) already outperforms EWC yet uses shared node-level uncertainty estimates that become unreliable under domain shift. C-Flat (Bian et al., NeurIPS 2024) is SAM-based and perturbs toward worst-case directions without parameter-wise differentiation. GAS perturbs inversely to curvature, so important parameters stay stable and flat ones adapt, giving strictly lower expected loss increase (Theorem 3.8). Refer to Table 20, Section I.3, I.6.
>
> Med Semi-Supervised-JASCL (S:Session):
>
> |Method|S0|S1|S2|S3|S4|S5|
> |------|--|--|--|--|--|--|
> |NNCSL|0.66|0.14|0.10|0.14|0.01|0.05|
> |CSL|0.66|0.04|0.02|0.00|0.00|0.00|
> |JASCL(PAS)|0.64|**0.43**|**0.37**|**0.34**|**0.32**|**0.29**|
>
> PAS outperforms consistency-based CSL (Liu et al., ICCV 2025) and continual semi-supervised NNCSL (Kang et al., ICCV 2023) by filtering pseudo-labels with both confidence and prototype similarity, rejecting confident but wrong predictions under domain shift that CSL/NNCSL cannot. All experiments under identical setups.
>
> **2.**
> Although JASCL is designed for **joint non-stationarity** (JNS) induced by domain shift and few-shot classes in class-incremental semantic segmentation (CISS), we adopt the prototype-based filtering from PAS and apply it to pseudo-features in CoGaMiD (Zhu et al., NeurIPS 2025), a recent CISS method. Results on Pascal VOC and ADE20K in three overlapped scenarios are as follows (CoGaMiD reproduced with its public code):
>
> a). VOC 15-1 (6 steps) :
>
> |Method|0-15|16-20|all|
> |--------|----|----|----|
> |CoGaMiD(NeurIPS’25)|79.6|52.2|73.1|
> |EIR(CVPR’25)|79.4|52.6|73.0|
> |Adapter(AAAI’25)|79.9|51.9|73.2|
> |STAR(TPAMI’25)|80.0|51.2|73.1|
> |Ours|79.6|**53.1**|**73.3**|
>
> b). VOC 15-5 (2 steps) :
>
> |Method|0-15|16-20|all|
> |--------|----|----|----|
> |EIR|79.1|58.4|74.2|
> |Adapter|79.7|59.7|75.0|
> |CoGaMiD|79.5|60.5|75.0|
> |STAR|79.7|59.4|74.8|
> |Ours|79.5|**60.8**|**75.1**|
>
> c). ADE20K 100-50 (2 steps):
>
> |Method|0-100|101-150|all|
> |--------|----|----|----|
> |EIR|41.9|21.9|35.3|
> |CoGaMiD|41.9|22.7|35.5|
> |Ours|**42.1**|22.7|**35.7**|
>
> Our method remains competitive with CISS despite targeting JNS, where methods like Adapter underperform (Table 2: Adapt_replay).
>
> **3.**
> We note that the gradient Fisher correspondence follows the same approximation as EWC (Kirkpatrick et al., 2017) and related methods, and holds near convergence where GAS is applied, with approximation error ($∣ϵ_i​∣≤δ$) explicitly bounded.
>
> The linear error mixing yields concrete predictions: Theorem 3.13 shows that unfiltered semi-supervised learning provides no asymptotic gain, confirmed by CSL collapsing to 0.0 Dice by Session 3 (Table 7) while JASCL maintains 0.293; Theorem 3.14 shows that PAS achieves strictly lower asymptotic error, confirmed across all semi-supervised benchmarks.
>
> We respectfully submit that all theoretical claims are empirically validated across the benchmarks, supporting practical interpretability.
>
> **4.**
> Thanks. We will add the suggestion.
>
> ## **Key Questions**
>
> **1.**
> See Weaknesses point 2
>
> **2.**
> Class increment, minimal domain shift (Med SS-JASCL Sessions 1-3, CT→CT)
> - GAS: 0.076/0.058/0.047→0.460/0.398/0.329 (+505/+586/+600%) (Table 3)
> - PAS: +20/+12/+26% (Table 7)
> → GAS dominates; PAS provides complementary gains via unlabeled data
>
> Severe domain shift (Session 4, CT→MRI)
> - GAS struggles: 0.030→0.025 (Table 3)
> - PAS: 0.025 → 0.058 (Table 7); further improves to 0.197 with 42 unlabeled samples (Table 21)
> →  PAS dominates under severe modality shift
>
> SS-Natural-JASCL (Table 6 and Fig.5)
> - Session 1 (BDD→Cityscapes, large shift): GAS 1.04→26.00, PAS 26.00→27.84
> - Session 3 (IDD→IDD, only Class increment): GAS 0.43→25.12, PAS 25.12→25.47
> → PAS scales with domain shift magnitude
>
> GAS prevents few-shot overfitting; PAS handles strong domain shift with unlabeled data, both complementary. Such constraints absent in CISS benchmarks
>
> **3.**
> Table 10 (Appendix D) shows stable performance near chosen threshold values, with shared thresholds across datasets confirming generalization.
>
> **4.**
> Train cost (per session) of recent methods:
>
> a).
> Table 2 (time):
> YoooP : 4h 03m
> Adapt_replay : 4h 32m
> STAR : 4h 13m
> C-Flat : 4h 08m
> UCB : 3h 23m
> JASCL : 4h 06m
>
> VRAM: 15GB; FLOPs: 0.52T
>
> b).
> Table 7 (time):
> CSL : 5h 10m
> UaD-CE : 5h 18m
> NNCSL : 5h 25m
> JASCL : 5h 18m
> VRAM: 19GB; FLOPs: 1.10T
>
> JASCL has comparable train cost with higher performance. VRAM/FLOPs are the same (Refer to Table 19).
>
> **5.**
> For domain shift: (a) Transformer backbones maintain Dice 0.323 at Session 4, Table 7, vs near-zero for U-Net, and (b) Unlabeled data via PAS (Dice 0.025 to 0.197 with 42 samples at Session 4, Table 21 (significant improvement)). See appendix J.1, L.1.
>
> We will include details in the camera-ready copy.

---

> > ### Author Rebuttal · Reviewer_rwDq · 2026-04-03
> >
> > My concerns have been addressed.

---

> > > ### Author Response · Authors · 2026-04-03
> > >
> > > Dear Reviewer rwDq,
> > >
> > > Thank you sincerely for your thorough and thoughtful engagement throughout the review process. Your time, careful consideration, and constructive feedback are greatly appreciated. We are glad that all concerns have been fully resolved. Given that all concerns are addressed, we kindly request your consideration of a higher score.

---

### Decision · Program_Chairs · 2026-04-30

**Decision:**

Accept (regular)

**Comment:**

This paper tackles continual segmentation in which class, domain, and supervision shifts occur together, and reviewers generally agree that this joint nonstationarity formulation is valuable. The main concerns were about readability, clearer attribution of gains to each component, and whether the strongest claims about plasticity are fully supported, especially in the hardest medical scenarios. After the rebuttal, however, most reviewers' concerns were resolved, and the authors provided additional clarification. While one reviewer remained skeptical about the absolute strength of the results on new classes, the paper consistently outperforms existing baselines under this much harder setting, which the AC views as the more important aspect. Overall, this is a meaningful contribution, and the AC would like to recommend positively.